# CompeteSMoE – Statistically Guaranteed Mixture of Experts Training via Competition

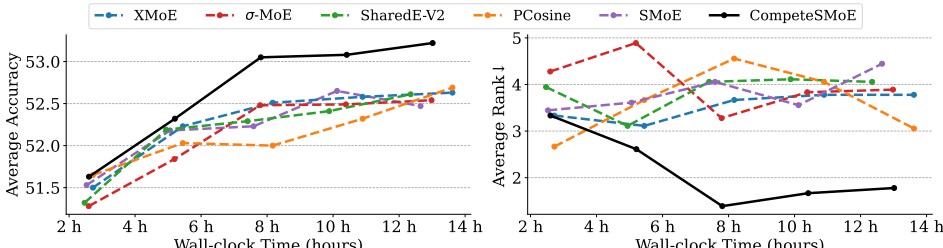

Figure 1: The evolution of zero-shot performance averaged over nine visual instruction tuning tasks throughout training of various SMoE algorithms using a 5.1B parameters backbone.

## ABSTRACT

Sparse mixture of experts (SMoE) offers an appealing solution to scale up the model complexity beyond the mean of increasing the network's depth or width. However, we argue that effective SMoE training remains challenging because of the suboptimal routing process, which often does not involve the experts computation. In this work, we propose *competition*, a novel mechanism to route tokens to experts with the highest neural response. Theoretically, we show that the competition mechanism enjoys a better sample efficiency than the traditional softmax routing. Furthermore, we develop CompeteSMoE, a simple yet effective algorithm for large models by deploying a router to learn the competition policy, thus enjoying strong performances at a low training overhead. Our extensive empirical evaluations on both the visual instruction tuning and language pre-training tasks demonstrate the efficacy, robustness, and scalability of CompeteSMoE compared to state-of-the-art SMoE strategies. We will publish the implementation upon acceptance.

## 1 INTRODUCTION

Large language models (LLMs) have emerged as a promising architecture for artificial general intelligence. In recent years, LLMs have shown remarkable success in solving many cognitive tasks, ranging from language, vision understanding (Bao et al., 2022b; Gulati et al., 2020; Dosovitskiy et al., 2021; Ruiz et al., 2021; Bao et al., 2022a; Li et al., 2022; 2023a), to code generation (Wang et al., 2021), reinforcement learning (Chow et al., 2023) and life sciences (Rives et al., 2021). Since the release of the original Transformer model (Vaswani et al., 2017), extensive efforts have been devoted to scaling the model complexity to take advantage of massive datasets and advanced computing hardware (Radford et al., 2019; Brown et al., 2020; Du et al., 2022). To go beyond simply increasing the depth and width of the network, Sparse Mixture-of-experts (SMoE) (Fedus et al., 2022) has emerged as an appealing solution for scaling LLMs. By modularizing the network and activating only subsets of experts per input, SMoE offers constant computational costs when increasing the model complexity and often resulting in improved performance.

Despite the initial success, practical SMoE training has been known to be notoriously challenging in both engineering and algorithmic aspects. Thus, despite the rapid development of advanced SMoE research in theory and algorithm (Lee-Thorp & Ainslie, 2022; Riquelme et al., 2021; Chi et al., 2022a), limited progress has been made in leading industrial models such as DeepSeek (DeepSeek-AI et al., 2024; 2025) or Phi-MoE (Abdin et al., 2024) as they still implement variants of the vanilla

routing mechanism since the original SMoE (Shazeer et al., 2017; Lepikhin et al., 2021; Fedus et al., 2022). We argue that this discrepancy exists because many state-of-the-art strategies often rely on intuitive conceptualizations, which can only offer greedy solutions that work training in the limited training data and small model regimes. Evidently, many of existing works (Le et al., 2025; Do et al., 2023; Nguyen et al., 2025; Dai et al., 2022a) still follow the in-domain evaluation and ignore the zero-shot generalization capabilities of pre-train language models, which are their main use cases.

This work makes a step towards a statistically guaranteed SMoE training strategy that can yield improvements over a wide range of training settings in large-scale models. To this end, we investigate the core mechanism of routing tokens to experts in SMoE and argue that it could be suboptimal because the experts performing the calculation do not directly contribute to the routing process. This limitation has motivated us to develop a radical routing strategy to distribute tokens to experts more effectively than using the traditional router. To this end, motivated by the Winner-take-all (WTA) principle (Grossberg & Grossberg, 1982; Riesenhuber & Poggio, 1999; Andersen et al., 1969; Eccles, 2013), we propose the *competition* mechanism for SMoE training. The core mechanism of competition is activating all experts and defining a winning criterion so that tokens are only sent to the winning experts. Thus, competition addresses the fundamental limitation of traditional routing schemes by involving experts in the routing process, which we rigorously show to achieve a better sample efficiency or convergence rate than the traditional softmax routing. Furthermore, we go beyond statistical analysis by developing the CompeteSMoE algorithm that implements the competition mechanism into large-scale models at a modest overhead. Specifically, CompeteSMoE improved the zero-shot performance across 16 common benchmarks in both vision-language finetuning (Figure 1) and language pre-training settings.

In summary, our work makes the following contributions. First, we propose a novel *competition* mechanism for training SMoE, which enjoys a better convergence rate than softmax routing. Second, we develop *CompeteSMoE*, a scalable and effective training strategy for SMoE training via competition. Lastly, we conduct extensive experiments to explore the behaviours of CompeteSMoE, including its performance, scalability, convergence property, and routing efficacy.

## 2 COMPETESMOE

We first recap the foundation of MoE in Section 2.1. Then, we introduce the competition mechanism in Section 2.2, discuss the scheduled router training in Section 2.3, and detail the CompeteSMoE algorithm in Section 2.4. We provide a list of all notations and their meanings in Table 9, Appendix A.

### 2.1 BACKGROUND

The traditional SMoE layer (Shazeer et al., 2017) consists of a router $\mathcal{R}(\cdot, W_r)$ parameterized by $W_r$ and $N$ experts $\{g(\cdot, W_{e_i})\}_{i=1}^N$ parameterized by $W_{e_i}, i \in [N]$, respectively. The router takes the input token $\boldsymbol{x}$ as input and produces an affinity score vector on experts as $\boldsymbol{s}_{\mathcal{R}} = \sigma(\text{TopK}_{-\infty}(\boldsymbol{x}^\top W_r))$, where $\sigma$ is a scoring function, often implemented as a softmax or sigmoid function. The $\text{TopK}_{-\infty}$ function keeps the largest $K$ elements in a vector and sets the other elements to negative infinity $(-\infty)$. With this notation, the SMoE layer takes an input token $\boldsymbol{x}$ and calculate the final output by aggregating the outputs of each expert weighted by their affinity scores as: $\hat{y} = \sum_{i=1}^N \boldsymbol{s}_{\mathcal{R}}^i \cdot g(\boldsymbol{x}; W_{e_i})$ In practice, it is common for $K$ to be smaller than $N$, i.e. $K < N$, to improve the model efficiency.

### 2.2 ROUTING VIA COMPETITION

We now introduce the *competition* mechanism as an effective routing strategy to facilitate SMoE training. The key idea of competition is allowing all experts to calculate their outputs, and selection is performed via the winner-take-all mechanism. Thus, experts will compete with one another and the best ones are selected to calculate the final output. To implement the competition, we propose to use the expert's neural response as its affinity score, i.e. $s_i = \mathbb{E}[\kappa(g(\boldsymbol{x}, W_{e_i}))]$, where $\kappa(\cdot)$ is an activation function over the expert's neural responses, and $\mathbb{E}$ denotes the mean over the elements of the expert's output vector. In the experiments, we implement $\kappa$ as the softplus function, unless otherwise stated. However, our competition mechanism and the theoretical analysis thereafter are general and do not make strong assumptions about $\kappa$. We will provide the results of other choices of $\kappa$ in Appendix C.3. With this notation, the training of SMoE with competition is formulated via the following steps:

1. Compute the output of all $N$ experts for a given input $\boldsymbol{x}$ as $g(\boldsymbol{x}, W_{e_i})$, $\quad \forall i \in [N]$.

2. Compute the affinity score of each expert: $\boldsymbol{s}_i = \mathbb{E}[\log(1 + e^{g(\boldsymbol{x}, W_{e_i})})]$, $\quad \forall i \in [N]$.

3. Select the Top-$K$ experts based on the highest neural response and compute the normalized affinity scores: $\hat{\boldsymbol{s}}_{\mathcal{C}}^i = \text{TopK}_0(s_i, K)$, $\boldsymbol{s}_{\mathcal{C}}^i = \frac{\hat{\boldsymbol{s}}_{\mathcal{C}}^i}{\sum_{j=1}^{N} \hat{\boldsymbol{s}}_{\mathcal{C}}^j}$. Here, $\text{TopK}_0$ is similar to the traditional $\text{TopK}_{-\infty}$ but sets the values outside the $K$ highest values to be 0 instead of $-\infty$.

4. Compute the final output as a weighted sum of the selected experts:
   $\hat{y} = \sum_{i=1}^{N} \boldsymbol{s}_{\mathcal{C}}^i \cdot g(\boldsymbol{x}, W_{e_i})$.

Competition starkly contrasts with the standard SMoE implementation discussed in Section 2.1 where the affinity score is calculated as the dot product between the input $\boldsymbol{x}$ and the columns of the router $W_r$, then only a few selected experts actually perform their calculation. Although efficient, it results in suboptimal routing policies because the expert selection is detached from the expert's forward calculation. In contrast, competition proposes that experts who respond the strongest to an input are selected to process that input, while suppressing the other experts. We will rigorously show the theoretical guarantees of routing via competition in Section 3.

## 2.3 Scheduled Training of the Router

One drawback of competition-based expert selection is the high computational overhead of activating all experts, which limits its viability to large-scale models. To make competition applicable to LLM training, we propose to incorporate it into the standard router in SMoE. Specifically, we propose a schedule training procedure that periodically trains the router $\mathcal{R}(\cdot; W_r)$ to jointly estimate the competition policy and minimize the task loss. It is important to note that our analysis in Section 3 will show that using the competition policy alone is theoretically sufficient to achieve a faster convergence rate than the vanilla SMoE. In practice, CompeteSMoE in modern architectures stacks many SMoE layers on top of each other, each of which is equipped with a competition mechanism independently. This deep architecture may require significantly more training samples for convergence, which could be much larger than the dataset size and makes training infeasible on our hardware. Therefore, we propose to jointly learn the task loss and match the competition policy to facilitate the router learning. Particularly, without competition activated, the task loss gradient tells the router how to adjust the affinity scores for selected experts only (since inactivated experts do not receive gradients). When competition is active, its gradient tells the router how to adjust the scores for all experts, including those that are not selected to make final predictions. Thus, CompeteSMoE router is expected to facilitate the training and improve the performance. In the following, we present the router loss for effective training and the router schedulers to ensure that training remains efficient.

### 2.3.1 Router Loss

The router is trained to learn the competition policy and use it to minimize the task loss. We propose to learn the competition policy by minimizing a distillation loss, $\mathcal{L}_{\mathcal{D}}$, which characterizes the discrepancy between the competition and router policies. For ease of notation, we use $I_{\mathcal{C}} \subset [N]$ to denote the indices of the experts who won the competition. Then, the distillation loss $\mathcal{L}_{\mathcal{D}}$ can be computed by minimizing the mean squared errors (MSE) between the competition and router policies, via their affinity scores as:

$$\mathcal{L}_{\mathcal{D}}(\boldsymbol{s}_{\mathcal{R}}, \boldsymbol{s}_C) = \text{MSE}(\boldsymbol{s}_{\mathcal{R}}, \boldsymbol{s}_C) + \frac{\alpha}{K} \cdot \sum_{j \in I_c} (\boldsymbol{s}_C^j - \boldsymbol{s}_{\mathcal{R}}^j)^2, \tag{1}$$

where $\alpha \in \mathbb{R}^+$ is a hyperparameter to encourage the router to pay more attention to winning experts.

**Diversity Loss** One of our main experimental settings is using sparse upcycling (Komatsuzaki et al., 2023) to bypass the expensive pre-training cost, which allows us to test SMoE algorithms on larger models with a low budget. However, sparse upcycling duplicates the experts and make them have similar outputs, which results in no competition in the early stages of training and limited training efficacy. To mitigate this issue, we introduce the Diversity Loss, $\mathcal{L}_{\text{div}}$, to promote diverse representations of the winning experts. Formally, given the output matrix $O \in \mathbb{R}^{K \times D}$ representing the outputs of $K$ winning experts for an input $\boldsymbol{x}$, the diversity loss is computed as the mean of the

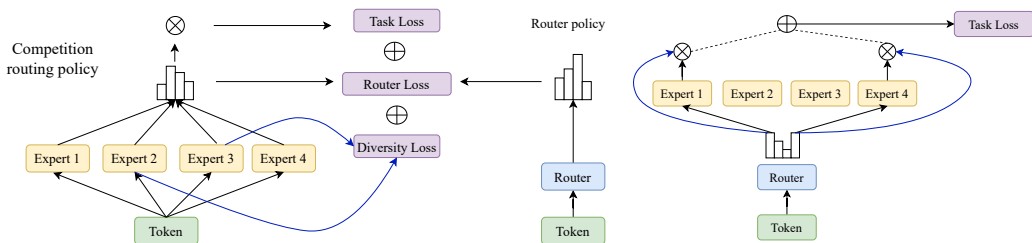

(a) The router learns the competition policy.  (b) Normal routing using the router.

Figure 2: An illustrative of the interleaved learning phases in CompeteSMoE: (a) activating all experts for the router to learn the competition policy; and (b) normal routing using the router.

off-diagonal elements in the correlation matrix constructed from $O$:

$$\mathcal{L}_{\mathrm{div}}(O) = \frac{1}{K(K-1)} \sum_{i=1}^{K} \sum_{\substack{j=1 \\ j \neq i}}^{K} C_{i,j}, \text{ where } C = \frac{O \cdot O^{\top}}{\|O\|_2^2}. \tag{2}$$

We apply the Diversity Loss only within the competition mechanism and emphasize the winning experts as defined in Eq. 2.2, rather than those selected by the router $\mathcal{R}(\cdot; W_r)$. By penalizing winning experts when they produce similar outputs, $\mathcal{L}_{\mathrm{div}}$ promotes a more effective competition outcome when using the sparse upcycling strategy.

### 2.3.2 ROUTER TRAINING SCHEDULE

Schedulers are essential to ensure that the routers can effectively learn a good routing policy while maintaining a limited computational overhead. In the worst case, when all layers of a deep network perform competition simultaneously, this SMoE becomes dense and could crash the training process. Thus, we need to carefully design a schedule to manage the competition frequency across layers. To this end, we employ two schedulers; one is applied per layer independently, while the other monitors the total competition frequency of all layers. For a layer $l$ in a deep network, we first employ a scheduler $\lambda_l(t)$ to determine whether competition should be activated at time step $t$ for this layer. We simply implement $\lambda_l(t)$ by sampling from a Bernoulli distribution with probability $\omega$, which is fixed for all layers. Furthermore, we also employ a global concurrency across layers. Specifically, we only allow the total number of layers performing competition at any time step to be $A_{\max}$. Any layers exceeding this threshold are deferred to perform competition in the next step. Appendix B will provide a detailed formulation of the global scheduler. Based on the number of layers in each model, we set $A_{\max} = 9$ for vision-language models and $A_{\max} = 6$ for the language model pre-training setting, in order to achieve an optimal trade-off between performance and computational feasibility.

### 2.4 THE COMPETESMOE ALGORITHM

We are now ready to describe the CompeteSMoE algorithm to enhance SMoE training of large-scale models. Before training, we use the schedulers to generate all time steps for which the competition mechanism is activated at each layer and store them in $\{\Lambda(l)\}_{l=1}^{L}$, where $\Lambda(l, t) = 1$ indicating that the $l-$layer will perform competition at time $t$. Note that this step is performed offline, only one time before training starts. Then, according to the schedule $\Lambda(l, t)$, the training dynamic involves: (i) training the activated experts to minimize the task loss, $\mathcal{L}_{\mathrm{NLL}}$, and (ii) training the activated router to minimize the task and router losses. We provide an illustration of CompeteSMoE training in Figure 2.

We now discuss a general guideline to set the hyper-parameters introduced by CompeteSMoE. We recommend the balancing hyper-parameters $\alpha, \beta, \gamma$ to be small values such as $0.01$ or $0.005$. The Bernoulli parameter $\omega$ should also be small (e.g. $0.07$) so that competition is not activated too often. The global scheduler threshold should be set based on the specific backbone architecture, model, and training infrastructure to ensure stability. We found $A_{\max} = 9$ for vision-language models and $A_{\max} = 6$ for language model pre-training to maximize the memory usage of our hardware. Lastly, we emphasize that the value ranges of these hyper-parameters can be derived by their definition,

which greatly reduces the effort for hyper-parameter searching. As long as they follow this guideline, we empirically validate the effectiveness of these guidelines through an extensive ablation study in Appendix C, showing that they consistently lead to strong and stable performance in all settings.

# 3 STATISTICAL GUARANTEE OF THE COMPETITION MECHANISM

In this section, we perform a convergence analysis of Gaussian MoE models equipped with the competition mechanism. Our primary objective is to theoretically justify the effectiveness of the competition mechanism by investigating its sample efficiency in terms of expert estimation.

**Problem setting.** Let $(X_1, Y_1), (X_2, Y_2), \ldots, (X_n, Y_n) \in \mathcal{X} \times \mathcal{Y}$ be i.i.d samples drawn from bounded subsets $\mathcal{X} \subset \mathbb{R}^{d_1}$ and $\mathcal{Y} \subset \mathbb{R}$ according to the following conditional density function:

$$p_{G_*}(Y|X) := \sum_{i=1}^{N^*} \frac{\exp(\log(1 + \exp(g(X, W_{e_i}^*))))}{\sum_{j=1}^{N^*} \exp(\log(1 + \exp(g(X, W_{e_j}^*))))} \cdot f(Y|g(X, W_{e_i}^*), \nu_i^*). \tag{3}$$

Here, $N^*$ is the number of ground-truth experts denoted by $g(X, W_{e_i}^*)$, while $f(\cdot|\mu, \nu)$ stands for the Gaussian density with mean $\mu$ and variance $\nu$. In addition, we also define $G_* := \sum_{i=1}^{N^*} \delta_{(W_{e_i}^*, \nu_i^*)}$ as a mixing measure with ground-truth parameters $(W_{e_i}^*, \nu_i^*)$, where $\delta$ denotes the Dirac measure. For the sake of theory, we assume that $(W_{e_1}^*, \nu_1^*), (W_{e_2}^*, \nu_2^*), \ldots, (W_{e_{N^*}}^*, \nu_{N^*}^*)$ are distinct parameters belonging to a compact space $\Theta \subset \mathbb{R}^{d_2} \times \mathbb{R}_+$ for some $d_2 \in \mathbb{N}$. Next, we assume that the expert function $g(X, W_e)$ is non-zero and differentiable with respect to its parameter $W_e$ for almost surely $X$. Furthermore, for any parameter $W_e \in \mathbb{R}^{d_2}$, if there exists $\alpha_1^{(u)}, \alpha_2^{(uv)}, \alpha_3^{(uv)} \in \mathbb{R}$ for $1 \leq u, v \leq d_2$ such that $\sum_{u=1}^{d_2} \alpha_1^{(u)} \frac{\partial g}{\partial W_e^{(u)}}(X, W_e) + \sum_{u,v=1}^{d_2} \alpha_2^{(uv)} \frac{\partial^2 g}{\partial W_e^{(u)} \partial W_e^{(v)}}(X, W_e) + \sum_{u,v=1}^{d_2} \alpha_3^{(uv)} \frac{\partial g}{\partial W_e^{(u)}}(X, W_e) \frac{\partial g}{\partial W_e^{(v)}}(X, W_e) = 0$ for almost surely $X$, then we must have $\alpha_1^{(u)} = \alpha_2^{(uv)} = \alpha_3^{(uv)} = 0$ for all $1 \leq u, v \leq d_2$. For example, it can be verified that feed-forward networks (FFNs) of the form $g(X, (W_{e,2}, W_{e,1}, b)) = W_{e,2} \text{Softplus}(W_{e,1}^\top X + b)$ we used in Section 2.2 satisfy this algebraic independence condition. On the other hand, since linear experts $g(X, (a, b)) = a^\top X + b$ does not meet this condition, we will conduct a separate convergence analysis for them in Appendix J.

**Maximum likelihood estimation.** Since the number of ground-truth experts $N^*$ is typically unknown in practice, we fit the model equation (3) with a mixture of $N > N^*$ experts. Then, we estimate the unknown parameters $(W_{e_i}^*, \nu_i^*)$, for $1 \leq i \leq N$, via estimating the ground-truth mixing measure $G_*$ using the maximum likelihood method as follows:

$$\widehat{G}_n \in \underset{G \in \mathcal{G}_N(\Theta)}{\arg\max} \frac{1}{n} \sum_{i=1}^{n} \log(p_G(Y_i|X_i)), \tag{4}$$

where we define $\mathcal{G}_N(\Theta) := \{G = \sum_{i=1}^{N'} \delta_{(W_{e_i}, \nu_i)} : 1 \leq N' \leq N, (W_{e_i}, \nu_i) \in \Theta\}$.

**Proposition 3.1.** *With the MLE defined in equation (4), the convergence rate of the density estimation* $p_{\widehat{G}_n}(Y|X)$ *to the ground-truth density* $p_{G_*}(Y|X)$ *is given by:*

$$\mathbb{E}_X[V(p_{\widehat{G}_n}(\cdot|X), p_{G_*}(\cdot|X))] = \mathcal{O}_P(\sqrt{\log(n)/n}),$$

*Above, we denote* $V(p_1, p_2) := \frac{1}{2} \int |p_1 - p_2| \mathrm{d}m$ *as the Total Variation distance between two probability density functions* $p_1, p_2$ *dominated by the Lebesgue measure* $m$.

The proof of Proposition 3.1 can be found in Appendix K.3. The above result indicates that the density estimation $p_{\widehat{G}_n}$ converges to its true counterpart $p_{G_*}$ at a parametric rate of order $\widetilde{\mathcal{O}}_P(n^{-1/2})$. Thus, if we can construct some loss function between two mixing measures $\widehat{G}_n$ and $G_*$, denoted by $\mathcal{L}(\widehat{G}_n, G_*)$, such that $\mathbb{E}_X[V(p_{\widehat{G}_n}(\cdot|X), p_{G_*}(\cdot|X))] \gtrsim \mathcal{L}(\widehat{G}_n, G_*)$, then we will obtain parameter and expert estimation rates via the bound $\mathcal{L}(\widehat{G}_n, G_*) = \mathcal{O}_P(\sqrt{\log(n)/n})$. For that purpose, let us introduce the concept of Voronoi loss proposed in Manole et al. Manole & Ho (2022).

**Voronoi loss.** For an arbitrary mixing measure $G$, we distribute its atoms to the following Voronoi cells generated by the support points of the ground-truth mixing measure $G_*$:

$$\mathcal{C}_j \equiv \mathcal{C}_j(G) := \{i \in [N] : \|\theta_i - \theta_j^*\| \leq \|\theta_i - \theta_\ell^*\|, \forall \ell \neq j\}, \tag{5}$$

where we denote $\theta_i := (W_{e_i}, \nu_i)$ and $\theta_j^* := (W_{e_j}^*, \nu_j^*)$ for all $i \in [N]$ and $j \in [N^*]$. Here, the cardinality of each Voronoi cell $\mathcal{C}_j$ indicates the number of fitted atoms for the ground-truth atom $\theta_j^*$. Then, we build a loss function based on these Voronoi cells as follows:

$$\mathcal{L}_1(G, G_*) := \sum_{j=1}^{N^*} \Big| \sum_{i \in \mathcal{C}_j} \exp(c_i) - \exp(c_j^*) \Big| + \sum_{j \in [N^*]: |\mathcal{C}_j|=1} \sum_{i \in \mathcal{C}_j} \exp(c_i) \Big[ \|W_{e_i} - W_{e_j}^*\| + |\nu_i - \nu_j^*| \Big]$$

$$+ \sum_{j \in [N^*]: |\mathcal{C}_j|>1} \sum_{i \in \mathcal{C}_j} \exp(c_i) \Big[ \|W_{e_i} - W_{e_j}^*\|^2 + |\nu_i - \nu_j^*|^2 \Big]. \quad (6)$$

Given the above Voronoi loss, we are ready to capture the convergence rates of parameter estimation and expert estimation in Theorem 3.2 whose proof can be found in Appendix K.1.

**Theorem 3.2.** *The following lower bound holds for any mixing measure $G \in \mathcal{G}_N(\Theta)$:*

$$\mathbb{E}_X[V(p_G(\cdot|X), p_{G_*}(\cdot|X))] \gtrsim \mathcal{L}_1(G, G_*). \quad (7)$$

*This lower bound and the result of Theorem 3.1 imply that $\mathcal{L}_1(\widehat{G}_n, G_*) = \mathcal{O}_P(\sqrt{\log(n)/n})$.*

A few remarks regarding Theorem 3.2 are in order.

*(i) Expert estimation rates.* From the above results and the formulation of the Voronoi loss $\mathcal{L}_1$, it follows that the rates for estimating exact-specified parameters $W_{e_j}^*, \nu_j^*$, i.e., for $j \in [N^*] : |\mathcal{C}_j| = 1$, are of parametric order $\widetilde{\mathcal{O}}_P(n^{-1/2})$. Meanwhile, those for over-specified parameters $W_{e_j}^*, \nu_j^*$, i.e., for $j \in [N^*] : |\mathcal{C}_j| > 1$, are slightly slower, of order $\widetilde{\mathcal{O}}_P(n^{-1/4})$. Since the expert function $g(X, W_e)$ is Lipschitz continuous w.r.t its parameter $W_e$, we have $|g(X, \widehat{W}_{e_i}^n) - g(X, W_{e_j}^*)| \lesssim \|\widehat{W}_{e_i}^n - W_{e_j}^*\|$ for almost surely $X$. As a result, the estimation rates for exact-specified and over-specified experts $g(X, W_{e_j}^*)$ are also of orders $\widetilde{\mathcal{O}}_P(n^{-1/2})$ and $\widetilde{\mathcal{O}}_P(n^{-1/4})$, respectively. Furthermore, we show in Appendix J that experts of linear form $g(X, (a, b)) = a^\top X + b$ also admit these estimation rates.

*(ii) Sample efficiency of the competition mechanism.* Therefore, we need at most $\mathcal{O}(\epsilon^{-4})$ data points to approximate these experts with a given error $\epsilon > 0$. On the other hand, when not using the competition mechanism Nguyen et al. (2023a), the convergence rates of expert estimation become significantly slow and decrease when the number of fitted experts increases. For instance, if an expert $g(X, W_{e_j}^*)$ is fitted by three experts, i.e., $|\mathcal{C}_j| = 3$, then its estimation rate is of order $\widetilde{\mathcal{O}}_P(n^{-1/12})$. Thus, we need much more data points, specifically $\mathcal{O}(\epsilon^{-12})$, to approximate this expert. Consequently, we conclude that the competition mechanism improves the sample efficiency in terms of expert estimation.

## 4 RELATED WORK

Mixture of Experts (MoE) is a fundamental model in machine learning (Jacobs et al., 1991; Jordan & Jacobs, 1994) and an instance of the conditional computation framework where different experts are responsible for different regions of the input space (Yuksel et al., 2012; Bengio, 2013; Masoudnia & Ebrahimpour, 2014; Nguyen & Chamroukhi, 2018; Nguyen, 2021). Extensive efforts have been devoted to establishing a theoretical foundation for MoE, including the universal approximation properties (Norets, 2010; Nguyen et al., 2016; 2019; 2020; 2021a; 2023b), model selection criterion (Khalili, 2010; Montuelle & Le Pennec, 2014; Nguyen et al., 2021b; 2022; 2023c), convergence rate for density estimations (Mendes & Jiang, 2012; Norets & Pelenis, 2021; 2022) and the problem of parameter estimation (Ho et al., 2022; Nguyen et al., 2023a; 2024b;a). SMoE, the sparse variant of MoE, is more commonly applied to scale large language models (Fedus et al., 2022). It is often the architecture of choice in many leading industrial models such as Mixtral (Jiang et al., 2024) and DeepSeek (Dai et al., 2024; DeepSeek-AI et al., 2024; 2025). Within the research community, developing novel routing strategies has been a major focus. Notable strategies include letting experts select tokens (Zhou et al., 2022), improving the expert selection process (Lepikhin et al., 2021; Fedus et al., 2022; Zuo et al., 2022; Chi et al., 2022a; Dai et al., 2022b; Chen et al., 2023; Do et al., 2023), or a global expert assignment scheme(Lewis et al., 2021; Clark et al., 2022). Despite the promising progress, many such strategies often do not scale well to LLMs with billions of parameters or the language pre-training setting. In contrast, our work goes beyond both the pure theoretical or analytical studies by developing a theoretically-grounded algorithm for effective training of large-scale LLM

models inspired by competitive learning, which models neural systems where only the most responsive units activate while suppressing others (McClelland et al., 1987). This principle has historically driven progress across diverse learning paradigms, including self-organization (Von der Malsburg, 1973; Kohonen, 1982), feature discovery (Rumelhart & Zipser, 1985), spiking models (Oster & Liu, 2005), and modern competitive architectures such as maxout networks (Goodfellow et al., 2013) and compete-to-compute (Srivastava et al., 2013). Orthogonal to the aforementioned papers, GShard (Lepikhin et al., 2021) developed an efficient framework to automatically sharding massive SMoE models across many devices. Lastly, sparse upcycling (Komatsuzaki et al., 2023) duplicated pre-trained models to build an MoE, which bypasses the expensive costs of training from scratch.

## 5 EXPERIMENT

### 5.1 EXPERIMENTAL SETTINGS

**Tasks.** We evaluate all methods on two challenging tasks: (i) visual instruction tuning (VIT) and (ii) language model pretraining. For VIT, we adopt the CuMo (Li et al., 2024) and LibMoE framework (Nguyen et al., 2024c), which follows a three-stage training pipeline: pre-training (PT), pre-finetuning (PFT), and visual instruction tuning (VIT). The first two stages are trained with a dense model. In the final VIT stage, sparse upcycling (Komatsuzaki et al., 2023) is applied by resuming from the PFT checkpoint and replacing selected MLP layers in the vision encoder and connector with SMoE blocks. Since one can easily replace the LLMs by other MoE models, the SMoE components are only the vision encoder and the vision-language connector. For the language pretraining task, we adopt the MoEUT (Csordás et al., 2024) framework under the **large setting** and train the SMoE models from scratch. While pre-training has been commonly explored for benchmarking SMoE algorithms (Csordás et al., 2024), it is expensive to scale to large models. Thus, we include the VIT task, which is an emerging and challenging setting that take advantage of pre-training checkpoints, allowing us to evaluate SMoE at a modest cost.

**Training.** For VIT, we follow Li et al. (2024) to use the LLaVA-558K (Liu et al., 2023a) for PT, ALLaVA (Chen et al., 2024a) for PFT, and LLaVA-665K (Liu et al., 2024a) for VIT. The total tokens for all stages is over 1B. We use Phi-3.5 Mini (Abdin et al., 2024) as the language model and SigLIP (Zhai et al., 2023) as the vision encoder, totaling 5.1B parameters. All MoE algorithms are applied during the VIT stage. We set $N = 4$ experts per layer and activate $K = 2$ experts per token. Training default uses both the balancing and z-losses (Fedus et al., 2022). For language pre-training, we follow MoEUT (Csordás et al., 2024) and use 13B tokens from the SlimPajama corpus (Soboleva et al., 2023). We implement a 1B-parameter decoder-only model, where each SMoE layer contains 24 experts with $K = 8$ experts active per token, and the balancing loss (Fedus et al., 2022). All experiments are conducted on 4×H100 GPUs with a fixed random seed.

**Evaluation Benchmarks.** All models are evaluated under the zero-shot settings using the well-established benchmarks from the community. For the VIT task, we consider the following benchmarks: AI2D (Kembhavi et al., 2016), TextVQA Validation (Singh et al., 2019), GQA (Hudson, 2019), HallusionBench (Guan et al., 2023), MathVista (test-mini split) (Lu et al., 2023), MMBench (English subset, dev version) (Liu et al., 2023b), MME RealWorld Lite (Zhang et al., 2025b), MMMU Validation (Yue et al., 2023), MMStar (Chen et al., 2024b), POPE (Li et al., 2023b), and OCR-Bench (Liu et al., 2024b). For benchmarks requiring GPT-based evaluation, such as MathVista and HallusionBench, we use GPT-4o (2024-08-06). These benchmarks are selected to cover a wide range of capabilities of the model, from perception, reasoning, to assessing hallucination. For the language pretraining task, we evaluate on LAMBADA (Paperno et al., 2016), BLiMP (Warstadt et al., 2023), Children's Book Test (CBT) (Zhang et al., 2025a), HellaSwag (Zellers et al., 2019), PIQA (Bisk et al., 2019), ARC-Easy (Clark et al., 2018), RACE (Lai et al., 2017), and SIQA (Sap et al., 2019), which are commonly used for models at this scale.

**Baseline.** We compare CompeteSMoE against a suite of state-of-the-art SMoE algorithms. First, SMoE (Fedus et al., 2022), the original SMoE and still stands strong in today's leading models. Then, we consider activation-based SMoE such as XMoE (Chi et al., 2022b), Perturbed Cosine Router (PCosine) (Nguyen et al., 2025), and $\sigma$-MoE (Csordás et al., 2023), which incorporate cosine similarity or sigmoid activation to improve routing efficiency. Furthermore, inspired by the DeepSeek V2 architecture (DeepSeek-AI et al., 2024), we also considered the SharedExpert V2 (SharedE-V2) baseline, which enhances SMoE with one shared expert. Similarly, for the language pretraining task, we also implement the SharedE-V3 baseline, which follows the DeepSeek V3 architecture (DeepSeek-AI et al., 2025). SharedE-V3 replaces the softmax routing in SharedE-V2

with the normalized sigmoid. We use the same hyper-parameter configuration as described above to validate the effectiveness of different SMoE algorithms.

## 5.2 MAIN RESULTS

Table 1: Performance comparison of SMoE strategies in the ViT sparse upcycling setting with a 5B-parameter model. **Bold** values denote the best results, while underlined values indicate the second best. Symbols ↑ and ↓ indicate that higher or lower values are better, respectively.

| Method | AI2D | Text VQA | GQA | MM Bench | Hallusion | Math Vista | MMMU | MMStar | POPE | OCR | MME RWL | Avg. Acc↑ | Avg. Rank↓ |
|---|---|---|---|---|---|---|---|---|---|---|---|---|---|
| SMoE (Fedus et al., 2022) | 65.90 | 41.23 | 60.96 | 70.88 | 39.64 | 31.40 | 42.22 | 40.52 | 86.56 | 32.10 | 31.89 | 49.39 | 4.55 |
| XMoE (Chi et al., 2022a) | 65.19 | 41.14 | 60.63 | 71.31 | **41.22** | 31.50 | **42.89** | **42.60** | 86.12 | 31.30 | 32.51 | 49.67 | 3.50 |
| PCosine (Nguyen et al., 2025) | 65.45 | 41.68 | 61.38 | 71.56 | 40.27 | 30.80 | 42.56 | 41.87 | 86.90 | 30.80 | 32.05 | 49.57 | 3.42 |
| σ-MoE (Csordás et al., 2023) | 65.09 | 41.37 | **61.48** | 71.39 | 41.01 | **31.90** | 41.78 | 42.10 | 86.52 | 32.20 | 30.95 | 49.62 | 3.64 |
| SharedE-V2 (DeepSeek-AI et al., 2024) | 64.93 | 41.53 | 61.15 | 71.05 | 41.20 | 31.20 | 42.56 | 41.44 | 86.08 | 32.40 | 32.36 | 49.63 | 4.05 |
| CompeteSMoE | **66.22** | **41.92** | 61.25 | **72.59** | **41.22** | 31.70 | 42.00 | 42.25 | 86.91 | **33.20** | 32.52 | **50.16** | **1.77** |

Table 2: Performance comparison of SMoE strategies in the language pretraining setting with a 1B-parameter model. **Bold** values denote the best results, underlined values indicate the second best.

| Method | LAMBADA | BLiMP | CBT | HellaSwag | PIQA | ARC-E | RACE | SIQA | Avg. Acc↑ | Avg. Rank↓ |
|---|---|---|---|---|---|---|---|---|---|---|
| SMoE (Fedus et al., 2022) | 41.24 | 80.68 | 90.63 | 39.17 | 65.18 | **39.66** | 34.53 | 38.28 | 53.67 | 3.81 |
| XMoE (Chi et al., 2022a) | 42.23 | 80.40 | 90.44 | 38.63 | 64.04 | 38.60 | 34.26 | 37.31 | 53.24 | 5.88 |
| PCosine (Nguyen et al., 2025) | 41.90 | 80.35 | 90.26 | 38.70 | 63.71 | **39.66** | 34.29 | 38.13 | 53.38 | 5.31 |
| σ-MoE (Csordás et al., 2023) | 42.39 | 80.64 | 90.63 | 39.12 | 64.96 | **39.66** | 33.81 | 38.33 | 53.69 | 3.88 |
| SharedE-V2 (DeepSeek-AI et al., 2024) | 41.65 | 80.65 | 90.63 | **39.57** | 65.73 | 38.60 | 34.71 | 37.89 | 53.75 | 3.75 |
| SharedE-V3 (DeepSeek-AI et al., 2025) | 41.91 | 80.23 | **91.02** | 39.19 | 65.45 | 39.53 | 34.86 | 37.97 | 53.77 | 3.63 |
| CompeteSMoE | **42.66** | **80.92** | 90.91 | 39.35 | 65.91 | 39.20 | 34.91 | 38.43 | **54.04** | **1.75** |

### 5.2.1 PERFORMANCE COMPARISON

We report the results of the VIT and language pre-training settings in Table 1 and Table 2, respectively. In general, we observe that CompeteSMoE offers significant improvements over many benchmarks. In addition, CompeteSMoE demonstrated the best performance in many of the challenging and important capabilities such as real-world visual perception and reasoning (MME RWL), reducing visual hallucination (Hallusion, POPE), OCR (OCRBench) and commonsense reasoning (PIQA, SIQA). Furthermore, we report the evolution of zero-shot performances of VIT benchmarks throughout training in Figure 1. The results showed that CompeteSMoE consistently achieved better results than the baselines throughout training. Notably, CompeteSMoE demonstrated a significant improvement in training efficiency, where the checkpoint at eight hours (8h) already outperformed all baselines at their final checkpoint of 14 hours. Lastly, we emphasize that the improvements of CompeteSMoE can be considered significant in the zero-shot evaluation setting because its power law indicates that reducing (zero-shot) errors requires a substantial increase in data and compute (Hoffmann et al., 2022; Cherti et al., 2023). Since we fixed the training data, the zero-shot improvements observed purely came from the advanced CompeteSMoE algorithm. Overall, the results corroborate our theoretical results that CompeteSMoE achieved a better sample efficiency and better zero-shot generalization.

### 5.2.2 EXPERT ROUTING BEHAVIOR ANALYSIS

Table 3: Performance of SMoE and CompeteSMoE when changing top-1 expert to top-(K+1). Numbers in parentheses indicates the changes compared to the original routing results in Table 1.

| Method | Text VQA | MMBench | MMMU | MMStar | POPE | OCR Bench | Avg. Change |
|---|---|---|---|---|---|---|---|
| SMoE | 41.09 (-0.14) | 71.39 (+0.52) | 43.22 (+1.00) | 42.94 (+2.42) | 86.40 (-0.16) | 31.50 (-0.60) | 0.51 |
| CompeteSMoE | 41.48 (-0.45) | 71.22 (-1.37) | 41.67 (-0.33) | 40.55 (-1.70) | 86.10 (-0.81) | 31.70 (-1.50) | -1.03 |

**(a) Evaluating the Effectiveness of Expert Routing.** We investigate the experts selection's quality of different policies. To this end, during inference, we replace the expert with the highest affinity score with the expert with the $K + 1$ highest score, which is equivalent to shifting the selected experts down by one rank. Table 3 reports the results of this experiment in the VIT setting. The results show that the SMoE routing policy is clearly suboptimal since selecting a worse expert led to improvements on several benchmarks. On the other hand, CompeteSMoE performances drop in all cases when we deliberately deviate from the router that learned the competition policy. This result shows that CompeteSMoE facilitated a more effective routing policy compared to the traditional SMoE.

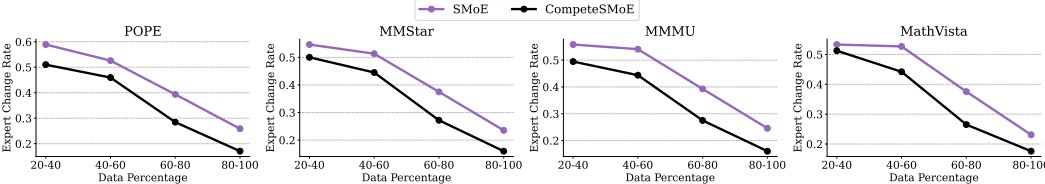

Figure 3: Comparison of expert change rates at different training stages. Lower values are better.

**(b) Stability of Expert Routing During Training.** We now investigate the router's convergence rate, showing that CompeteSMoE can quickly find a good routing policy. To this end, we introduce *Expert Change Rate (ECR)* to measure the convergence rate of routers. Specifically, given a dataset $\mathcal{D}$, we record the expert assignments in all layers for each token in $\mathcal{D}$ using two model checkpoints at time steps $T$ and $T'$. Then, the ECR of $\mathcal{D}$ from $T$ to $T'$ is the number of mismatched assignments normalized by all assignments. We expect ECR at convergence to be low while high ECR values indicate that the router's policy is changing and unstable. Figure 3 reports the ECR throughout training on four VIT zero-shot benchmarks. We can see that CompeteSMoE has a lower ECR in all cases, indicating that its routers have a faster convergence rate. This results further support the faster convergence rate and better performance of CompeteSMoE observed in Figure 1 and Table 1.

## 5.3 COMPLEXITY ANALYSIS

We compare the computational complexities of various methods in Table 4. We report the wall-clock training time, training throughput, inference throughput, and peak GPU memory (excluding cached memory blocks) in the VIT setting of the 5.1B model. The results show that CompeteSMoE's training overhead compared to SMoE is almost negligible. During inference, CompeteSMoE only uses the simple router, which is exactly the same as SMoE, and is more efficient than cosine similarity-based strategies such as XMoE and PCosine because they introduce additional parameters to the router. CompeteSMoE also incurs a slightly higher peak memory usage compared to other baselines (up to +5%), which was affordable by our hardware. In

Table 4: Computation complexities of various SMoE algorithms.

| Method | Training | Throughput *(samples/s)* | | Peak Mem |
|---|---|---|---|---|
| | Time | Train | Infer | (GB) |
| SMoE | 12h39m | 14.59 | 9.87 | 43.86 |
| XMoE | 13h37m | 13.57 | 8.97 | 44.02 |
| $\sigma$-MoE | 12h59m | 14.23 | 9.61 | 43.93 |
| PCosine | 13h37m | 13.57 | 8.59 | 44.12 |
| SharedE-V2 | 12h21m | 14.95 | 9.66 | 42.29 |
| CompeteSMoE | 13h01m | 14.18 | 9.88 | 46.45 |

general, users can adjust the competition's concurrency threshold $A_{\max}$ to achieve a good trade-off between efficiency and efficacy. In summary, this result shows that CompeteSMoE can effectively leverage competition to improve the result at a modest overhead.

## 5.4 HYPERPARAMETER SENSITIVITY ANALYSIS

We conduct a thorough hyperparameter sensitivity analysis of CompeteSMoE, focusing exclusively on the most critical hyperparameters $(\omega, \alpha, \gamma, \beta, A_{max})$. All results are reported on 9 vision–language benchmarks using the 5.1B-parameter VLM.

Table 5: Effect of $\omega$ on competition training.

| $\omega$ | Avg. Acc ↑ | $\Delta$ SMoE |
|---|---|---|
| 3% | 52.81 | +0.34 |
| 5% | 52.92 | +0.45 |
| 7% | **53.21** | **+0.74** |
| 9% | 52.82 | +0.35 |

Table 6: Effect of $\alpha$ on the distillation loss.

| $\alpha$ | Avg. Acc ↑ | $\Delta$ SMoE |
|---|---|---|
| 0.0 | 52.92 | +0.45 |
| 0.1 | **53.21** | **+0.74** |
| 0.2 | 52.98 | +0.51 |
| 0.3 | 52.87 | +0.40 |

**Effect of the Distillation Loss Coefficients.** We analyze the influence of the two hyperparameters in the distillation loss: the main scaling factor $\gamma$ and the auxiliary regularization coefficient $\alpha$. The

Table 7: Effect of $\gamma$ on the distillation loss.

| $\gamma$ | Avg. Acc $\uparrow$ | $\Delta$ SMoE |
|---|---|---|
| 0.001 | 52.74 | +0.27 |
| 0.01 | **53.21** | **+0.74** |
| 0.03 | 52.83 | +0.36 |

Table 8: Effect of $\beta$ on the diversity loss.

| $\beta$ | Avg. Acc $\uparrow$ | $\Delta$ SMoE |
|---|---|---|
| 0.001 | 52.79 | +0.32 |
| 0.005 | **53.21** | **+0.74** |
| 0.01 | 52.91 | +0.44 |

coefficient $\gamma$ controls the overall weight of $\mathcal{L}_{\mathcal{D}}$, while $\alpha$ modulates the strength of the penalty applied to the winning experts. As shown in Table 7 and Table 6, the setting $\gamma = 0.01$ and $\alpha = 0.1$ consistently yields the best results. Notably, $\alpha = 0.1$ provides sufficient regularization to guide the router without dominating the main objective, leading to stable and effective learning.

**Analysis of Competition Mechanism Activation Frequency.** We next examine how frequently the Competition Mechanism (CM) should be activated during training. Table 5 reports performance under different activation frequencies $\omega$. Small values (e.g., $\omega = 3\%$) underperform, likely due to insufficient competitive pressure. Increasing $\omega$ improves performance, with the best accuracy (53.21%) obtained at $\omega = 7\%$. Higher activation rates (e.g., $\omega = 9\%$) do not yield additional gains and may introduce instability, indicating a saturation effect. These results suggest that a moderate activation frequency, with $\omega$ typically in the range of 5%–7%, is optimal for balancing competitive learning with stable training dynamics.

**Effect of the Diversity Loss Coefficient.** The diversity coefficient $\beta$ regulates the dispersion of expert affinity scores by discouraging overly similar outputs among winning experts. As shown in Table 8, model performance peaks at $\beta = 0.005$, which delivers the largest improvement over the SMoE baseline. Smaller values fail to impose sufficient diversity, whereas larger values over-regularize the router and slightly degrade accuracy. Overall, these results indicate that moderate diversity regularization is optimal, promoting balanced expert utilization without interfering with the primary routing objective.

**Key Findings.** Across all four hyperparameters $(\omega, \alpha, \gamma, \beta)$, CompeteSMoE consistently outperforms the standard SMoE baseline even under suboptimal settings. This robustness demonstrates that the method is not overly sensitive to precise hyperparameter tuning and remains stable across a broad range of values. At the same time, the moderate settings we recommend activation frequency $\omega$=5–7%, distillation coefficients $\alpha$=0.1 and $\gamma$=0.01, and diversity coefficient $\beta$=0.005 yield the strongest empirical performance, providing the best balance between competitive pressure, router regularization, and expert diversity. Furthermore, our $A_{\max}$ sensitivity analysis in Appendix G demonstrates that increasing $A_{\max}$ generally leads to more stable and robust model performance, with diminishing returns beyond a moderate threshold. We therefore recommend setting $A_{\max}$ according to the guidelines in Section 2.4, which balance computational efficiency and empirical gains.

## 6 CONCLUSION

This work proposes competition, a novel strategy to route tokens to experts, and rigorously show that it enjoys a better sample efficiency than softmax routing. Building upon this foundation, we develop CompeteSMoE, an effective algorithm to train large-scale SMoE models with competition at a low computational overhead. Extensive experiments on the visual instruction tuning and language pre-training tasks demonstrate that CompeteSMoE enjoys both a faster convergence rate and final performance on many common zero-shot benchmarks at a minimal overhead.

Despite achieving encouraging results, CompeteSMoE introduces several hyper-parameters, which may increase the cost for hyper-parameter search. In Section 2.4, we provided a guideline for hyper-parameter configuration to alleviate this issue. Algorithmically, CompeteSMoE applies competition on each SMoE layer independently and does not take into account the interactions among experts at different layers. An ideal solution is to perform a graph traversal algorithm through the network depth to determine an optimal expert selection at all layers simultaneously. However, this idea goes beyond the scope of this work, and we will leave it for future studies.

## REPRODUCIBILITY STATEMENT

We provide full details of our experimental setup in Section 5 and Appendix H. All necessary code, configuration files are included in the Supplementary Materials. Formal proofs supporting our theoretical claims are presented in Appendix K.

## ETHICS STATEMENT

This work focuses on the fundamental research involving a theoretical analysis and a training strategy for Sparse Mixture-of-Experts architectures. Due to the abstract nature of our study, it does not involve human-subject data, privacy-sensitive content, or downstream applications. As such, we do not foresee any issues with respect to fairness, privacy, or societal harm.

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

# Supplement to "CompeteSMoE – Statistically Guaranteed Mixture of Experts Training via Competition"

This document provides the suppplementary materials for the paper CompeteSMoE – Statistically Guaranteed Mixture of Experts Training via Competition, and is organized as follows.

# Contents

**M Broader Impact** 48

## A SUMMARY OF MAIN NOTATIONS

We summarize the main notations used in the main paper in Table 9, including those introduced later in the supplementary material.

Table 9: Summary of Main Notations.

| Symbol | Description |
|---|---|
| $\mathcal{R}, W_r$ | Router network (function) and its parameter |
| $g, W_e$ | Expert network (function), and its parameter |
| $\boldsymbol{x}$ | Input |
| $\boldsymbol{s}, \boldsymbol{s}_{\mathcal{R}}, \boldsymbol{s}_{\mathcal{C}}$ | Affinity scores, affinity scores from the router, affinity scores from competition |
| $\text{TopK}_{-\infty}$ | Function retaining the $K$ largest vector elements and setting others to $-\infty$ |
| $\text{TopK}_0$ | Function retaining the $K$ largest vector elements and setting others to $0$ |
| $K$ | Number of experts activated per input |
| $N$ | The total number of experts |
| $[M]$ | Set of $\{1, 2, ..., M\}$ for any positive integer $M$ |
| $\hat{y}, y$ | Predicted output, ground truth |
| $t$ | Current $t$-th iteration |
| $T$ | Total number of training steps |
| $l$ | The $l$-th SMoE layer |
| $L$ | Total number of SMoE layers in the model |
| $\kappa$ | Activation function |
| $\sigma$ | Scoring function |
| $\mathbb{E}[\cdot]$ | Mean of vector elements |
| $e$ | Base of the exponential function |
| $I_{\mathcal{C}}$ | Indices of experts who won in the competition mechanism |
| $\alpha$ | Hyper-parameter prioritizing winning experts in distillation loss |
| $\gamma$ | Hyper-parameter for distillation loss |
| $\beta$ | Hyper-parameter for diversity loss |
| $\omega$ | Bernoulli probability for scheduling competition in each layer |
| $A_{\max}$ | Maximum number of layers that can perform competition on a single time step |
| $\lambda(t)$ | A scheduler determining whether to perform competition at the $t$-th step |
| $\Lambda(l)$ | A vector storing the results of the scheduler $\lambda(t)$ at all time steps of the $l$-th layer |
| $\mathcal{L}_{\text{NLL}}$ | Negative log-likelihood function (task loss) |
| $\mathcal{L}_{\mathcal{D}}$ | Distillation loss |
| $\mathcal{L}_{div}$ | Diversity loss |
| $\xi_t$ | Step size |
| $\mathcal{D}$ | A benchmark dataset for evaluation |
| $Q_{\text{prev}}$ | Cumulative competition activations over layers 1 to $l-1$ |
| $a_n = \mathcal{O}(b_n)$ or $a_n \lesssim b_n$ | If $a_n \leq C b_n$ for all $n \in \mathbb{N}$, where $C > 0$ is some universal constant |
| $a_n = \mathcal{O}_P(b_n)$ | $\forall \epsilon > 0, \exists M > 0 : \mathbb{P}(A_n/b_n > M) < \epsilon$ for all sufficiently large $n$ |
| $a_n = \widetilde{\mathcal{O}}_P(b_n)$ | $a_n = \mathcal{O}_P(b_n \log^c(b_n))$, for some $c > 0$. |
| $w^{(u)}, w_u$ | The $u$-th entry of a vector $w \in \mathbb{R}^d$ |
| $w^z$ | $w^z = w_1^{z_1} w_2^{z_2} \ldots w_d^{z_d}$, for any vector $w \in \mathbb{R}^d$ and $z \in \mathbb{N}^d$ |
| $\lvert w \rvert$ | $\lvert w \rvert := w_1 + w_2 + \ldots + w_d$, for any vector $w \in \mathbb{R}^d$ |
| $z!$ | $z! := z_1! z_2! \ldots z_d!$, for any vector $z \in \mathbb{N}^d$ |
| $N^*$ | The number of ground-truth experts |
| $f(\cdot \lvert \mu, \nu)$ | Univariate Gaussian density with mean $\mu$ and variance $\nu$ |
| $G_*$ | Ground-truth mixing measure |
| $\delta$ | Dirac measure |
| $m$ | Lebesgue measure |
| $\Theta$ | Parameter space |
| $d_1$ | Dimension of input space |
| $d_2$ | Dimension of expert parameter space |
| $\widehat{G}_n$ | Maximum likelihood estimator for $G_*$ |
| $\lVert \cdot \rVert, \lVert \cdot \rVert_1$ | $\ell_2$-norm and $\ell_1$-norm value |
| $\lvert A \rvert$ | Cardinality of any set $A$ |
| $h(p_1, p_2)$ | Hellinger distance $h(p_1, p_2) := \left(\frac{1}{2} \int (\sqrt{p_1} - \sqrt{p_2})^2 dm \right)^{1/2}$ for any densities $p_1, p_2$ |
| $V(p_1, p_2)$ | Total Variation distance $V(p_1, p_2) := \frac{1}{2} \int \lvert p_1 - p_2 \rvert dm$ for any densities $p_1, p_2$ |

## B  ADAPTIVE LAYER-WISE COMPETITION CONTROL

While scheduled training reduces computational overhead, excessive simultaneous competition activations across multiple SMoE layers can destabilize the training process. To address this, we propose a dynamic mechanism that regulates the number of active competition layers at each training step, enhancing training efficiency. This is achieved by enforcing a global constraint on the maximum number of simultaneously active layers.

For a given layer $l$, we compute the cumulative competition activations from all preceding layers (i.e., layers 1 through $l-1$) as:

$$Q_{\text{prev}} = \sum_{i=1}^{l-1} \Lambda(i), \tag{8}$$

where $\Lambda(i) \in \mathbb{R}^T$ denotes the activation state vector of layer $i$ over $T$ training steps, and $Q_{\text{prev}} \in \mathbb{R}^T$ represents the cumulative competition activations up to layer $l-1$.

A predefined threshold $A_{\text{max}} \in \mathbb{R}$ governs the total number of active layers permitted per training step. If activating layer $l$ at step $t$ exceeds this threshold i.e., if $Q_{\text{prev}}(t) + \Lambda(l, t) > A_{\text{max}}$ with $\Lambda(l, t) = 1$ we redistribute the activation to an alternative step $t' \neq t$ satisfying:

$$Q_{\text{prev}}(t') + 1 \leq A_{\text{max}}, \quad t' \in \{1, \ldots, T\}, \quad \Lambda(l, t') = 0. \tag{9}$$

Upon identifying $t'$, we update the activation schedule by setting $\Lambda(l, t') = 1$ and $\Lambda(l, t) = 0$. Empirical results indicate that only 0% to 7% of layers are active at any step, ensuring the availability of suitable $t'$ satisfying Eq. 9.

In summary, this approach dynamically balances competition activations across layers, substantially reducing computational overhead while maintaining training stability for CompeteSMoE. Notably, the value of $A_{\text{max}}$ depends on several factors such as model architecture, batch size, and available GPU memory, and may vary if the experiments are conducted in a different environment.

## C  ABLATION STUDY

We conducted an ablation study on a 5.1B parameter VLM, evaluating performance across various configurations. The best performance was observed with the large-scale model.

### C.1  EFFECT OF COMPONENT-WISE DESIGN ON MODEL PERFORMANCE.

Table 10: Comprehensive component ablation study of CompeteSMoE across nine benchmarks.

| Method | Scheduler | Competition Mechanism | Diversity Loss | Dense MoE | AVG Acc ↑ | AVG Rank ↓ |
|---|---|---|---|---|---|---|
| CompeteSMoE | ✓ | ✓ | ✓ | ✗ | **53.21** | **1.78** |
| (1) | ✓ | ✓ | ✗ | ✗ | 52.90 | 3.11 |
| (2) | ✓ | ✗ | ✓ | ✗ | 52.71 | 3.89 |
| (3) | ✓ | ✗ | ✗ | ✓ | 52.70 | 4.11 |
| (4) | ✓ | ✗ | ✓ | ✓ | 52.79 | 3.44 |
| SMoE | ✗ | ✗ | ✗ | ✗ | 52.47 | 4.67 |

As shown in Table 10, we conduct a component-wise ablation study of the proposed CompeteSMoE model across nine benchmark datasets. Both the Competition Mechanism (1) and Diversity Loss (2) independently yield improvements over the standard SMoE baseline. Specifically, disabling DL results in a 0.49% drop in average accuracy, while removing CM leads to a smaller degradation of 0.30%. These findings suggest that CM contributes more significantly to overall model performance than DL when assessed in isolation.

In addition, we introduce two extended variants, models (3) and (4), inspired by the DenseMoE design (Pan et al., 2024), which activate all experts to compute the output but only occasionally during training using the same scheduler configuration. Interestingly, CompeteSMoE still consistently

outperforms. While dense activation provides a modest improvement over vanilla SMoE, it remains inferior to CompeteSMoE. This indicates that the performance gain arises not from dense expert activation per se, but from the competitive dynamics introduced by CM.

Crucially, in DenseMoE like variants (3) and (4), all experts are activated during both the forward and backward passes, leading to significantly increased computational cost. In contrast, CompeteSMoE activates all experts only to compute affinity scores, and then selects only the top-K winning experts to contribute to the final output and receive gradients. This hybrid mechanism enables more effective routing supervision while maintaining the computational efficiency characteristic of sparse MoE models.

## C.2 JOINT OPTIMIZATION OF TASK LOSS AND COMPETITION POLICY

Table 11: Ablation study showing the impact of task optimization and competition policy matching on performance across 9 benchmarks.

| Model | Task Loss | Match Competition Policy | AVG Acc | AVG Rank |
|---|---|---|---|---|
| CompeteSMoE | ✓ | ✓ | **53.21** | **1.33** |
| CompeteSMoE – Competition Policy Only | ✗ | ✓ | 52.84 | 2.11 |
| SMoE | ✓ | ✗ | 52.47 | 2.56 |

Our analysis in Section3 established that the competition policy alone is theoretically sufficient to achieve faster convergence compared to vanilla SMoE. However, in practice, CompeteSMoE stacks multiple SMoE layers, each independently equipped with a competition mechanism. Such a deep architecture requires significantly more training samples for convergence, often exceeding the dataset sizes available and making training infeasible under our hardware constraints. Therefore, we jointly optimize for the task loss and match the competition policy to facilitate practical and efficient learning.

The two supervision signals play complementary roles. When competition is inactive, the task loss gradient updates the router by adjusting affinity scores for the selected experts only, since inactive experts receive no gradients. When competition is active, its gradient provides updates for all experts, including those not selected in the final prediction. Combining both objectives thus provides more robust supervision and accelerates learning.

To validate this intuition, we conducted an ablation study on 9 benchmarks, isolating the effect of each supervision signal. Results are shown in Table 11. Jointly optimizing both signals yields the best average accuracy and rank. Interestingly, training the router solely to match the competition policy without any task loss supervision already surpasses the standard SMoE. This demonstrates that competition driven learning alone is capable of discovering stronger routing policies, even though the competition policy is active in only 7% of training steps. Despite such sparse updates, the router still learns significantly better expert selection than SMoE, indicating that the competition policy acts as a strong inductive bias.

Finally, the key properties of competition remain central in CompeteSMoE. As shown in Figure 1, CompeteSMoE achieves both faster and stronger convergence: after 8 hours of training, it already outperforms baselines trained for up to 14 hours. Moreover, Section 5.2.2 a) demonstrates that when deviating from the learned router policy (replacing the Top-1 expert with the Top-$(K+1)$ expert), SMoE surprisingly improves, whereas CompeteSMoE degrades. This indicates that the routing policy learned by SMoE is suboptimal, while CompeteSMoE produces a stronger and more consistent routing strategy.

### C.3 Effectiveness of Activation Functions in the Competition Mechanism

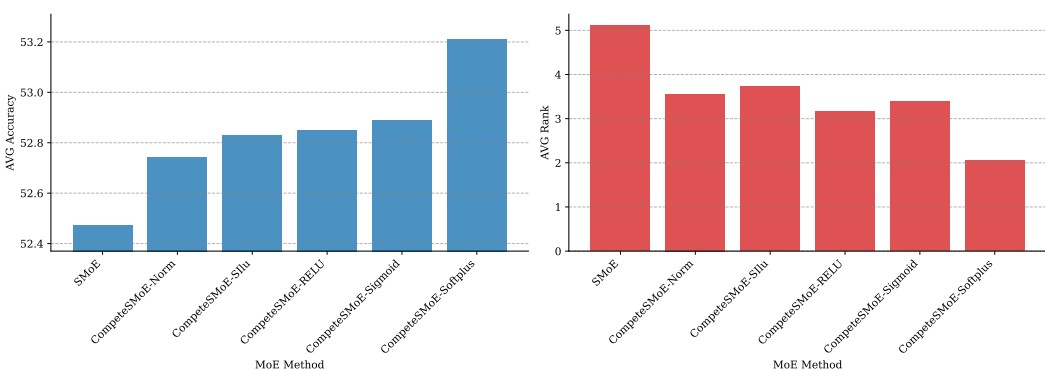

Figure 4: Performance comparison of different activation functions used within the Competition Mechanism across 9 benchmarks.

This section investigates how different activation functions influence the effectiveness of the Competition Mechanism. Specifically, we examine their role in computing expert affinity scores, originally defined in Eq. 2.2. To support a broader class of diversity-inducing functions, we generalize the affinity score formulation as follows:

$$s_i = \mathbb{E}[\kappa(g(\boldsymbol{x}, W_{e_i}))], \quad \forall i \in [N], \tag{10}$$

where $\kappa(\cdot)$ denotes the activation function applied to the expert response. This generalization allows the Competition Mechanism to flexibly incorporate a variety of activation profiles for expert selection. As shown in Figure 4, we compare several widely used activation functions within this framework, including Softplus, SiLU, Sigmoid, ReLU, and Softmax. Among them, Softplus consistently yields the highest average accuracy and ranking across tasks. We attribute this to its smooth and well behaved response curve, which softly suppresses negative values while preserving the magnitude of positive inputs. This behavior enables it to retain informative signals across the activation range, maintaining both representational richness and continuous gradient flow two properties critical for stable optimization. By contrast, Sigmoid compresses the entire input domain into the $[0, 1]$ interval, which can lead to vanishing gradients and loss of signal, especially for inputs with large magnitude. ReLU, although preserving positive values, entirely discards negative activations, potentially eliminating useful information. SiLU and Softmax lie between these extremes but still fall short of the balance offered by Softplus. We also explored an alternative formulation using the exponential function: $\mathbb{E}[e^{g(\boldsymbol{x}, W_{e_i})}]$. However, this variant led to uncontrolled growth in output magnitudes, causing numerical instability and NaN values during training. In contrast, Softplus provides a smooth approximation to the exponential function while mitigating such instability, making it a more robust choice for this setting.

In summary, activation functions that gently suppress negative activations while maintaining near-linear behavior on the positive side such as Softplus are better aligned with the needs of the Competition Mechanism. Their balanced characteristics lead to more stable expert affinity computation and improved end-task performance.

### C.4 Evaluation of Mean and Norm Strategies for Competition Mechanism

We conduct an empirical investigation to compare the mean-based strategy, as defined in Eq. 2.2, with a norm-based formulation. Specifically, we compute the affinity score of expert $i$ using the L2 norm of its output vector:

$$s_i = \|g(\boldsymbol{x}, W_{e_i})\|, \quad \forall i \in [N], \tag{11}$$

As shown in Figure 4, the CompeteSMoE-Norm variant using Equation 11 yields higher performance compared to the SMoE standard. However, when we switch to the CompeteSMoE-Softplus configuration that employs a mean based strategy, a substantial improvement is observed in both average accuracy and ranking. In conclusion, the mean-based strategy proves to be the most effective setting for expert output aggregation within the Competition Mechanism.

## C.5 EVALUATION OF DISTILLATION LOSS EFFECTIVENESS

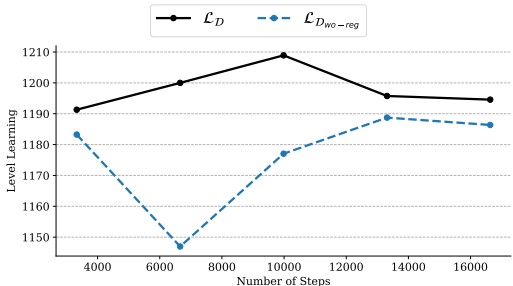

Figure 5: Learning performance of $\mathcal{L}_{\mathcal{D}}$ and $\mathcal{L}_{\mathcal{D}_{\text{wo-reg}}}$ measured by the Level Learning metric at every 20% of training steps on the MMBench-EN benchmark.

Table 12: Performance comparison between $\mathcal{L}_{\mathcal{D}}$ and $\mathcal{L}_{\mathcal{D}_{\text{wo-reg}}}$ across 9 benchmark datasets.

| Loss Function | Avg. Acc | Avg. Rank |
|---|---|---|
| $\mathcal{L}_{\mathcal{D}_{\text{wo-reg}}}$ | 52.92 | 1.78 |
| $\mathcal{L}_{\mathcal{D}}$ | **53.21** | **1.22** |

In Section 3, we established the theoretical foundation for the competition mechanism and demonstrated its empirical effectiveness in Table 1. A key challenge in optimizing the router network is accurately modeling the distribution of competitive routing decisions. We carefully investigated two objective functions: the distillation loss $\mathcal{L}_{\mathcal{D}}$ (see details in Eq. 1) and a variant distillation loss $\mathcal{L}_{\mathcal{D}_{\text{wo-reg}}}$ without the regularization term, which emphasizes penalizing experts who won the competition. We define $\mathcal{L}_{\mathcal{D}_{\text{wo-reg}}}$ as follows:

$$\mathcal{L}_{\mathcal{D}_{\text{wo-reg}}}(\boldsymbol{s}_{\mathcal{R}}, \boldsymbol{s}_{\mathcal{C}}) = \text{MSE}(\boldsymbol{s}_{\mathcal{R}}, \boldsymbol{s}_{\mathcal{C}}) \tag{12}$$

Figure 5 illustrates the progression of the Level Learning (LL) metric, which measures the average number of Top-$K$ experts selected by the router network that align with the Top-$K$ experts from the competition mechanism. A high LL value indicates that the router network effectively learns from the competition mechanism, whereas a low value suggests poor learning performance. Notably, $\mathcal{L}_{\mathcal{D}}$ consistently enables faster and more stable convergence compared to $\mathcal{L}_{\mathcal{D}_{\text{wo-reg}}}$. In particular, during the initial 60% of training (up to 9,600 steps), $\mathcal{L}_{\mathcal{D}}$ maintains a clear advantage, effectively mitigating the early performance drop observed with $\mathcal{L}_{\mathcal{D}_{\text{wo-reg}}}$. Moreover, $\mathcal{L}_{\mathcal{D}}$ achieves a peak LL score of 1210 by 12,000 steps, surpassing the $\mathcal{L}_{\mathcal{D}_{\text{wo-reg}}}$ peak of 1190, and exhibits more stable learning dynamics in later stages.

Additionally, quantitative results in Table 12 further confirm this trend, with $\mathcal{L}_{\mathcal{D}}$ yielding a higher average accuracy (53.21% vs. 52.92%) and a lower average rank (1.22 vs. 1.78) across nine benchmarks. These findings underscore the effectiveness of $\mathcal{L}_{\mathcal{D}}$ in guiding the router network to better approximate the competition mechanism. Furthermore, they suggest its potential as a preferred optimization objective in competitive MoE architectures.

## D EVOLUTIONARY ANALYSIS OF COMPETESMOE BEHAVIOR

### D.1 ACTIVATION MAGNITUDE DYNAMICS ACROSS LAYERS

In Figure 6, we analyze how expert output norms evolve across layers and training checkpoints for both SMoE and CompeteSMoE, following the norm-based methodology in MoEUT (Csordás et al., 2024). Although CompeteSMoE exhibits slightly larger activations, this increase is highly localized to the middle layers (approximately Layers 16–25). The early layers and the final layers

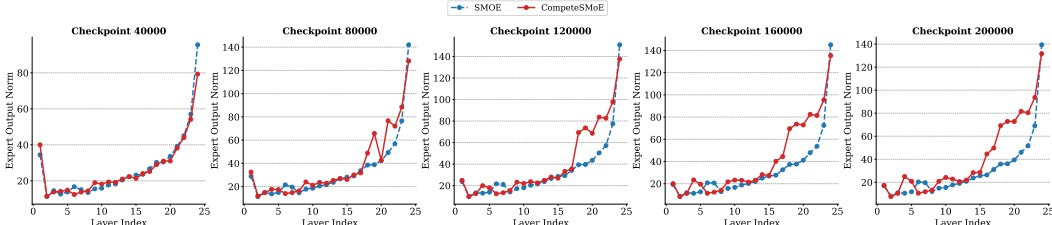

Figure 6: Layer-wise expert output norms across training progress on a 1B language pretraining model.

remain largely unchanged, and several of the deepest layers even show smaller activation magnitudes than SMoE. These trends indicate that the competition mechanism does not induce a monotonic or runaway growth pattern. Overall, the activation changes introduced by CompeteSMoE are moderate, layer-specific, and non-accumulative, suggesting that the method maintains stable activation scales across the network and avoids globally inflated magnitudes.

## D.2 EVOLUTION OF ROUTING SCORE DYNAMICS DURING TRAINING

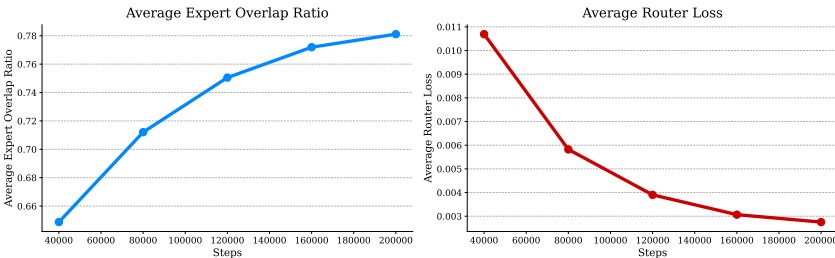

Figure 7: Evolution of router learning dynamics during training in the CompeteSMoE algorithm. The left plot presents the **expert overlap ratio** between the router's Top-$K$ selections and the competition mechanism, while the right plot shows the **router distillation loss**, which quantifies the discrepancy between router scores and competition scores.

In this section, we examine how efficiently the router network learns to align with the competition mechanism. In Figure 7, we report two key metrics: the **expert overlap ratio**, which measures the fraction of Top-$K$ experts selected by the router that coincide with those selected by the competition mechanism, and the **router loss**, computed as the MSE between router scores and competition scores across all experts. The expert overlap ratio steadily increases throughout training, indicating that the router progressively matches the competition mechanism in selecting experts. Similarly, the router loss consistently decreases to a small value, showing improved score alignment. Together, these trends confirm that the router network effectively learns and aligns its routing behavior with the competition policy over time.

## D.3 EVOLUTION ANALYSIS OF COMPETESMOE ACROSS MULTIPLE SPARSE-EXPERT CONFIGURATIONS

We evaluate the robustness of CompeteSMoE under varying levels of expert sparsity. As illustrated in Figure 8, we train a 0.3B-parameter language model pretrain on a 13B-token corpus and perform a three-level sparsity ablation with $K \in \{2, 4, 8\}$ under an $N = 24$-expert architecture. This setting enables us to isolate the effect of expert activation density on model performance. Across all sparsity configurations, CompeteSMoE consistently outperforms the standard SMoE baseline throughout the training trajectory. Importantly, the improvements are stable and do not depend on any particular selection of $K$. This pattern indicates that the performance gains arise from the competition-based routing mechanism itself, rather than from architectural idiosyncrasies or favorable hyperparameter choices.

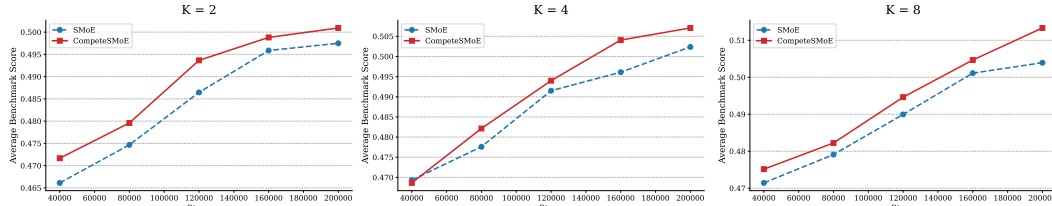

Figure 8: Comparison of average performance across 8 benchmarks between SMoE and CompeteS-MoE under expert configurations $K \in \{2, 4, 8\}$ and an $N = 24$-expert architecture, evaluated over training steps.

Overall, these results demonstrate that CompeteSMoE provides a reliable performance advantage across sparse-expert regimes. The method remains effective even when the degree of sparsity varies substantially, underscoring that its benefits stem from genuine algorithmic advances rather than from artifacts of model configuration, dataset size, or training dynamics.

## E    FURTHER ANALYSIS OF ROUTER BEHAVIOR

In this section, we further analyst about router behavior in SMoE and CompeteSMoE.

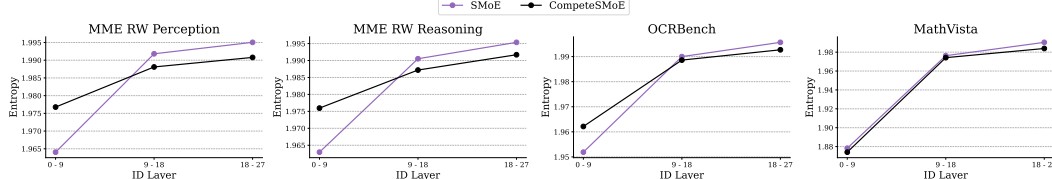

Figure 9: Entropy analysis of expert selection frequency across perception and reasoning tasks. Lower entropy indicates higher specialization in expert routing.

**(a) Experts distribution on Reasoning and Perception.** As illustrated in Figure 9, we analyze the entropy of expert distribution across layers for SMoE and CompeteSMoE algorithms, evaluated on three benchmarks: MME Real-World Perception and OCR Bench for perception capacity, and MME Real-World Reasoning and MathVista for reasoning capacity. On perception tasks, CompeteSMoE exhibits higher entropy in the early layers, indicating exploratory behavior, but significantly reduces entropy in the middle and final layers. In contrast, on MathVista a benchmark requiring higher-level reasoning CompeteSMoE maintains low entropy in the early and intermediate layers, approaching entropy levels similar to SMoE in the final layers. Both models demonstrate increasing entropy toward the final layers, suggesting more balanced expert allocation as the network deepens, consistent with typical Transformer-based architectures where later layers aggregate information from multiple upstream experts. Regarding the **representation collapse issue**, both SMoE and CompeteSMoE achieve a high degree of balance in expert distribution, with entropy scores exceeding 1.99 (compared to the maximum entropy of 2 for four experts).

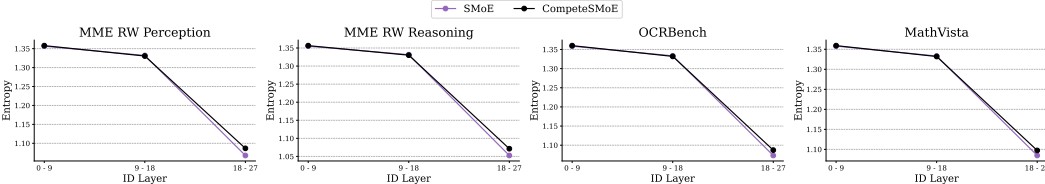

Figure 10: Layer-wise entropy of expert weight distributions for CompeteSMoE and SMoE across three tasks: Real-World Perception, Real-World Reasoning, and Mathematical Reasoning.

**(b) Effective Expert Aggregation via Weight Distribution.** As shown in Figure 10, we analyze the entropy of expert weight distributions across layers and tasks, which reflects how expert contributions are aggregated. Lower entropy typically suggests more confident expert selection. Both SMoE and CompeteSMoE exhibit decreasing entropy across layers, implying increased decisiveness in expert routing at deeper layers. While SMoE generally maintains lower entropy, especially on MathVista, it tends to concentrate weights heavily on a small subset of experts. In contrast, CompeteSMoE distributes weights more evenly among the selected experts. This balanced aggregation allows CompeteSMoE to better leverage complementary knowledge from multiple experts. Finally, we observe a slight difference between the two models, with both showing a trend toward more confident weight distributions in the final layers.

# F    ADDITIONAL EXPERIMENTAL RESULTS

Table 13: Performance comparison of SMoE strategies in the VIT pretraining setting, where only the MLP connectors are initialized as SMoE layers from scratch, using a ≈**4B-parameter model**. **Bold** values indicate the best result, while underlined values denote the second best. ↑ / ↓ indicate higher/lower is better.

| Method | AI2D | Text VQA | GQA | MM Bench | Hallusion | Math Vista | MMMU | MMStar | POPE | OCR | MME RWL | Avg. Acc↑ | Avg. Rank↓ |
|---|---|---|---|---|---|---|---|---|---|---|---|---|---|
| SMoE | 59.84 | 39.56 | 56.29 | 66.32 | 46.69 | 29.10 | 38.11 | 37.55 | 86.14 | **32.40** | 32.86 | 47.72 | 4.36 |
| XMoE | 61.59 | 39.40 | 55.92 | 65.98 | 45.32 | 29.80 | 39.00 | 39.82 | 86.58 | 32.10 | 31.21 | 47.88 | 3.86 |
| PCosine | 56.09 | 39.61 | 49.98 | 52.41 | 40.59 | 27.00 | 38.78 | 37.38 | 86.20 | 22.70 | 30.22 | 43.72 | 6.27 |
| $\sigma$-MoE | 61.59 | 39.18 | 56.60 | 65.98 | 44.80 | 30.00 | 39.00 | 37.89 | 86.28 | 32.10 | 31.21 | 47.69 | 4.14 |
| SharedE-V2 | 61.06 | 39.20 | 56.14 | 64.00 | 46.58 | 29.30 | 39.11 | 39.06 | **87.13** | 31.80 | 33.25 | 47.88 | 3.91 |
| SharedE-V3 | 61.37 | 39.60 | 56.33 | 66.58 | 45.01 | 29.30 | 37.56 | 39.12 | 86.97 | 32.20 | 33.13 | 47.92 | 3.41 |
| **CompeteSMoE** | **61.76** | **39.71** | 56.00 | **67.61** | **46.79** | 29.30 | **39.56** | **40.12** | 86.36 | 32.20 | **34.18** | **48.51** | **2.05** |

In Table 13, we report additional results using a vision-language model where only the MLP connectors are replaced with SMoE layers consisting of 8 experts, with 4 experts active per token. The vision encoder and language model (LLM) are kept dense and frozen during training, while the MLP connectors are unfrozen, following the setup described in Xu et al. (2024). These SMoE layers are trained from scratch using the same dataset and training configuration as the VIT stage, enabling a controlled analysis of sparse upcycling in isolation. Under this setup, CompeteSMoE consistently outperforms all baseline methods.

# G    ADDITIONAL $A_{max}$ SENSITIVITY RESULTS

To systematically assess the sensitivity of $A_{max}$ across a broader range of values, we conducted ablation studies by pretraining a smaller 0.3B-parameter language model using $A_{max} \in \{3, 6, 9\}$, with $N = 24$ and $K = 8$ experts, trained on 13B tokens. The results are summarized in Table 14.

Table 14: Comparison of CompeteSMoE performance across maximum activation thresholds ($A_{max} \in \{3, 6, 9\}$).

| Method | SMoE | CompeteSMoE | | |
|---|---|---|---|---|
| | | $A_{max} = 3$ | $A_{max} = 6$ | $A_{max} = 9$ |
| Avg. across 8 benchmarks | 50.39 | 50.95 | 51.33 | 51.11 |

As shown in Table 14, the performance of CompeteSMoE steadily increases as $A_{max}$ rises, peaking at $A_{max} = 6$ with a clear improvement from 3 to 6, then slightly decreases as $A_{max}$ increases from 6 to 9, although this fluctuation is negligible. This trend indicates that larger $A_{max}$ values generally result in better and more stable performance, and the differences between configurations gradually narrow, reflecting greater robustness. Notably, all CompeteSMoE settings with $A_{max} \in \{3, 6, 9\}$ outperform the SMoE baseline. Similarly, we also adopted $A_{max} = 6$ for the 1B language model pretraining experiment in Table 2, where it again achieved the best results. These findings further reinforce our recommendation regarding the choice of $A_{max}$.

# H    EXPERIMENTAL SETUP DETAILS

## H.1    VISION-LANGUAGE MODEL (VLM)

**Training Stages.**    We adopt a three-stage training pipeline inspired by prior works (Nguyen et al., 2024c; Li et al., 2024), designed to incrementally adapt and integrate the vision and language modalities for multimodal instruction tuning. Table 15 summarizes the training status of each model component throughout the stages.

Table 15: Component Training States at Each Stage

| Stage | LLM | MLP Connector | Vision Encoder |
|---|---|---|---|
| Pre-Training | Frozen | Trainable | Frozen |
| Pre-Finetuning | Trainable | Trainable | Trainable |
| Visual Instruction Tuning | Trainable | Trainable | Trainable |

- **(1) Pre-Training (PT):** In the first stage, only the MLP connector is trained, while the vision encoder and the language model (LLM) are kept frozen. This stage focuses on aligning visual features with the language embedding space, establishing a stable initialization without perturbing the frozen backbones.

- **(2) Pre-Finetuning (PFT):** All components including the vision encoder, MLP connector, and LLM are unfrozen and trained jointly using a dense architecture. This warm-up stage strengthens the cross-modal representation and stabilizes the model before introducing sparsity.

- **(3) Visual Instruction Tuning (VIT):** In the final stage, we apply Sparse Upcycling (Komatsuzaki et al., 2023) by replacing selected MLP layers in both the vision encoder and the connector with Top-$K$ sparsely gated MoE blocks. As this setup has become the standard practice in recent vision-language research (Nguyen et al., 2024c; Li et al., 2024; Shu et al., 2024; Lin et al., 2024), evaluating MoE algorithms under this setting offers more meaningful and practically relevant comparisons. Each expert is initialized from its corresponding pretrained MLP, while the Top-$K$ router is learned from scratch. To promote balanced expert utilization, we apply standard auxiliary objectives, including load-balancing loss and z-loss. All components remain fully trainable during this stage, and all compared methods are trained and evaluated under this unified VIT setup.

**Architecture.**    We adopt a modular design for the VIT-stage model, composed of a vision encoder, an MLP-based connector, and a pretrained language model backbone. The detailed architecture and MoE configuration are summarized in Table 16. During this stage, MoE layers are applied to the vision encoder and connector, while the language model remains dense.

Table 16: Architecture and parameter breakdown for each component in the 5.1B VIT-stage model, with MoE usage indicated.

| Component | Version / Variant | Parameters | SMoE |
|---|---|---|---|
| Vision Encoder | SigLIP-SO400M-Patch14-224 | 1.20B | ✓ |
| MLP Connector | - | 66M | ✓ |
| Language Model | Phi-3.5 Mini Instruct | 3.82B | ✗ |
| **Total** | – | **5.1B** | – |

**Hyperparameters.**    Table 17 lists the training hyperparameters for each stage. During VIT, SMoE layers are introduced by upcycling dense MLPs in the vision encoder and MLP connector. Each SMoE block contains $N_E = 4$ experts, with $K = 2$ experts selected per token. Given the sensitivity of large-scale model training to initialization and randomness, we ensure that all baseline models share identical starting conditions. Specifically, the router networks within MoE blocks are initialized

from scratch using Gaussian noise $\mathcal{N}(0, 0.02)$, following the scheme used in the official GPT-2 implementation.[1] To guarantee reproducibility, we fix the random seed to 42 across all experiments.

Table 17: Hyperparameter configurations for the three training stages of Phi-3.5 Mini.

| Hyperparameter | PT | PFT | VIT |
|---|---|---|---|
| Learning rate | 1e-3 | 2e-6 | 4e-6 |
| Schedule | Cosine | Cosine | Cosine |
| Batch size / GPU | 64 | 6 | 5 |
| GPUs | 4×H100 | 4×H100 | 4×H100 |
| ZeRO stage | ZeRO-2 | ZeRO-2 | ZeRO-3 |
| Optimizer | AdamW | AdamW | AdamW |
| MoE blocks | No | No | Yes |
| Balance loss coeff. | 0.0 | 0.0 | 0.01 |
| Z-loss coeff. | 0.0 | 0.0 | 0.001 |
| Max sequence length | 2048 | 2048 | 2048 |

**Dataset.** The full pipeline uses over 1B tokens, spanning: (1) LCS-558K (Liu et al., 2023a) (pretraining), (2) ALLaVA-Caption (Chen et al., 2024a) 708K (pre-finetuning), and (3) LLaVA-665K (instruction tuning) (Liu et al., 2024a), consistent with LibMoE (Nguyen et al., 2024c) and CuMo (Li et al., 2024).

## H.2 LANGUAGE MODEL PRETRAINING

**Architecture.** We pretrain a 1B-parameter and language model following the *original Transformer architecture*, where SMoE layers are integrated exclusively within the MLP blocks. Each SMoE layer consists of $N_E=24$ experts, with $K=8$ experts activated per token. Our architectural design is inspired by prior configurations such as SmolLM2-1.7B (Allal et al., 2025) and MoEUT (Csordás et al., 2024). The model specification is summarized in Table 18.

Table 18: Architecture configuration of the pretrained 1B language model.

| #Params | $n_{\text{layers}}$ | $d_{\text{model}}$ | $d_{\text{expert}}$ | $H$ | $d_{\text{head}}$ | $N_E$ | $K$ |
|---|---|---|---|---|---|---|---|
| 1B | 24 | 1024 | 512 | 32 | 128 | 24 | 8 |

**Hyperparameters.** We adopt the training configuration proposed in MoEUT Csordás et al. (2024), as detailed in Table 19. All model weights are initialized following the MoEUT initialization scheme, and a fixed random seed of 42 is set for reproducibility.

Table 19: Pretraining hyperparameters for MoE language model.

| Learning Rate | Schedule | Batch size | GPUs | Optimizer | Balance Coeff. | $N_{\text{warmup}}$ |
|---|---|---|---|---|---|---|
| 0.00025 | Cosine | 64 | 4×H100 | AdamW | 0.01 | 4000 |

**Dataset.** We train our models on 13B tokens sampled from the SlimPajama corpus (Soboleva et al., 2023). Training is conducted for 200K steps under this data regime.

---

[1] https://github.com/openai/gpt-2

### H.3 CompeteSMoE Configuration

**Hyperparameters.** Table 20 provides the key hyperparameters for training CompeteSMoE on the 5.1B VLM model. We warm up the MoE layers for 5% of total steps before enabling competition. The parameter $A_{\max}$ limits the number of concurrently active competition layers and is tuned for training stability. All runs use the same seed for fair comparison. Additional ablations on $\omega$ and $\alpha$ are included in Appendix C.

Table 20: CompeteSMoE hyperparameters on the 5.1B-parameter model.

| Warm-up | $\omega$ | $\gamma$ | $\alpha$ | $\beta$ | $A_{\max}$ |
|---------|----------|----------|----------|---------|------------|
| 0.05 | 0.07 | 0.01 | 0.1 | 0.005 | 9 |

## I Training Curves on Vision-Language Benchmarks

In Figure 11, we include additional training performance curves for 9 benchmarks, supplementing the results presented in Figure 1.

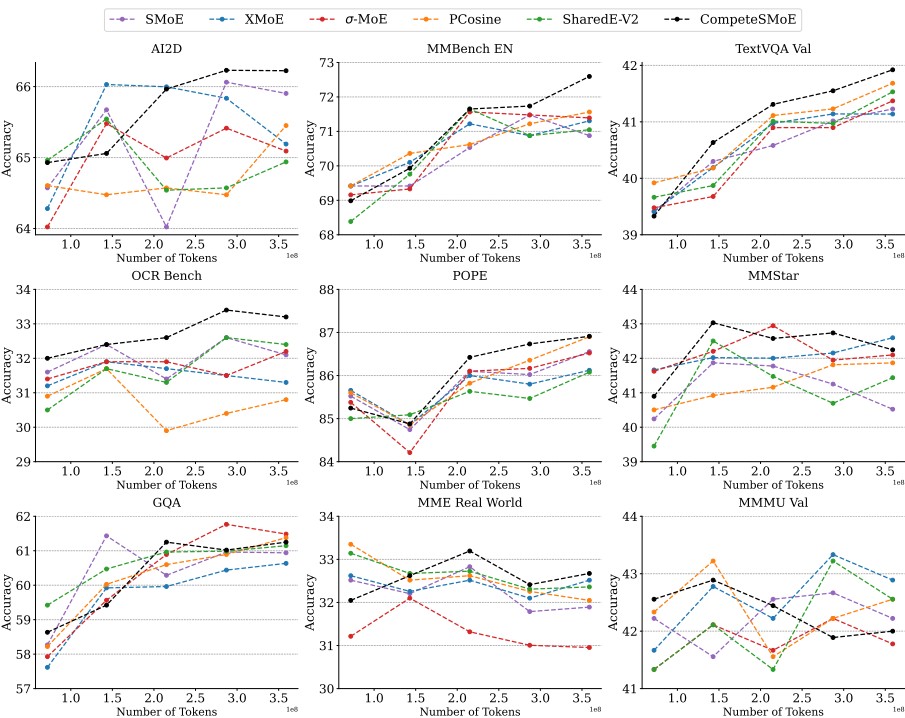

Figure 11: Training curves of CompeteSMoE compared to five advanced MoE algorithms on vision-language benchmarks.

## J Additional Theoretical Results

In this appendix, we analyze the convergence behavior of Gaussian mixture of linear experts equipped with the competition mechanism. In particular, we consider experts of the linear form $g(X, (a, b)) := a^\top X + b$, where $a \in \mathbb{R}^d$ and $b \in \mathbb{R}$. Then, the conditional density function $p_{G_*}(Y|X)$ in equation equation (3) becomes

$$p_{G_*}(Y|X) := \sum_{i=1}^{N^*} \frac{\exp(\log(1 + \exp((a_i^*)^\top X + b_i^*)))}{\sum_{j=1}^{N^*} \exp(\log(1 + \exp((a_j^*)^\top X + b_j^*)))} \cdot f(Y|(a_i^*)^\top X + b_i^*, \nu_i^*). \quad (13)$$

Our ultimate goal is to compare the sample efficiency of this model to that without the competition mechanism (Nguyen et al., 2023a) in terms of expert estimation. For that purpose, we use a Voronoi loss tailored to the setting of linear experts, which is given by

$$
\mathcal{L}_2(G, G_*) := \sum_{j=1}^{N^*} \Big| \sum_{i \in \mathcal{C}_j} \exp(c_i) - \exp(c_j^*) \Big|
$$
$$
+ \sum_{j \in [N^*]:|\mathcal{C}_j|=1} \sum_{i \in \mathcal{C}_j} \exp(c_i) \Big[ \|a_i - a_j^*\| + |b_i - b_j^*| + |\nu_i - \nu_j^*| \Big]
$$
$$
+ \sum_{j \in [N^*]:|\mathcal{C}_j|=1} \sum_{i \in \mathcal{C}_j} \exp(c_i) \Big[ \|a_i - a_j^*\|^2 + |b_i - b_j^*|^2 + |\nu_i - \nu_j^*|^2 \Big]. \tag{14}
$$

Equipped with the above Voronoi loss, we establish the convergence rate of parameter and expert estimations in the Gaussian mixture of linear experts with the competition in Theorem J.1.

**Theorem J.1.** *The following lower bound holds for any mixing measure $G \in \mathcal{G}_N(\Theta)$:*

$$
\mathbb{E}_X[V(p_G(\cdot|X), p_{G_*}(\cdot|X))] \gtrsim \mathcal{L}_2(G, G_*). \tag{15}
$$

*This lower bound indicates that $\mathcal{L}_2(\widehat{G}_n, G_*) = \mathcal{O}_P(\sqrt{\log(n)/n})$.*

The proof of Theorem J.1 can be found in Appendix K.2. A few remarks regarding the results of this theorem are in order.

*(i) Parameter estimation rates.* The bound of the Voronoi loss $\mathcal{L}_2(\widehat{G}_n, G_*)$ in Theorem J.1 reveals that the estimation rates for exact-specified parameters $a_j^*, b_j^*, \nu_j^*$, i.e., for $j \in [N^*] : |\mathcal{C}_j| = 1$, are of parametric order $\widetilde{\mathcal{O}}_P(n^{-1/2})$, whereas those for their over-specified counterparts, i.e., for $j \in [N^*] : |\mathcal{C}_j| > 1$, are slightly slower, of order $\widetilde{\mathcal{O}}_P(n^{-1/4})$.

*(ii) Expert estimation rates.* Note that the input space is bounded, then we have

$$
\Big| (\widehat{a}_i^n)^\top X + \widehat{b}_i^n - (a_j^*)^\top X - b_j^* \Big| \lesssim \|\widehat{a}_i^n - a_j^*\| + |\widehat{b}_i^n - b_j^*|,
$$

for almost surely $X$. Consequently, the estimation rates for exact-specified and over-specified experts $(a_j^*)^\top X + b_j^*$ are also of orders $\widetilde{\mathcal{O}}_P(n^{-1/2})$ and $\widetilde{\mathcal{O}}_P(n^{-1/4})$, respectively.

*(iii) Sample efficiency of the competition mechanism.* Thus, we need polynomially many data points $\mathcal{O}(\epsilon^{-4})$ to estimate these linear experts with a given error $\epsilon > 0$. By contrast, when not using the competition mechanism Nguyen et al. (2023a), the linear expert estimation rates are substantially slowed down since they hinge on the solvability of some complex system of polynomial equations and are decelerated as the number of fitted experts grows. For example, if a linear expert $(a_j^*)^\top X + b_j^*$ is fitted by two experts (or three experts), that is, $|\mathcal{C}_j| = 2$ (or $|\mathcal{C}_j| = 3$), then the rate for estimating this linear expert is of order $\widetilde{\mathcal{O}}_P(n^{-1/8})$ (or $\widetilde{\mathcal{O}}_P(n^{-1/12})$). Therefore, we need $\mathcal{O}(\epsilon^{-8})$ (or $\mathcal{O}(\epsilon^{-12})$), to estimate this expert. For that reason, we claim that the Gaussian MoE becomes more sample-efficient when equipped with the competition mechanism.

# K  PROOF OF THEORETICAL RESULTS

## K.1  PROOF OF THEOREM 3.2

In this proof, we aim to demonstrate that the following lower bound holds for any $G \in \mathcal{G}_N(\Theta)$:

$$
\mathbb{E}_X[V(p_G(\cdot|X), p_{G_*}(\cdot|X))] \gtrsim \mathcal{L}_1(G, G_*). \tag{16}
$$

For that purpose, we first establish the local part of the above bound, that is,

$$
\lim_{\varepsilon \to 0} \inf_{G \in \mathcal{G}_N(\Theta):\mathcal{L}_1(G,G_*) \le \varepsilon} \frac{\mathbb{E}_X[V(p_G(\cdot|X), p_{G_*}(\cdot|X))]}{\mathcal{L}_1(G, G_*)} > 0. \tag{17}
$$

This local part implies that there exists a positive constant $\varepsilon'$ that satisfies

$$
\inf_{G \in \mathcal{G}_N(\Theta):\mathcal{L}_1(G,G_*) \le \varepsilon'} \frac{\mathbb{E}_X[V(p_G(\cdot|X), p_{G_*}(\cdot|X))]}{\mathcal{L}_1(G, G_*)} > 0.
$$

Then, it is sufficient to derive the following global part of the bound in equation (16):

$$\inf_{G \in \mathcal{G}_N(\Theta): \mathcal{L}_1(G, G_*) > \varepsilon'} \frac{\mathbb{E}_X[V(p_G(\cdot|X), p_{G_*}(\cdot|X))]}{\mathcal{L}_1(G, G_*)} > 0. \tag{18}$$

**Local part:** In this part, we will establish the local part in equation equation (17) using the proof by contradiction method.

Suppose that the local part is not true, then we can find a sequence of mixing measures $(G_n)$ given by $G_n := \sum_{i=1}^N \exp(c_i^n) \delta_{(W_{e_i}^n, \nu_i^n)} \in \mathcal{G}_N(\Theta)$ such that $\mathcal{L}_1(G_n, G_*) \to 0$ and

$$\mathbb{E}_X[V(p_{G_n}(\cdot|X), p_{G_*}(\cdot|X))]/\mathcal{L}_1(G_n, G_*) \to 0,$$

as $n \to \infty$. As we use asymptotic arguments in this proof, we may assume without loss of generality (WLOG) that the Voronoi cells $\mathcal{C}_j^n := \mathcal{C}_j(G_n)$ is independent of the sample size $n$. Then, the Voronoi loss of interest turns into

$$\mathcal{L}_1(G_n, G_*) := \sum_{j=1}^{N^*} \Big| \sum_{i \in \mathcal{C}_j} \exp(c_i^n) - \exp(c_j^*) \Big| + \sum_{j \in [N^*]:|\mathcal{C}_j|=1} \sum_{i \in \mathcal{C}_j} \exp(c_i^n) \Big[ \|W_{e_i}^n - W_{e_j}^*\| + |\nu_i^n - \nu_j^*| \Big]$$

$$+ \sum_{j \in [N^*]:|\mathcal{C}_j|=1} \sum_{i \in \mathcal{C}_j} \exp(c_i^n) \Big[ \|W_{e_i}^n - W_{e_j}^*\|^2 + |\nu_i^n - \nu_j^*|^2 \Big]. \tag{19}$$

Since $\mathcal{L}_1(G_n, G_*) \to 0$ as $n \to \infty$, we have $(W_{e_i}^n, \nu_i^n) \to (W_{e_j}^*, \nu_j^*)$ for all $j \in [N^*]$ and $i \in \mathcal{C}_j$.

Subsequently, we divide the rest of this proof into three main steps.

**Step 1: Taylor expansion.** In this step, we aim to decompose the term $T_n(Y|X) := \Big[ \sum_{j=1}^{N^*} \exp(\log(1 + \exp(g(x, W_{e_j}^*)))) \Big] \cdot [p_{G_n}(Y|X) - p_{G_*}(Y|X)]$ can be decomposed as

$$T_n(Y|X) = \sum_{j=1}^{N^*} \sum_{i \in \mathcal{C}_j} \exp(c_i^n) \Big[ \exp(\log(1 + \exp(g(X, W_{e_i}^n)))) f(Y|g(X, W_{e_i}^n), \nu_i^n)$$

$$- \exp(\log(1 + \exp(g(X, W_{e_j}^*)))) f(Y|g(X, W_{e_j}^*), \nu_j^*) \Big]$$

$$- \sum_{j=1}^{N^*} \sum_{i \in \mathcal{C}_j} \exp(c_i^n) \Big[ \exp(\log(1 + \exp(g(X, W_{e_i}^n)))) - \exp(\log(1 + \exp(g(X, W_{e_j}^*)))) \Big] p_{G_n}(Y|X)$$

$$+ \sum_{j=1}^{N^*} \Big[ \sum_{i \in \mathcal{C}_j} \exp(c_i^n) - \exp(c_j^*) \Big] \cdot \exp(\log(1 + \exp(g(X, W_{e_j}^*)))) [f(Y|g(X, W_{e_j}^*), \nu_j^*) - p_{G_n}(Y|X)]$$

$$:= T_{n,1}(Y|X) - T_{n,2}(Y|X) + T_{n,3}(Y|X).$$

Next, we continue to decompose the term $T_{n,1}(Y|X)$ as

$$T_{n,1}(Y|X) = \sum_{j \in [N^*]:|\mathcal{C}_j|=1} \sum_{i \in \mathcal{C}_j} \exp(c_i^n) \Big[ \exp(\log(1 + \exp(g(x, W_{e_i}^n)))) f(Y|g(X, W_{e_i}^n), \nu_i^n)$$

$$- \exp(\log(1 + \exp(g(x, W_{e_j}^*)))) f(Y|g(X, W_{e_j}^*), \nu_j^*) \Big]$$

$$+ \sum_{j \in [N^*]:|\mathcal{C}_j|>1} \sum_{i \in \mathcal{C}_j} \exp(c_i^n) \Big[ \exp(\log(1 + \exp(g(x, W_{e_i}^n)))) f(Y|g(X, W_{e_i}^n), \nu_i^n)$$

$$- \exp(\log(1 + \exp(g(x, W_{e_j}^*)))) f(Y|g(X, W_{e_j}^*), \nu_j^*) \Big]$$

$$:= T_{n,1,1}(Y|X) + T_{n,1,2}(Y|X).$$

Let us denote $F_\rho(Y|X; W_e, \nu) := \exp(\log(1 + \exp(g(X, W_e)))) \frac{\partial^\rho f}{\partial g^\rho}(Y|g(X, W_e), \nu)$. By applying the first-order Taylor expansion to the function $F_0(Y|X; W_e, \nu)$ around the point $(W_{e_j}^*, \nu_j^*)$, we rewrite the term $T_{n,1,1}(Y|X)$ as

$$T_{n,1,1}(Y|X) = \sum_{j \in [N^*]:|\mathcal{C}_j|=1} \sum_{\rho=0}^{2} T_{n,1,1,\rho}^{(j)}(X) F_\rho(Y; X, W_{e_j}^*, \nu_j^*) + R_{n,1,1}(Y|X),$$

where $R_{n,1,1}(Y|X)$ is the Taylor remainder such that $R_{n,1,1}(Y|X)/\mathcal{L}_1(G_n, G_*) \to 0$ as $n \to \infty$, and

$$T_{n,1,1,0}^{(j)}(X) := \sum_{i \in \mathcal{C}_j} \exp(c_i^n) \sum_{u=1}^{d_2} (\Delta W_{e_{ij}}^n)^{(u)} \frac{\partial g}{\partial W_e^{(u)}}(X, W_{e_j}^*) \cdot \frac{1}{1 + \exp(-g(X, W_{e_j}^*))},$$

$$T_{n,1,1,1}^{(j)}(X) := \sum_{i \in \mathcal{C}_j} \exp(c_i^n) \sum_{u=1}^{d_2} (\Delta W_{e_{ij}}^n)^{(u)} \frac{\partial g}{\partial W_e^{(u)}}(X, W_{e_j}^*),$$

$$T_{n,1,1,2}^{(j)}(X) := \sum_{i \in \mathcal{C}_j} \frac{1}{2} \exp(c_i^n)(\Delta \nu_{ij}^n),$$

in which $\Delta W_{e_{ij}}^n := W_{e_i}^n - W_{e_j}^*$ and $\Delta \nu_{ij}^n := \nu_i^n - \nu_j^*$.

Meanwhile, by means of the second-order Taylor expansion, the term $T_{n,1,2}(Y|X)$ can be represented as

$$T_{n,1,2}(Y|X) = \sum_{j \in [N^*]:|\mathcal{C}_j|>1} \sum_{\rho=0}^{4} T_{n,1,2,\rho}^{(j)}(X) F_\rho(Y; X, W_{e_j}^*, \nu_j^*) + R_{n,1,2}(Y|X),$$

where $R_{n,1,2}(Y|X)$ is the Taylor remainder such that $R_{n,1,2}(Y|X)/\mathcal{L}_1(G_n, G_*) \to 0$ as $n \to \infty$, and

$$T_{n,1,2,0}^{(j)}(X) := \sum_{i \in \mathcal{C}_j} \exp(c_i^n) \left[ \sum_{u=1}^{d_2} (\Delta W_{e_{ij}}^n)^{(u)} \cdot \frac{\frac{\partial g}{\partial W_e^{(u)}}(X, W_{e_j}^*)}{1 + \exp(-g(X, W_{e_j}^*))} \right.$$

$$\left. + \sum_{u,v=1}^{d_2} \frac{(\Delta W_{e_{ij}}^n)^{(u)}(\Delta W_{e_{ij}}^n)^{(v)}}{1 + 1_{\{u=v\}}} \cdot \frac{\frac{\partial^2 g}{\partial W_e^{(u)} \partial W_e^{(v)}}(X, W_{e_j}^*) + \frac{\partial g}{\partial W_e^{(u)}}(X, W_{e_j}^*)\frac{\partial g}{\partial W_e^{(v)}}(X, W_{e_j}^*)}{1 + \exp(-g(X, W_{e_j}^*))} \right],$$

$$T_{n,1,2,1}^{(j)}(X) := \sum_{i \in \mathcal{C}_j} \exp(c_i^n) \left[ \sum_{u=1}^{d_2} (\Delta W_{e_{ij}}^n)^{(u)} \frac{\partial g}{\partial W_e^{(u)}}(X, W_{e_j}^*) \right.$$

$$\left. + \sum_{u,v=1}^{d_2} \frac{(\Delta W_{e_{ij}}^n)^{(u)}(\Delta W_{e_{ij}}^n)^{(v)}}{1 + 1_{\{u=v\}}} \left( \frac{2\frac{\partial g}{\partial W_e^{(u)}}(X, W_{e_j}^*)\frac{\partial g}{\partial W_e^{(v)}}(X, W_{e_j}^*)}{1 + \exp(-g(X, W_{e_j}^*))} + \frac{\partial^2 g}{\partial W_e^{(u)} \partial W_e^{(v)}}(X, W_{e_j}^*) \right) \right],$$

$$T_{n,1,2,2}^{(j)}(X) := \sum_{i \in \mathcal{C}_j} \exp(c_i^n) \left[ \frac{1}{2}(\Delta \nu_{ij}^n) + \sum_{u,v=1}^{d_2} \frac{(\Delta W_{e_{ij}}^n)^{(u)}(\Delta W_{e_{ij}}^n)^{(v)}}{1 + 1_{\{u=v\}}} \cdot \frac{\partial g}{\partial W_e^{(u)}}(X, W_{e_j}^*)\frac{\partial g}{\partial W_e^{(v)}}(X, W_{e_j}^*) \right.$$

$$\left. + \sum_{u=1}^{d_2} (\Delta W_{e_{ij}}^n)^{(u)}(\Delta \nu_{ij}^n) \cdot \frac{1}{2}\frac{\frac{\partial g}{\partial W_e^{(u)}}(X, W_{e_j}^*)}{1 + \exp(-g(X, W_{e_j}^*))} \right],$$

$$T_{n,1,2,3}^{(j)}(X) := \sum_{i \in \mathcal{C}_j} \exp(c_i^n) \sum_{u=1}^{d_2} \frac{1}{2}(\Delta W_{e_{ij}}^n)^{(u)}(\Delta \nu_{ij}^n)\frac{\partial g}{\partial W_e^{(u)}}(X, W_{e_j}^*),$$

$$T_{n,1,2,4}^{(j)}(X) := \sum_{i \in \mathcal{C}_j} \exp(c_i^n) \cdot \frac{1}{4}(\Delta \nu_{ij}^n)^2.$$

Next, we decompose the term $T_{n,2}(Y|X)$ as

$$T_{n,2}(Y|X)$$

$$:= \sum_{j \in [N^*]:|\mathcal{C}_j|=1} \sum_{i \in \mathcal{C}_j} \exp(c_i^n) \Big[ \exp(\log(1 + \exp(g(X, W_{e_i}^n)))) - \exp(\log(1 + \exp(g(X, W_{e_j}^*)))) \Big] p_{G_n}(Y|X)$$

$$+ \sum_{j \in [N^*]:|\mathcal{C}_j|>1} \sum_{i \in \mathcal{C}_j} \exp(c_i^n) \Big[ \exp(\log(1 + \exp(g(X, W_{e_i}^n)))) - \exp(\log(1 + \exp(g(X, W_{e_j}^*)))) \Big] p_{G_n}(Y|X)$$

$$:= T_{n,2,1}(Y|X) + T_{n,2,2}(Y|X).$$

Note that we can rewrite the term $T_{n,1,2}(Y|X)$ using the first-order Taylor expansion to the function $\exp(\log(1 + \exp(g(W_{e_i}^n))))$ around the point $W_{e_j}^*$ as

$$T_{n,2,1}(Y|X) = \sum_{j \in [N^*]: |\mathcal{C}_j| = 1} \sum_{i \in \mathcal{C}_j} \exp(c_i^n) \sum_{u=1}^{d_2} (\Delta W_{e_{ij}}^n)^{(u)} \cdot \frac{\frac{\partial g}{\partial W_e^{(u)}}(X, W_{e_j}^*)}{1 + \exp(-g(X, W_{e_j}^*))} H_n(Y|X; W_{e_j}^*)$$

$$+ R_{n,2,1}(Y|X),$$

where we denote $H_n(Y|X; W_e) = \exp(\log(1 + \exp(g(X, W_e)))) p_{G_n}(Y|X)$ and $R_{n,2,1}(Y|X)$ is the Taylor remainder such that $R_{n,2,1}(Y|X)/\mathcal{L}_1(G_n, G_*) \to 0$ as $n \to \infty$.

On the other hand, by means of the second-order Taylor expansion, we have

$$T_{n,2,2}(Y|X) = \sum_{j \in [N^*]: |\mathcal{C}_j| > 1} \sum_{i \in \mathcal{C}_j} \exp(c_i^n) \left[ \sum_{u=1}^{d_2} (\Delta W_{e_{ij}}^n)^{(u)} \cdot \frac{\frac{\partial g}{\partial W_e^{(u)}}(X, W_{e_j}^*)}{1 + \exp(-g(X, W_{e_j}^*))} \right.$$

$$\left. + \sum_{u,v=1}^{d_2} \frac{(\Delta W_{e_{ij}}^n)^{(u)} (\Delta W_{e_{ij}}^n)^{(v)}}{1 + 1_{\{u=v\}}} \cdot \frac{\frac{\partial^2 g}{\partial W_e^{(u)} \partial W_e^{(v)}}(X, W_{e_j}^*) + \frac{\partial g}{\partial W_e^{(u)}}(X, W_{e_j}^*) \frac{\partial g}{\partial W_e^{(v)}}(X, W_{e_j}^*)}{1 + \exp(-g(X, W_{e_j}^*))} \right] H_n(Y|X; W_{e_j}^*)$$

$$+ R_{n,2,2}(Y|X),$$

where $R_{n,2,1}(Y|X)$ is the Taylor remainder such that $R_{n,2,2}(Y|X)/\mathcal{L}_1(G_n, G_*) \to 0$ as $n \to \infty$.

From the above equation, $[T_{n,1,1}(Y|X) - R_{n,1,1}(Y|X)]$, $[T_{n,1,2}(Y|X) - R_{n,1,2}(Y|X)]$, $[T_{n,2,1}(Y|X) - R_{n,2,1}(Y|X)]$, $[T_{n,2,2}(Y|X) - R_{n,2,2}(Y|X)]$ and $[T_{n,3}(Y|X)]$ can be seen as a combination of elements of the set $\mathcal{S} := \bigcup_{j=1}^{N} \bigcup_{\rho=0}^{5} \mathcal{S}_{\rho,j}$, where we define

$$\mathcal{S}_{0,j} := \left\{ \frac{\frac{\partial g}{\partial W_e^{(u)}}(X, W_{e_j}^*)}{1 + \exp(-g(X, W_{e_i}^*))} F_0(Y|X; W_{e_j}^*, \nu_j^*), \ \frac{\frac{\partial^2 g}{\partial W_e^{(u)} \partial W_e^{(v)}}(X, W_{e_j}^*)}{1 + \exp(-g(X, W_{e_j}^*))} F_0(Y|X; W_{e_j}^*, \nu_j^*), \right.$$

$$\left. \frac{\frac{\partial g}{\partial W_e^{(u)}}(X, W_{e_j}^*) \frac{\partial g}{\partial W_e^{(v)}}(X, W_{e_j}^*)}{1 + \exp(-g(X, W_{e_j}^*))} F_0(Y|X; W_{e_j}^*, \nu_j^*), \ F_0(Y|X; W_{e_j}^*, \nu_j^*) : 1 \le u, v \le d_2 \right\},$$

$$\mathcal{S}_{1,j} := \left\{ \frac{\frac{\partial g}{\partial W_e^{(u)}}(X, W_{e_j}^*)}{1 + \exp(-g(X, W_{e_i}^*))} F_1(Y|X; W_{e_j}^*, \nu_j^*), \ \frac{\frac{\partial g}{\partial W_e^{(u)}}(X, W_{e_j}^*) \frac{\partial g}{\partial W_e^{(v)}}(X, W_{e_j}^*)}{1 + \exp(-g(X, W_{e_j}^*))} F_1(Y|X; W_{e_j}^*, \nu_j^*) \right.$$

$$\left. \frac{\partial^2 g}{\partial W_e^{(u)} \partial W_e^{(v)}}(X, W_{e_j}^*) F_1(Y|X; W_{e_j}^*, \nu_j^*) : 1 \le u, v \le d_2 \right\},$$

$$\mathcal{S}_{2,j} := \left\{ F_2(Y|X; W_{e_j}^*, \nu_j^*), \ \frac{\frac{\partial g}{\partial W_e^{(u)}}(X, W_{e_j}^*)}{1 + \exp(-g(X, W_{e_i}^*))} F_2(Y|X; W_{e_j}^*, \nu_j^*), \right.$$

$$\left. \frac{\partial g}{\partial W_e^{(u)}}(X, W_{e_j}^*) \frac{\partial g}{\partial W_e^{(v)}}(X, W_{e_j}^*) F_2(Y|X; W_{e_j}^*, \nu_j^*) : 1 \le u, v \le d_2 \right\},$$

$$\mathcal{S}_{3,j} := \left\{ \frac{\partial g}{\partial W_e^{(u)}}(X, W_{e_i}^*) F_3(Y|X; W_{e_j}^*, \nu_j^*) : 1 \le u \le d_2 \right\},$$

$$\mathcal{S}_{4,j} := \left\{ F_4(Y|X; W_{e_j}^*, \nu_j^*) \right\},$$

$$\mathcal{S}_{5,j} := \left\{ \frac{\frac{\partial g}{\partial W_e^{(u)}}(X, W_{e_j}^*)}{1 + \exp(-g(X, W_{e_i}^*))} H_n(Y|X; W_{e_j}^*, \nu_j^*), \ \frac{\frac{\partial^2 g}{\partial W_e^{(u)} \partial W_e^{(v)}}(X, W_{e_j}^*)}{1 + \exp(-g(X, W_{e_j}^*))} H_n(Y|X; W_{e_j}^*, \nu_j^*), \right.$$

$$\left. \frac{\frac{\partial g}{\partial W_e^{(u)}}(X, W_{e_j}^*) \frac{\partial g}{\partial W_e^{(v)}}(X, W_{e_j}^*)}{1 + \exp(-g(X, W_{e_j}^*))} H_n(Y|X; W_{e_j}^*, \nu_j^*), \ H_n(Y|X; W_{e_j}^*, \nu_j^*) : 1 \le u, v \le d_2 \right\}.$$

**Step 2: Non-vanishing coefficients.** In this step, we will show that at least one among the coefficients in the representations of $[T_{n,1,1}(Y|X) - R_{n,1,1}(Y|X)]/\mathcal{L}_1(G_n, G_*)$, $[T_{n,1,2}(Y|X) - R_{n,1,2}(Y|X)]/\mathcal{L}_1(G_n, G_*)$, $[T_{n,2,1}(Y|X) - R_{n,2,1}(Y|X)]/\mathcal{L}_1(G_n, G_*)$, $[T_{n,2,2}(Y|X) - R_{n,2,2}(Y|X)]/\mathcal{L}_1(G_n, G_*)$ and $[T_{n,3}(Y|X)]/\mathcal{L}_1(G_n, G_*)$ does not approach zero when $n$ goes to infinity. Assume by contrary that all of them vanish as $n \to \infty$. Then, by considering the coefficients of the term

- $F_0(Y|X; W^*_{e_j}, \nu^*_j)$ for $j \in [N^*]$, we have

$$\frac{1}{\mathcal{L}_1(G_n, G_*)} \cdot \sum_{j=1}^{N^*} \Big| \sum_{i \in \mathcal{C}_j} \exp(c_i^n) - \exp(c_j^*) \Big| \to 0.$$

- $\frac{\frac{\partial g}{\partial w_e^{(u)}}(X, W^*_{e_j})}{1 + \exp(-g(X, W^*_{e_i}))} F_0(Y|X; W^*_{e_j}, \nu^*_j)$ for $j \in [N^*] : |\mathcal{C}_j| = 1$, we have

$$\frac{1}{\mathcal{L}_1(G_n, G_*)} \cdot \sum_{j \in [N^*]:|\mathcal{C}_j|=1} \sum_{i \in \mathcal{C}_j} \exp(c_i^n) \|\Delta W^n_{e_{ij}}\|_1 \to 0.$$

  Due to the equivalence between the $\ell_1$-norm and the $\ell_2$-norm, we obtain

$$\frac{1}{\mathcal{L}_1(G_n, G_*)} \cdot \sum_{j \in [N^*]:|\mathcal{C}_j|=1} \sum_{i \in \mathcal{C}_j} \exp(c_i^n) \|\Delta W^n_{e_{ij}}\| \to 0.$$

- $F_2(Y|X; W^*_{e_j}, \nu^*_j)$ for $j \in [N^*] : |\mathcal{C}_j| = 1$, we have

$$\frac{1}{\mathcal{L}_1(G_n, G_*)} \cdot \sum_{j \in [N^*]:|\mathcal{C}_j|=1} \sum_{i \in \mathcal{C}_j} \exp(c_i^n) |\Delta \nu^n_{ij}| \to 0.$$

- $\frac{\frac{\partial g}{\partial w_e^{(u)}}(X, W^*_{e_j}) \frac{\partial g}{\partial w_e^{(u)}}(X, W^*_{e_j})}{1 + \exp(-g(X, W^*_{e_j}))} F_0(Y|X; W^*_{e_j}, \nu^*_j)$ for $j \in [N^*] : |\mathcal{C}_j| > 1$, we have

$$\frac{1}{\mathcal{L}_1(G_n, G_*)} \cdot \sum_{j \in [N^*]:|\mathcal{C}_j|>1} \sum_{i \in \mathcal{C}_j} \exp(c_i^n) \|\Delta W^n_{e_{ij}}\|^2 \to 0.$$

- $F_4(Y|X; W^*_{e_j}, \nu^*_j)$ for $j \in [N^*] : |\mathcal{C}_j| > 1$, we have

$$\frac{1}{\mathcal{L}_1(G_n, G_*)} \cdot \sum_{j \in [N^*]:|\mathcal{C}_j|=1} \sum_{i \in \mathcal{C}_j} \exp(c_i^n) |\Delta \nu^n_{ij}|^2 \to 0.$$

By taking the sum of the above limits, we obtain $1 = \frac{\mathcal{L}_1(G_n, G_*)}{\mathcal{L}_1(G_n, G_*)} \to 0$ as $n \to \infty$, which is a contradiction. Thus, not all the coefficients in the representations of $[T_{n,1,1}(Y|X) - R_{n,1,1}(Y|X)]/\mathcal{L}_1(G_n, G_*)$, $[T_{n,1,2}(Y|X) - R_{n,1,2}(Y|X)]/\mathcal{L}_1(G_n, G_*)$, $[T_{n,2,1}(Y|X) - R_{n,2,1}(Y|X)]/\mathcal{L}_1(G_n, G_*)$, $[T_{n,2,2}(Y|X) - R_{n,2,2}(Y|X)]/\mathcal{L}_1(G_n, G_*)$ and $[T_{n,3}(Y|X)]/\mathcal{L}_1(G_n, G_*)$ converge to zero as $n \to \infty$.

**Stage 3 - Fatou's argument:** In this stage, we use the Fatou's lemma to show a contradiction to the result of Step 2. For that purpose, let us denote $m_n$ as the maximum of the absolute values of the coefficients in the representations of $[T_{n,1,1}(Y|X) - R_{n,1,1}(Y|X)]/\mathcal{L}_1(G_n, G_*)$, $[T_{n,1,2}(Y|X) - R_{n,1,2}(Y|X)]/\mathcal{L}_1(G_n, G_*)$, $[T_{n,2,1}(Y|X) - R_{n,2,1}(Y|X)]/\mathcal{L}_1(G_n, G_*)$, $[T_{n,2,2}(Y|X) - R_{n,2,2}(Y|X)]/\mathcal{L}_1(G_n, G_*)$ and $[T_{n,3}(Y|X)]/\mathcal{L}_1(G_n, G_*)$. It follows from the result of Step 2 that $1/m_n \not\to \infty$ as $n \to \infty$. In addition, we also denote

$$\frac{\sum_{i \in \mathcal{C}_j} \exp(c_i^n)(\Delta W^n_{e_{ij}})^{(u)}}{m_n \mathcal{L}_1(G_n, G_*)} \to \alpha^{(u)}_{1,j}, \qquad \frac{\sum_{i \in \mathcal{C}_j} \exp(c_i^n)(\Delta \nu^n_{ij})}{m_n \mathcal{L}_1(G_n, G_*)} \to \beta_{1,j},$$

$$\frac{\sum_{i \in \mathcal{C}_j} \exp(c_i^n)(\Delta W^n_{e_{ij}})^{(u)}(\Delta W^n_{e_{ij}})^{(v)}}{m_n \mathcal{L}_1(G_n, G_*)} \to \alpha^{(uv)}_{2,j}, \qquad \frac{\sum_{i \in \mathcal{C}_j} \exp(c_i^n)(\Delta \nu^n_{ij})^2}{m_n \mathcal{L}_1(G_n, G_*)} \to \beta_{2,j},$$

$$\frac{\sum_{i \in \mathcal{C}_j} \exp(c_i^n)(\Delta W^n_{e_{ij}})^{(u)}(\Delta \nu^n_{ij})}{m_n \mathcal{L}_1(G_n, G_*)} \to \gamma^{(u)}_j, \qquad \frac{\sum_{i \in \mathcal{C}_j} \exp(c_i^n) - \exp(c_j^*)}{m_n \mathcal{L}_1(G_n, G_*)} \to \xi_j,$$

as $n \to \infty$ for any $j \in [N^*]$ and $u, v \in [d_2]$ with a note that at least one among $\alpha_{1,j}^{(u)}, \beta_{1,j}, \alpha_{2,j}^{(uv)}, \beta_{2,j}$,
$\gamma_j^{(u)}$ and $\xi_j$ is non-zero.

By applying the Fatou's lemma, we have

$$0 = \lim_{n \to \infty} \frac{\mathbb{E}_X[V(p_G(\cdot|X), p_{G_*}(\cdot|X))]}{m_n \mathcal{L}_1(G_n, G_*)} = \frac{1}{2} \int \liminf_{n \to \infty} \frac{|p_{G_n}(Y|X) - p_{G_*}(Y|X)|}{m_n \mathcal{L}_1(G_n, G_*)} \mathrm{d}(X, Y),$$

which implies that $[p_{G_n}(Y|X) - p_{G_*}(Y|X)]/[m_n \mathcal{L}_1(G_n, G_*)] \to 0$ as $n \to \infty$ for almost surely $(X, Y)$. Since the term $\sum_{j=1}^{N^*} \exp(\log(1 + \exp(g(x, W_{e_i}^*))))$ is bounded, we also have $T_n(Y|X)/[m_n \mathcal{L}_1(G_n, G_*)] \to 0$ as $n \to \infty$. Then, it follows that

$$0 = \lim_{n \to \infty} \frac{T_{n,1,1}(Y|X) + T_{n,1,2}(Y|X)}{m_n \mathcal{L}_1(G_n, G_*)} - \lim_{n \to \infty} \frac{T_{n,2,1}(Y|X) + T_{n,2,2}(Y|X)}{m_n \mathcal{L}_1(G_n, G_*)} + \lim_{n \to \infty} \frac{T_{n,3}(Y|X)}{m_n \mathcal{L}_1(G_n, G_*)}, \tag{20}$$

for almost surely $(X, Y) \in \mathcal{X} \times \mathcal{Y}$, where we have

$$\lim_{n \to \infty} \frac{T_{n,1,1}(Y|X)}{m_n \mathcal{L}_1(G_n, G_*)} := \sum_{j \in [N^*]: |\mathcal{C}_j| = 1} \left[ \sum_{u=1}^{d_2} \alpha_{1,j}^{(u)} \frac{\frac{\partial g}{\partial W_e^{(u)}}(X, W_{e_j}^*)}{1 + \exp(-g(X, W_{e_j}^*))} F_{0,j}(Y|X) \right.$$

$$\left. + \sum_{u=1}^{d_2} \alpha_{1,j}^{(u)} \frac{\partial g}{\partial W_e^{(u)}}(X, W_{e_j}^*) F_{1,j}(Y|X) + \frac{1}{2} \beta_{1,j} F_{2,j}(Y|X) \right],$$

$$\lim_{n \to \infty} \frac{T_{n,1,2}(Y|X)}{m_n \mathcal{L}_1(G_n, G_*)} := \sum_{j \in [N^*]: |\mathcal{C}_j| > 1} \left[ \left( \sum_{u=1}^{d_2} \alpha_{1,j}^{(u)} \frac{\frac{\partial g}{\partial W_e^{(u)}}(X, W_{e_j}^*)}{1 + \exp(-g(X, W_{e_j}^*))} \right. \right.$$

$$+ \sum_{u,v=1}^{d_2} \frac{\alpha_{2,j}^{(uv)}}{1 + 1_{\{u=v\}}} \cdot \frac{\frac{\partial^2 g}{\partial W_e^{(u)} \partial W_e^{(v)}}(X, W_{e_j}^*) + \frac{\partial g}{\partial W_e^{(u)}}(X, W_{e_j}^*) \frac{\partial g}{\partial W_e^{(v)}}(X, W_{e_j}^*)}{1 + \exp(-g(X, W_{e_j}^*))} \right) F_{0,j}(Y|X)$$

$$+ \left( \sum_{u=1}^{d_2} \alpha_{1,j}^{(u)} \frac{\partial g}{\partial W_e^{(u)}}(X, W_{e_j}^*) + \sum_{u,v=1}^{d_2} \frac{\alpha_{2,j}^{(uv)}}{1 + 1_{\{u=v\}}} \left( \frac{2 \frac{\partial g}{\partial W_e^{(u)}}(X, W_{e_j}^*) \frac{\partial g}{\partial W_e^{(v)}}(X, W_{e_j}^*)}{1 + \exp(-g(X, W_{e_j}^*))} \right. \right.$$

$$\left. \left. + \frac{\partial^2 g}{\partial W_e^{(u)} \partial W_e^{(v)}}(X, W_{e_j}^*) \right) \right) F_{1,j}(Y|X) + \left( \frac{1}{2} \beta_{1,j} + \sum_{u=1}^{d_2} \gamma_j^{(u)} \cdot \frac{1}{2} \frac{\frac{\partial g}{\partial W_e^{(u)}}(X, W_{e_j}^*)}{1 + \exp(-g(X, W_{e_j}^*))} \right.$$

$$\left. + \sum_{u,v=1}^{d_2} \frac{\alpha_{2,j}^{(uv)}}{1 + 1_{\{u=v\}}} \cdot \frac{\partial g}{\partial W_e^{(u)}}(X, W_{e_j}^*) \frac{\partial g}{\partial W_e^{(v)}}(X, W_{e_j}^*) \right) F_{2,j}(Y|X)$$

$$+ \sum_{u=1}^{d_2} \frac{1}{2} \gamma_j^{(u)} \frac{\partial g}{\partial W_e^{(u)}}(X, W_{e_j}^*) F_{3,j}(Y|X) + \frac{1}{4} \beta_{2,j} F_{4,j}(Y|X) \right],$$

and

$$\lim_{n \to \infty} \frac{T_{n,2,1}(Y|X)}{m_n \mathcal{L}_1(G_n, G_*)} := \sum_{j \in [N^*]: |\mathcal{C}_j| = 1} \sum_{u=1}^{d_2} \alpha_{1,j}^{(u)} \cdot \frac{\frac{\partial g}{\partial W_e^{(u)}}(X, W_{e_j}^*)}{1 + \exp(-g(X, W_{e_j}^*))} H_j(Y|X),$$

$$\lim_{n \to \infty} \frac{T_{n,2,2}(Y|X)}{m_n \mathcal{L}_1(G_n, G_*)} := \sum_{j \in [N^*]: |\mathcal{C}_j| > 1} \left[ \sum_{u=1}^{d_2} \alpha_{1,j}^{(u)} \cdot \frac{\frac{\partial g}{\partial W_e^{(u)}}(X, W_{e_j}^*)}{1 + \exp(-g(X, W_{e_j}^*))} \right.$$

$$\left. + \sum_{u,v=1}^{d_2} \frac{\alpha_{2,j}^{(uv)}}{1 + 1_{\{u=v\}}} \cdot \frac{\frac{\partial^2 g}{\partial W_e^{(u)} \partial W_e^{(v)}}(X, W_{e_j}^*) + \frac{\partial g}{\partial W_e^{(u)}}(X, W_{e_j}^*) \frac{\partial g}{\partial W_e^{(v)}}(X, W_{e_j}^*)}{1 + \exp(-g(X, W_{e_j}^*))} \right] H_j(Y|X),$$

and

$$\lim_{n \to \infty} \frac{T_{n,3}(Y|X)}{m_n \mathcal{L}_1(G_n, G_*)} := \sum_{j=1}^{N^*} \xi_j [F_{0,j}(Y|X) - H_j(Y|X)].$$

It is worth noting that for almost every $X$, the set

$$\left\{ F_{\rho,j}(Y|X), \ H_j(Y|X) : 0 \le \rho \le 4, j \in [N^*] \right\}$$

is linearly independent w.r.t $Y$. Therefore, it follows that the coefficients of those terms in the limit in equation equation (20) become zero.

For $j \in [N^*]$ such that $|\mathcal{C}_j| = 1$, by considering the coefficients of

- $F_{0,j}(Y|X)$, we have $\xi_j + \sum_{u=1}^{d_2} \alpha_{1,j}^{(u)} \cdot \frac{\frac{\partial g}{\partial W_e^{(u)}}(X, W_{e_j}^*)}{1 + \exp(-g(X, W_{e_j}^*))} = 0$ for almost surely $X$. Since the expert function $g$ is strongly identifiable, we deduce $\xi_j = \alpha_{1,j}^{(u)} = 0$ for all $u \in [d_2]$;

- $F_{2,j}(Y|X)$, we have $\beta_{1,j} = 0$.

For $j \in [N^*]$ such that $|\mathcal{C}_j| > 1$, by considering the coefficients of

- $F_{0,j}(Y|X)$, we have

$$\xi_j + \sum_{u=1}^{d_2} \alpha_{1,j}^{(u)} \frac{\frac{\partial g}{\partial W_e^{(u)}}(X, W_{e_j}^*)}{1 + \exp(-g(X, W_{e_j}^*))}$$

$$+ \sum_{u,v=1}^{d_2} \frac{\alpha_{2,j}^{(uv)}}{1 + 1_{\{u=v\}}} \cdot \frac{\frac{\partial^2 g}{\partial W_e^{(u)} \partial W_e^{(v)}}(X, W_{e_j}^*) + \frac{\partial g}{\partial W_e^{(u)}}(X, W_{e_j}^*)\frac{\partial g}{\partial W_e^{(v)}}(X, W_{e_j}^*)}{1 + \exp(-g(X, W_{e_j}^*))} = 0$$

for almost surely $X$. Since the expert function $g$ is strongly identifiable, we deduce $\xi_j = \alpha_{1,j}^{(u)} = \alpha_{2,j}^{(uv)} = 0$ for all $u, v \in [d_2]$;

- $F_{3,j}(Y|X)$, we have $\sum_{u=1}^{d_2} \frac{1}{2} \gamma_j^{(u)} \frac{\partial g}{\partial W_e^{(u)}}(X, W_{e_j}^*) = 0$ for almost surely $X$. Since the expert function $g$ is strongly identifiable, we deduce $\gamma_j^{(u)} = 0$ for all $u \in [d_2]$;

- $F_{4,j}(Y|X)$, we have $\beta_{2,j} = 0$.

Putting the above results together, we have (ii) $\xi_j = \alpha_{1,j}^{(u)} = \beta_{1,j} = \alpha_{2,j}^{(uv)} = \beta_{2,j} = \gamma_j^{(u)} = 0$ for all $j \in [N^*]$ and $u, v \in [d_2]$. This contradicts to the fact that at least one among them is non-zero. Consequently, we achieve the local part in equation (17).

**Global part:** Now, it suffices to demonstrate that

$$\inf_{G \in \mathcal{G}_N(\Theta): \mathcal{L}_1(G, G_*) > \varepsilon'} \frac{\mathbb{E}_X[V(p_G(\cdot|X), p_{G_*}(\cdot|X))]}{\mathcal{L}_1(G, G_*)} > 0,$$

for some positive constant $\varepsilon'$. Given the above result, it is sufficient to derive the global part in equation (18), that is,

$$\inf_{G \in \mathcal{G}_N(\Theta): \mathcal{L}_1(G, G_*) > \varepsilon'} \mathbb{E}_X[V(p_G(\cdot|X), p_{G_*}(\cdot|X))]/\mathcal{L}_1(G, G_*) > 0.$$

Assume by contrary that the global part does not hold true, then we can find a sequence $\widetilde{G}_n \in \mathcal{G}_N(\Theta)$ such that $\mathcal{L}_1(\widetilde{G}_n, G_*) > \varepsilon'$ and $\mathbb{E}_X[V(p_{\widetilde{G}_n}(\cdot|X), p_{G_*}(\cdot|X))] \to 0$ as $n \to \infty$. Since $\Theta$ is a compact set, we are able to replace $\widetilde{G}_n$ with its subsequence which converges to some mixing measure $\widetilde{G} \in \mathcal{G}_N(\Theta)$. Recall that $\mathcal{L}_1(\widetilde{G}_n, G_*) > \varepsilon'$, then we also get that $\mathcal{L}_1(\widetilde{G}, G_*) > \varepsilon'$.

On the other hand, by means of the Fatou's lemma, we have

$$0 = \lim_{n \to \infty} \mathbb{E}_X[2V(p_{\widetilde{G}_n}(\cdot|X), p_{G_*}(\cdot|X))] \ge \int \liminf_{n \to \infty} |p_{\widetilde{G}_n}(Y|X) - p_{G_*}(Y|X)| d(X, Y),$$

which follows that $p_{\widetilde{G}}(Y|X) - p_{G_*}(Y|X) = 0$ for almost surely $(X, Y)$. Thus, we achieve that $\widetilde{G} \equiv G_*$, or equivalently $\mathcal{L}_1(\widetilde{G}, G_*) = 0$. This contradicts to the fact that $\mathcal{L}_1(\widetilde{G}, G_*) > \varepsilon' > 0$.

Hence, we reach the conclusion in equation (18), and the proof is completed.

## K.2   PROOF OF THEOREM J.1

As in Appendix K.1, we also start with establishing the local part

$$\lim_{\varepsilon \to 0} \inf_{G \in \mathcal{G}_N(\Theta): \mathcal{L}_2(G,G_*) \leq \varepsilon} \frac{\mathbb{E}_X[V(p_G(\cdot|X), p_{G_*}(\cdot|X))]}{\mathcal{L}_2(G, G_*)} > 0. \tag{21}$$

Assume by contrary that the local part is not true, then we can find a sequence of mixing measures $(G_n)$ given by $G_n := \sum_{i=1}^N \exp(c_i^n)\delta_{(a_i^n, b_i^n, \nu_i^n)} \in \mathcal{G}_N(\Theta)$ such that $\mathcal{L}_2(G_n, G_*) \to 0$ and

$$\mathbb{E}_X[V(p_{G_n}(\cdot|X), p_{G_*}(\cdot|X))]/\mathcal{L}_2(G_n, G_*) \to 0,$$

as $n \to \infty$. Recall that the Voronoi loss $\mathcal{L}_2(G_n, G_*)$ is given by

$$\mathcal{L}_2(G_n, G_*) := \sum_{j=1}^{N^*} \Big| \sum_{i \in \mathcal{C}_j} \exp(c_i^n) - \exp(c_j^*) \Big| + \sum_{j \in [N^*]:|\mathcal{C}_j|=1} \sum_{i \in \mathcal{C}_j} \exp(c_i^n)\Big[ \|W_{e_i}^n - W_{e_j}^*\| + |\nu_i^n - \nu_j^*| \Big]$$

$$+ \sum_{j \in [N^*]:|\mathcal{C}_j|=1} \sum_{i \in \mathcal{C}_j} \exp(c_i^n)\Big[ \|W_{e_i}^n - W_{e_j}^*\|^2 + |\nu_i^n - \nu_j^*|^2 \Big]. \tag{22}$$

Since $\mathcal{L}_2(G_n, G_*) \to 0$ as $n \to \infty$, we obtain $(a_i^n, b_i^n, \nu_i^n) \to (a_j^*, b_j^*, \nu_j^*)$ for all $j \in [N^*]$ and $i \in \mathcal{C}_j$.

Next, we divide the rest of this proof into three main steps.

**Step 1: Taylor expansion.** In this step, we aim to decompose the term $T_n(Y|X) := \Big[ \sum_{j=1}^{N^*} \exp(\log(1 + \exp((a_j^*)^\top X + b_j^*))) \Big] \cdot [p_{G_n}(Y|X) - p_{G_*}(Y|X)]$ can be decomposed as

$$T_n(Y|X) = \sum_{j=1}^{N^*} \sum_{i \in \mathcal{C}_j} \exp(c_i^n) \Big[ \exp(\log(1 + \exp((a_i^n)^\top X + b_i^n))) f(Y|(a_i^n)^\top X + b_i^n, \nu_i^n)$$

$$- \exp(\log(1 + \exp((a_j^*)^\top X + b_j^*))) f(Y|(a_j^*)^\top X + b_j^*, \nu_j^*) \Big]$$

$$- \sum_{j=1}^{N^*} \sum_{i \in \mathcal{C}_j} \exp(c_i^n) \Big[ \exp(\log(1 + \exp((a_i^n)^\top X + b_i^n))) - \exp(\log(1 + \exp((a_j^*)^\top X + b_j^*))) \Big] p_{G_n}(Y|X)$$

$$+ \sum_{j=1}^{N^*} \Big[ \sum_{i \in \mathcal{C}_j} \exp(c_i^n) - \exp(c_j^*) \Big] \cdot \exp(\log(1 + \exp((a_j^*)^\top X + b_j^*)))[f(Y|(a_j^*)^\top X + b_j^*, \nu_j^*) - p_{G_n}(Y|X)]$$

$$:= T_{n,1}(Y|X) - T_{n,2}(Y|X) + T_{n,3}(Y|X).$$

Next, we continue to decompose the term $T_{n,1}(Y|X)$ as

$$T_{n,1}(Y|X) = \sum_{j \in [N^*]:|\mathcal{C}_j|=1} \sum_{i \in \mathcal{C}_j} \exp(c_i^n) \Big[ \exp(\log(1 + \exp((a_i^n)^\top X + b_i^n))) f(Y|(a_i^n)^\top X + b_i^n, \nu_i^n)$$

$$- \exp(\log(1 + \exp((a_j^*)^\top X + b_j^*))) f(Y|(a_j^*)^\top X + b_j^*, \nu_j^*) \Big]$$

$$+ \sum_{j \in [N^*]:|\mathcal{C}_j|>1} \sum_{i \in \mathcal{C}_j} \exp(c_i^n) \Big[ \exp(\log(1 + \exp((a_i^n)^\top X + b_i^n))) f(Y|(a_i^n)^\top X + b_i^n, \nu_i^n)$$

$$- \exp(\log(1 + \exp((a_j^*)^\top X + b_j^*))) f(Y|(a_j^*)^\top X + b_j^*, \nu_j^*) \Big]$$

$$:= T_{n,1,1}(Y|X) + T_{n,1,2}(Y|X).$$

Let us denote $F_\rho(Y|X; a, b, \nu) := \exp(\log(1 + \exp(a^\top X + b))) \frac{\partial^\rho f}{\partial g^\rho}(Y|a^\top X + b, \nu)$. By applying the first-order Taylor expansion to the function $F_0(Y|X; a, b, \nu)$ around the point $(a_j^*, b_j^*, \nu_j^*)$, we rewrite the term $T_{n,1,1}(Y|X)$ as

$$T_{n,1,1}(Y|X) = \sum_{j \in [N^*]:|\mathcal{C}_j|=1} \sum_{\rho=0}^{2} T_{n,1,1,\rho}^{(j)}(X) F_\rho(Y; X, a_j^*, b_j^*, \nu_j^*) + R_{n,1,1}(Y|X),$$

where $R_{n,1,1}(Y|X)$ is the Taylor remainder such that $R_{n,1,1}(Y|X)/\mathcal{L}_2(G_n, G_*) \to 0$ as $n \to \infty$, and

$$T_{n,1,1,0}^{(j)}(X) := \sum_{i \in \mathcal{C}_j} \exp(c_i^n) \cdot \frac{\sum_{u=1}^d (\Delta a_{ij}^n)^{(u)} X^{(u)} + (\Delta b_{ij}^n)}{1 + \exp(-(a_j^*)^\top X - b_j^*)},$$

$$T_{n,1,1,1}^{(j)}(X) := \sum_{i \in \mathcal{C}_j} \exp(c_i^n) \left[ \sum_{u=1}^d (\Delta a_{ij}^n)^{(u)} X^{(u)} + (\Delta b_{ij}^n) \right],$$

$$T_{n,1,1,2}^{(j)}(X) := \sum_{i \in \mathcal{C}_j} \frac{1}{2} \exp(c_i^n)(\Delta \nu_{ij}^n),$$

in which $\Delta a_{ij}^n := a_i^n - a_j^*$, $\Delta b_{ij}^n := b_i^n - b_j^*$ and $\Delta \nu_{ij}^n := \nu_i^n - \nu_j^*$.

Meanwhile, by means of the second-order Taylor expansion, the term $T_{n,1,2}(Y|X)$ can be represented as

$$T_{n,1,2}(Y|X) = \sum_{j \in [N^*]:|\mathcal{C}_j|>1} \sum_{\rho=0}^4 T_{n,1,2,\rho}^{(j)}(X) F_\rho(Y; X, a_j^*, b_j^*, \nu_j^*) + R_{n,1,2}(Y|X),$$

where $R_{n,1,2}(Y|X)$ is the Taylor remainder such that $R_{n,1,2}(Y|X)/\mathcal{L}_2(G_n, G_*) \to 0$ as $n \to \infty$, and

$$T_{n,1,2,0}^{(j)}(X) := \sum_{i \in \mathcal{C}_j} \exp(c_i^n) \left[ \frac{\sum_{u=1}^d (\Delta a_{ij}^n)^{(u)} X^{(u)} + (\Delta b_{ij}^n)}{1 + \exp(-(a_j^*)^\top X - b_j^*)} + \frac{\sum_{u,v=1}^d \frac{(\Delta a_{ij}^n)^{(u)}(\Delta a_{ij}^n)^{(v)}}{1 + 1_{\{u=v\}}} X^{(u)} X^{(v)}}{1 + \exp(-(a_j^*)^\top X - b_j^*)} \right. $$
$$ \left. + \frac{\sum_{u=1}^d (\Delta a_{ij}^n)^{(u)}(\Delta b_{ij}^n) X^{(u)} + \frac{1}{2}(\Delta b_{ij}^n)^2}{1 + \exp(-(a_j^*)^\top X - b_j^*)} \right],$$

$$T_{n,1,2,1}^{(j)}(X) := \sum_{i \in \mathcal{C}_j} \exp(c_i^n) \left[ \sum_{u=1}^d (\Delta a_{ij}^n)^{(u)} X^{(u)} + (\Delta b_{ij}^n) + \frac{2 \sum_{u,v=1}^d \frac{(\Delta a_{ij}^n)^{(u)}(\Delta a_{ij}^n)^{(v)}}{1 + 1_{\{u=v\}}} X^{(u)} X^{(v)}}{1 + \exp(-(a_j^*)^\top X - b_j^*)} \right. $$
$$ \left. + \frac{(\Delta b_{ij}^n)^2 + 2 \sum_{u=1}^d (\Delta a_{ij}^n)^{(u)}(\Delta b_{ij}^n) X^{(u)}}{1 + \exp(-(a_j^*)^\top X - b_j^*)} \right],$$

$$T_{n,1,2,2}^{(j)}(X) := \sum_{i \in \mathcal{C}_j} \exp(c_i^n) \left[ \frac{1}{2}(\Delta \nu_{ij}^n) + \sum_{u,v=1}^d \frac{(\Delta a_{ij}^n)^{(u)}(\Delta a_{ij}^n)^{(v)} X^{(u)} X^{(v)}}{1 + 1_{\{u=v\}}} + \frac{1}{2}(\Delta b_{ij}^n)^2 \right. $$
$$ \left. + \sum_{u=1}^d (\Delta a_{ij}^n)^{(u)}(\Delta b_{ij}^n) X^{(u)} + \frac{1}{2} \cdot \frac{\sum_{u=1}^d (\Delta a_{ij}^n)^{(u)}(\Delta \nu_{ij}^n) X^{(u)} + (\Delta b_{ij}^n)(\Delta \nu_{ij}^n)}{1 + \exp(-(a_j^*)^\top X - b_j^*)} \right],$$

$$T_{n,1,2,3}^{(j)}(X) := \sum_{i \in \mathcal{C}_j} \exp(c_i^n) \left[ \sum_{u=1}^d \frac{1}{2}(\Delta a_{ij}^n)^{(u)}(\Delta \nu_{ij}^n) X^{(u)} + \frac{1}{2}(\Delta b_{ij}^n)(\Delta \nu_{ij}^n) \right],$$

$$T_{n,1,2,4}^{(j)}(X) := \sum_{i \in \mathcal{C}_j} \exp(c_i^n) \cdot \frac{1}{4}(\Delta \nu_{ij}^n)^2.$$

Next, we decompose the term $T_{n,2}(Y|X)$ as

$$T_{n,2}(Y|X)$$
$$:= \sum_{j \in [N^*]:|\mathcal{C}_j|=1} \sum_{i \in \mathcal{C}_j} \exp(c_i^n) \left[ \exp(\log(1 + \exp((a_i^n)^\top X + b_i^n))) - \exp(\log(1 + \exp((a_j^*)^\top X + b_j^*))) \right] p_{G_n}(Y|X)$$
$$+ \sum_{j \in [N^*]:|\mathcal{C}_j|>1} \sum_{i \in \mathcal{C}_j} \exp(c_i^n) \left[ \exp(\log(1 + \exp((a_i^n)^\top X + b_i^n))) - \exp(\log(1 + \exp((a_j^*)^\top X + b_j^*))) \right] p_{G_n}(Y|X)$$
$$:= T_{n,2,1}(Y|X) + T_{n,2,2}(Y|X).$$

Note that we can rewrite the term $T_{n,1,2}(Y|X)$ using the first-order Taylor expansion to the function $\exp(\log(1 + \exp((a_i^n)^\top X + b_i^n)))$ around the point $(a_j^*, b_j^*)$ as

$$T_{n,2,1}(Y|X) = \sum_{j \in [N^*]:|\mathcal{C}_j|=1} \sum_{i \in \mathcal{C}_j} \exp(c_i^n) \cdot \frac{\sum_{u=1}^d (\Delta a_{ij}^n)^{(u)} X^{(u)} + (\Delta b_{ij}^n)}{1 + \exp(-(a_j^*)^\top X - b_j^*)} H_n(Y|X; a_j^*, b_j^*)$$
$$+ R_{n,2,1}(Y|X),$$

where we denote $H_n(Y|X; a, b) = \exp(\log(1 + \exp(a^\top X + b))) p_{G_n}(Y|X)$ and $R_{n,2,1}(Y|X)$ is the Taylor remainder such that $R_{n,2,1}(Y|X)/\mathcal{L}_2(G_n, G_*) \to 0$ as $n \to \infty$.

On the other hand, by means of the second-order Taylor expansion, we have

$$T_{n,2,2}(Y|X) = \sum_{j \in [N^*]:|\mathcal{C}_j|>1} \sum_{i \in \mathcal{C}_j} \exp(c_i^n) \left[ \frac{\sum_{u=1}^d (\Delta a_{ij}^n)^{(u)} X^{(u)} + (\Delta b_{ij}^n)}{1 + \exp(-(a_j^*)^\top X - b_j^*)} \right.$$
$$\left. + \frac{\sum_{u,v=1}^d \frac{(\Delta a_{ij}^n)^{(u)}(\Delta a_{ij}^n)^{(v)}}{1+1_{\{u=v\}}} X^{(u)} X^{(v)}}{1 + \exp(-(a_j^*)^\top X - b_j^*)} + \frac{\sum_{u=1}^d (\Delta a_{ij}^n)^{(u)}(\Delta b_{ij}^n) X^{(u)} + \frac{1}{2}(\Delta b_{ij}^n)^2}{1 + \exp(-(a_j^*)^\top X - b_j^*)} \right] H_n(Y|X; W_{e_j}^*)$$
$$+ R_{n,2,2}(Y|X),$$

where $R_{n,2,1}(Y|X)$ is the Taylor remainder such that $R_{n,2,2}(Y|X)/\mathcal{L}_2(G_n, G_*) \to 0$ as $n \to \infty$.

From the above equation, $[T_{n,1,1}(Y|X) - R_{n,1,1}(Y|X)]$, $[T_{n,1,2}(Y|X) - R_{n,1,2}(Y|X)]$, $[T_{n,2,1}(Y|X) - R_{n,2,1}(Y|X)]$, $[T_{n,2,2}(Y|X) - R_{n,2,2}(Y|X)]$ and $[T_{n,3}(Y|X)]$ can be seen as a combination of elements of the set $\mathcal{S} := \bigcup_{j=1}^N \bigcup_{\rho=0}^5 \mathcal{S}_{\rho,j}$, where we define

$$\mathcal{S}_{0,j} := \left\{ \frac{X^{(u)}}{1 + \exp(-(a_j^*)^\top X - b_j^*)} F_{0,j}(Y|X), \ \frac{X^{(u)} X^{(v)}}{1 + \exp(-(a_j^*)^\top X - b_j^*)} F_{0,j}(Y|X), \right.$$
$$\left. \frac{1}{1 + \exp(-(a_j^*)^\top X - b_j^*)} F_{0,j}(Y|X), \ F_{0,j}(Y|X) : 1 \le u, v \le d \right\},$$

$$\mathcal{S}_{1,j} := \left\{ F_{1,j}(Y|X), \ X^{(u)} F_{1,j}(Y|X), \ \frac{X^{(u)}}{1 + \exp(-(a_j^*)^\top X - b_j^*)} F_{1,j}(Y|X), \right.$$
$$\left. \frac{X^{(u)} X^{(v)}}{1 + \exp(-(a_j^*)^\top X - b_j^*)} F_{1,j}(Y|X), \ \frac{1}{1 + \exp(-(a_j^*)^\top X - b_j^*)} F_{1,j}(Y|X) : 1 \le u, v \le d \right\},$$

$$\mathcal{S}_{2,j} := \left\{ F_{2,j}(Y|X), \ X^{(u)} F_{2,j}(Y|X), \ X^{(u)} X^{(v)} F_{2,j}(Y|X), \right.$$
$$\left. \frac{X^{(u)}}{1 + \exp(-(a_j^*)^\top X - b_j^*)} F_{2,j}(Y|X), \ \frac{1}{1 + \exp(-(a_j^*)^\top X - b_j^*)} F_{2,j}(Y|X) : 1 \le u, v \le d_2 \right\},$$

$$\mathcal{S}_{3,j} := \left\{ F_{3,j}(Y|X), \ X^{(u)} F_{3,j}(Y|X) : 1 \le u \le d \right\},$$

$$\mathcal{S}_{4,j} := \left\{ F_{4,j}(Y|X) \right\},$$

$$\mathcal{S}_{5,j} := \left\{ \frac{X^{(u)}}{1 + \exp(-(a_j^*)^\top X - b_j^*)} H_{n,j}(Y|X), \ \frac{X^{(u)} X^{(v)}}{1 + \exp(-(a_j^*)^\top X - b_j^*)} H_{n,j}(Y|X), \right.$$
$$\left. \frac{1}{1 + \exp(-(a_j^*)^\top X - b_j^*)} H_{n,j}(Y|X), \ H_{n,j}(Y|X) : 1 \le u, v \le d \right\}.$$

**Step 2: Non-vanishing coefficients.** In this step, we will show that at least one among the coefficients in the representations of $[T_{n,1,1}(Y|X) - R_{n,1,1}(Y|X)]/\mathcal{L}_2(G_n, G_*)$,

$[T_{n,1,2}(Y|X) - R_{n,1,2}(Y|X)]/\mathcal{L}_2(G_n, G_*)$, $[T_{n,2,1}(Y|X) - R_{n,2,1}(Y|X)]/\mathcal{L}_2(G_n, G_*)$, $[T_{n,2,2}(Y|X) - R_{n,2,2}(Y|X)]/\mathcal{L}_2(G_n, G_*)$ and $[T_{n,3}(Y|X)]/\mathcal{L}_2(G_n, G_*)$ does not approach zero when $n$ goes to infinity. Assume by contrary that all of them vanish as $n \to \infty$. Then, by considering the coefficients of the term

- $F_{0,j}(Y|X)$ for $j \in [N^*]$, we have

$$\frac{1}{\mathcal{L}_2(G_n, G_*)} \cdot \sum_{j=1}^{N^*} \Big| \sum_{i \in \mathcal{C}_j} \exp(c_i^n) - \exp(c_j^*) \Big| \to 0.$$

- $\frac{X^{(u)}}{1+\exp(-(a_j^*)^\top X - b_j^*)} F_{0,j}(Y|X)$ for $j \in [N^*] : |\mathcal{C}_j| = 1$, we have

$$\frac{1}{\mathcal{L}_2(G_n, G_*)} \cdot \sum_{j \in [N^*]:|\mathcal{C}_j|=1} \sum_{i \in \mathcal{C}_j} \exp(c_i^n) \|\Delta a_{ij}^n\| \to 0.$$

- $\frac{1}{1+\exp(-(a_j^*)^\top X - b_j^*)} F_{0,j}(Y|X)$ for $j \in [N^*] : |\mathcal{C}_j| = 1$, we have

$$\frac{1}{\mathcal{L}_2(G_n, G_*)} \cdot \sum_{j \in [N^*]:|\mathcal{C}_j|=1} \sum_{i \in \mathcal{C}_j} \exp(c_i^n) \|\Delta b_{ij}^n\| \to 0.$$

- $F_{2,j}(Y|X)$ for $j \in [N^*] : |\mathcal{C}_j| = 1$, we have

$$\frac{1}{\mathcal{L}_2(G_n, G_*)} \cdot \sum_{j \in [N^*]:|\mathcal{C}_j|=1} \sum_{i \in \mathcal{C}_j} \exp(c_i^n) |\Delta \nu_{ij}^n| \to 0.$$

- $\frac{X^{(u)} X^{(v)}}{1+\exp(-(a_j^*)^\top X - b_j^*))} F_{0,j}(Y|X)$ for $j \in [N^*] : |\mathcal{C}_j| > 1$, we have

$$\frac{1}{\mathcal{L}_2(G_n, G_*)} \cdot \sum_{j \in [N^*]:|\mathcal{C}_j|>1} \sum_{i \in \mathcal{C}_j} \exp(c_i^n) \|\Delta a_j^n\|^2 \to 0.$$

- $\frac{1}{1+\exp(-(a_j^*)^\top X - b_j^*))} F_{1,j}(Y|X)$ for $j \in [N^*] : |\mathcal{C}_j| > 1$, we have

$$\frac{1}{\mathcal{L}_2(G_n, G_*)} \cdot \sum_{j \in [N^*]:|\mathcal{C}_j|>1} \sum_{i \in \mathcal{C}_j} \exp(c_i^n) |\Delta b_j^n|^2 \to 0.$$

- $F_{4,j}(Y|X)$ for $j \in [N^*] : |\mathcal{C}_j| > 1$, we have

$$\frac{1}{\mathcal{L}_2(G_n, G_*)} \cdot \sum_{j \in [N^*]:|\mathcal{C}_j|=1} \sum_{i \in \mathcal{C}_j} \exp(c_i^n) |\Delta \nu_{ij}^n|^2 \to 0.$$

By taking the sum of the above limits, we obtain $1 = \frac{\mathcal{L}_2(G_n, G_*)}{\mathcal{L}_2(G_n, G_*)} \to 0$ as $n \to \infty$, which is a contradiction. Thus, not all the coefficients in the representations of $[T_{n,1,1}(Y|X) - R_{n,1,1}(Y|X)]/\mathcal{L}_2(G_n, G_*)$, $[T_{n,1,2}(Y|X) - R_{n,1,2}(Y|X)]/\mathcal{L}_2(G_n, G_*)$, $[T_{n,2,1}(Y|X) - R_{n,2,1}(Y|X)]/\mathcal{L}_2(G_n, G_*)$, $[T_{n,2,2}(Y|X) - R_{n,2,2}(Y|X)]/\mathcal{L}_2(G_n, G_*)$ and $[T_{n,3}(Y|X)]/\mathcal{L}_2(G_n, G_*)$ converge to zero as $n \to \infty$.

**Stage 3 - Fatou's argument:** In this stage, we use the Fatou's lemma to show a contradiction to the result of Step 2. For that purpose, let us denote $m_n$ as the maximum of the absolute values of the coefficients in the representations of $[T_{n,1,1}(Y|X) - R_{n,1,1}(Y|X)]/\mathcal{L}_2(G_n, G_*)$, $[T_{n,1,2}(Y|X) - R_{n,1,2}(Y|X)]/\mathcal{L}_2(G_n, G_*)$, $[T_{n,2,1}(Y|X) - R_{n,2,1}(Y|X)]/\mathcal{L}_2(G_n, G_*)$, $[T_{n,2,2}(Y|X) - R_{n,2,2}(Y|X)]/\mathcal{L}_2(G_n, G_*)$ and $[T_{n,3}(Y|X)]/\mathcal{L}_2(G_n, G_*)$. It follows from the result of Step 2 that $1/m_n \not\to \infty$ as $n \to \infty$.

In addition, we also denote

$$\frac{\sum_{i\in\mathcal{C}_j}\exp(c_i^n)(\Delta a_{ij}^n)^{(u)}}{m_n\mathcal{L}_2(G_n,G_*)}\to\alpha_{1,j}^{(u)}, \qquad \frac{\sum_{i\in\mathcal{C}_j}\exp(c_i^n)(\Delta\nu_{ij}^n)}{m_n\mathcal{L}_2(G_n,G_*)}\to\beta_{1,j},$$

$$\frac{\sum_{i\in\mathcal{C}_j}\exp(c_i^n)(\Delta a_{ij}^n)^{(u)}(\Delta a_{ij}^n)^{(v)}}{m_n\mathcal{L}_2(G_n,G_*)}\to\alpha_{2,j}^{(uv)}, \qquad \frac{\sum_{i\in\mathcal{C}_j}\exp(c_i^n)(\Delta\nu_{ij}^n)^2}{m_n\mathcal{L}_2(G_n,G_*)}\to\beta_{2,j},$$

$$\frac{\sum_{i\in\mathcal{C}_j}\exp(c_i^n)(\Delta b_{ij}^n)}{m_n\mathcal{L}_2(G_n,G_*)}\to\phi_{1,j}^{(u)}, \qquad \frac{\sum_{i\in\mathcal{C}_j}\exp(c_i^n)(\Delta b_{ij}^n)^2}{m_n\mathcal{L}_2(G_n,G_*)}\to\phi_{2,j},$$

$$\frac{\sum_{i\in\mathcal{C}_j}\exp(c_i^n)(\Delta a_{ij}^n)^{(u)}(\Delta\nu_{ij}^n)}{m_n\mathcal{L}_2(G_n,G_*)}\to\gamma_{1,j}^{(u)}, \qquad \frac{\sum_{i\in\mathcal{C}_j}\exp(c_i^n)(\Delta a_{ij}^n)^{(u)}(\Delta b_{ij}^n)}{m_n\mathcal{L}_2(G_n,G_*)}\to\gamma_{2,j}^{(u)},$$

$$\frac{\sum_{i\in\mathcal{C}_j}\exp(c_i^n)(\Delta b_{ij}^n)(\Delta\nu_{ij}^n)}{m_n\mathcal{L}_2(G_n,G_*)}\to\gamma_{3,j}, \qquad \frac{\sum_{i\in\mathcal{C}_j}\exp(c_i^n)-\exp(c_j^*)}{m_n\mathcal{L}_2(G_n,G_*)}\to\xi_j,$$

as $n\to\infty$ for any $j\in[N^*]$ and $u,v\in[d_2]$ with a note that at least one among $\alpha_{1,j}^{(u)},\beta_{1,j},\alpha_{2,j}^{(uv)},\beta_{2,j}$, $\phi_{1,j},\phi_{2,j},\gamma_{1,j}^{(u)},\gamma_{2,j}^{(u)},\gamma_{3,j}$ and $\xi_j$ is non-zero.

By applying the Fatou's lemma, we have

$$0=\lim_{n\to\infty}\frac{\mathbb{E}_X[V(p_G(\cdot|X),p_{G_*}(\cdot|X))]}{m_n\mathcal{L}_2(G_n,G_*)}=\frac{1}{2}\int\liminf_{n\to\infty}\frac{|p_{G_n}(Y|X)-p_{G_*}(Y|X)|}{m_n\mathcal{L}_2(G_n,G_*)}\mathrm{d}(X,Y),$$

which implies that $[p_{G_n}(Y|X)-p_{G_*}(Y|X)]/[m_n\mathcal{L}_2(G_n,G_*)]\to 0$ as $n\to\infty$ for almost surely $(X,Y)$. Since the term $\sum_{j=1}^{N^*}\exp(\log(1+\exp((a_j^*)^\top X+b_j^*)))$ is bounded, we also have $T_n(Y|X)/[m_n\mathcal{L}_2(G_n,G_*)]\to 0$ as $n\to\infty$. Then, it follows that

$$0=\lim_{n\to\infty}\frac{T_{n,1,1}(Y|X)+T_{n,1,2}(Y|X)}{m_n\mathcal{L}_2(G_n,G_*)}-\lim_{n\to\infty}\frac{T_{n,2,1}(Y|X)+T_{n,2,2}(Y|X)}{m_n\mathcal{L}_2(G_n,G_*)}+\lim_{n\to\infty}\frac{T_{n,3}(Y|X)}{m_n\mathcal{L}_2(G_n,G_*)},$$
$$\tag{23}$$

for almost surely $(X,Y)\in\mathcal{X}\times\mathcal{Y}$, where we have

$$\lim_{n\to\infty}\frac{T_{n,1,1}(Y|X)}{m_n\mathcal{L}_2(G_n,G_*)}:=\sum_{j\in[N^*]:|\mathcal{C}_j|=1}\left[\frac{\sum_{u=1}^d\alpha_{1,j}^{(u)}X^{(u)}+\phi_{1,j}}{1+\exp(-(a_j^*)^\top X-b_j^*)}F_{0,j}(Y|X)\right.$$

$$\left.+\Big(\sum_{u=1}^d\alpha_{1,j}^{(u)}X^{(u)}+\phi_{1,j}\Big)F_{1,j}(Y|X)+\frac{1}{2}\beta_{1,j}F_{2,j}(Y|X)\right],$$

$$\lim_{n\to\infty}\frac{T_{n,1,2}(Y|X)}{m_n\mathcal{L}_2(G_n,G_*)}:=\sum_{j\in[N^*]:|\mathcal{C}_j|>1}\left[\left(\frac{\sum_{u=1}^d\alpha_{1,j}^{(u)}X^{(u)}+\phi_{1,j}}{1+\exp(-(a_j^*)^\top X-b_j^*)}\right.\right.$$

$$+\frac{\sum_{u,v=1}^d\frac{\alpha_{2,j}^{(uv)}}{1+1_{\{u=v\}}}X^{(u)}X^{(v)}}{1+\exp(-(a_j^*)^\top X-b_j^*)}+\frac{\sum_{u=1}^d\gamma_{2,j}^{(u)}X^{(u)}+\frac{1}{2}\phi_{2,j}}{1+\exp(-(a_j^*)^\top X-b_j^*)}\Bigg)F_{0,j}(Y|X)$$

$$+\left(\sum_{u=1}^d\alpha_{1,j}^{(u)}X^{(u)}+\phi_{1,j}+\frac{2\sum_{u,v=1}^d\frac{\alpha_{2,j}^{(uv)}}{1+1_{\{u=v\}}}X^{(u)}X^{(v)}}{1+\exp(-(a_j^*)^\top X-b_j^*)}\right.$$

$$+\frac{\phi_{2,j}+2\sum_{u=1}^d\gamma_{2,j}^{(u)}X^{(u)}}{1+\exp(-(a_j^*)^\top X-b_j^*)}\Bigg)F_{1,j}(Y|X)+\left(\frac{1}{2}\beta_{1,j}+\sum_{u,v=1}^d\frac{\alpha_{2,j}^{(uv)}X^{(u)}X^{(v)}}{1+1_{\{u=v\}}}+\frac{1}{2}\phi_{2,j}\right.$$

$$+\sum_{u=1}^d\gamma_{2,j}^{(u)}X^{(u)}+\frac{1}{2}\cdot\frac{\sum_{u=1}^d\gamma_{1,j}^{(u)}X^{(u)}+\gamma_{3,j}}{1+\exp(-(a_j^*)^\top X-b_j^*)}\Bigg)F_{2,j}(Y|X)$$

$$+\Big(\sum_{u=1}^d\frac{1}{2}\gamma_{1,j}^{(u)}X^{(u)}+\frac{1}{2}\gamma_{3,j}\Big)F_{3,j}(Y|X)+\frac{1}{4}\beta_{2,j}F_{4,j}(Y|X)\Bigg],$$

and

$$\lim_{n\to\infty} \frac{T_{n,2,1}(Y|X)}{m_n \mathcal{L}_2(G_n, G_*)} := \sum_{j \in [N^*]:|\mathcal{C}_j|=1} \frac{\sum_{u=1}^d \alpha_{1,j}^{(u)} X^{(u)} + \phi_{1,j}}{1 + \exp(-(a_j^*)^\top X - b_j^*)} H_j(Y|X),$$

$$\lim_{n\to\infty} \frac{T_{n,2,2}(Y|X)}{m_n \mathcal{L}_2(G_n, G_*)} := \sum_{j \in [N^*]:|\mathcal{C}_j|>1} \left[ \frac{\sum_{u=1}^d \alpha_{1,j}^{(u)} X^{(u)} + \phi_{1,j}}{1 + \exp(-(a_j^*)^\top X - b_j^*)} \right.$$

$$+ \frac{\sum_{u,v=1}^d \frac{\alpha_{2,j}^{(uv)}}{1+1_{\{u=v\}}} X^{(u)} X^{(v)}}{1 + \exp(-(a_j^*)^\top X - b_j^*)} + \left. \frac{\sum_{u=1}^d \gamma_{2,j}^{(u)} X^{(u)} + \frac{1}{2}\phi_{2,j}}{1 + \exp(-(a_j^*)^\top X - b_j^*)} \right] H_j(Y|X),$$

and

$$\lim_{n\to\infty} \frac{T_{n,3}(Y|X)}{m_n \mathcal{L}_2(G_n, G_*)} := \sum_{j=1}^{N^*} \xi_j [F_{0,j}(Y|X) - H_j(Y|X)].$$

It is worth noting that for almost every $X$, the set

$$\left\{ F_{\rho,j}(Y|X),\ H_j(Y|X) : 0 \le \rho \le 4, j \in [N^*] \right\}$$

is linearly independent w.r.t $Y$. Therefore, it follows that the coefficients of those terms in the limit in equation equation (23) become zero.

For $j \in [N^*]$ such that $|\mathcal{C}_j| = 1$, by considering the coefficients of

- $F_{1,j}(Y|X)$, we have $\sum_{u=1}^d \alpha_{1,j}^{(u)} X^{(u)} + \phi_{1,j} = 0$ for almost surely $X$, indicating that $\alpha_{1,j}^{(u)} = \phi_{1,j} = 0$ for all $u \in [d]$;

- $F_{0,j}(Y|X)$, we have $\xi_j + \sum_{u=1}^d \alpha_{1,j}^{(u)} \cdot \frac{X^{(u)}}{1+\exp(-(a_j^*)^\top X - b_j^*)} + \frac{\phi_{1,j}}{1+\exp(-(a_j^*)^\top X - b_j^*)} = 0$ for almost surely $X$. Since $\alpha_{1,j}^{(u)} = \phi_{1,j} = 0$ for all $u \in [d]$, we also get $\xi_j = 0$.

- $F_{2,j}(Y|X)$, we have $\beta_{1,j} = 0$.

For $j \in [N^*]$ such that $|\mathcal{C}_j| > 1$, by considering the coefficients of

- $F_{1,j}(Y|X)$, we have

$$\sum_{u=1}^d \alpha_{1,j}^{(u)} X^{(u)} + \phi_{1,j} + \frac{2\sum_{u,v=1}^d \frac{\alpha_{2,j}^{(uv)}}{1+1_{\{u=v\}}} X^{(u)} X^{(v)}}{1 + \exp(-(a_j^*)^\top X - b_j^*)} + \frac{\phi_{2,j} + 2\sum_{u=1}^d \gamma_{2,j}^{(u)} X^{(u)}}{1 + \exp(-(a_j^*)^\top X - b_j^*)} = 0,$$

for almost surely $X$. Since the set

$$\left\{ 1,\ X^{(u)},\ \frac{1}{1 + \exp(-(a_j^*)^\top X - b_j^*)},\ \frac{X^{(u)}}{1 + \exp(-(a_j^*)^\top X - b_j^*)}, \right.$$

$$\left. \frac{X^{(u)} X^{(v)}}{1 + \exp(-(a_j^*)^\top X - b_j^*)} : u, v \in [d] \right\}$$

is linearly independent w.r.t $X$, we deduce $\alpha_{1,j}^{(u)} = \phi_{1,j} = \alpha_{2,j}^{(uv)} = \phi_{2,j} = \gamma_{2,j}^{(u)} = 0$ for all $u, v \in [d]$.

- $F_{0,j}(Y|X)$, we have

$$\xi_j + \frac{\sum_{u=1}^d \alpha_{1,j}^{(u)} X^{(u)} + \phi_{1,j}}{1 + \exp(-(a_j^*)^\top X - b_j^*)}$$

$$+ \frac{\sum_{u,v=1}^d \frac{\alpha_{2,j}^{(uv)}}{1+1_{\{u=v\}}} X^{(u)} X^{(v)}}{1 + \exp(-(a_j^*)^\top X - b_j^*)} + \frac{\sum_{u=1}^d \gamma_{2,j}^{(u)} X^{(u)} + \frac{1}{2}\phi_{2,j}}{1 + \exp(-(a_j^*)^\top X - b_j^*)} = 0,$$

for almost surely $X$. Since $\alpha_{1,j}^{(u)} = \phi_{1,j} = \alpha_{2,j}^{(uv)} = \phi_{2,j} = \gamma_{2,j}^{(u)} = 0$ for all $u, v \in [d]$, we get $\xi_j = 0$.

- $F_{3,j}(Y|X)$, we have $\sum_{u=1}^{d} \frac{1}{2}\gamma_{1,j}^{(u)} X^{(u)} + \frac{1}{2}\gamma_{3,j} = 0$ for almost surely $X$, indicating that $\gamma_{1,j}^{(u)} = \gamma_{3,j} = 0$ for all $u \in [d]$;

- $F_{2,j}(Y|X)$, we have

$$\frac{1}{2}\beta_{1,j} + \sum_{u,v=1}^{d} \frac{\alpha_{2,j}^{(uv)} X^{(u)} X^{(v)}}{1 + 1_{\{u=v\}}} + \frac{1}{2}\phi_{2,j} + \sum_{u=1}^{d} \gamma_{2,j}^{(u)} X^{(u)} + \frac{1}{2}\frac{\sum_{u=1}^{d} \gamma_{1,j}^{(u)} X^{(u)} + \gamma_{3,j}}{1 + \exp(-(a_j^*)^\top X - b_j^*)} = 0,$$

for almost surely $X$. Since $\alpha_{2,j}^{(uv)} = \phi_{2,j} = \gamma_{2,j}^{(u)} = \gamma_{1,j}^{(u)} = \gamma_{3,j} = 0$ for all $u, v \in [d]$, we also get $\beta_{1,j} = 0$.

- $F_{4,j}(Y|X)$, we have $\beta_{2,j} = 0$.

Putting the above results together, we have $\xi_j = \alpha_{1,j}^{(u)} = \phi_{1,j} = \beta_{1,j} = \alpha_{2,j}^{(uv)} = \phi_{2,j} = \beta_{2,j} = \gamma_{1,j}^{(u)} = \gamma_{2,j}^{(u)} = \gamma_{3,j} = 0$ for all $j \in [N^*]$ and $u, v \in [d]$. This contradicts the fact that at least one among them is different from zero. Consequently, we achieve the local part in equation (21).

## K.3 PROOF OF PROPOSITION 3.1

In this proof, we first present some fundamental results on the density estimation problem for M-estimators in van de Geer (2000) in Appendix K.3.1, and then provide the main proof in Appendix K.3.2.

### K.3.1 PRELIMINARIES

To streamline our discussion, let us introduce some necessary concepts from the empirical process theory. In particular, let $\mathcal{P}_k(\Theta)$ be the set of all conditional densities with respect to mixing measures in $\mathcal{G}_N(\Theta)$, i.e.

$$\mathcal{P}_N(\Theta) := \{p_G(Y|X) : G \in \mathcal{G}_N(\Theta)\}.$$

Additionally, we also consider two following variants of the set $\mathcal{P}_N(\Theta)$:

$$\overline{\mathcal{P}}_k(\Theta) := \{p_{(G+G_*)/2}(Y|X) : G \in \mathcal{G}_N(\Theta)\},$$
$$\overline{\mathcal{P}}_N^{1/2}(\Theta) := \{p_{(G+G_*)/2}^{1/2}(Y|X) : G \in \mathcal{G}_N(\Theta)\}.$$

Next, we define for each $\delta > 0$ a Hellinger ball centered around the true conditional density $p_{G_*}(Y|X)$ and intersect with the set $\overline{\mathcal{P}}_N^{1/2}(\Theta)$ as below

$$\overline{\mathcal{P}}_N^{1/2}(\Theta, \delta) := \{p^{1/2}(Y|X) \in \overline{\mathcal{P}}_N^{1/2}(\Theta) : h(p_G, p_{G_*}) \leq \delta\}.$$

Moreover, the size of this Hellinger ball is quantified by the following term:

$$\mathcal{J}_B(\delta, \overline{\mathcal{P}}_N^{1/2}(\Theta, \delta)) := \int_{\delta^2/2^{13}}^{\delta} H_B^{1/2}(t, \overline{\mathcal{P}}_N^{1/2}(\Theta, t), \|\cdot\|_2) dt \vee \delta, \tag{24}$$

where $H_B(t, \overline{\mathcal{P}}_N^{1/2}(\Theta, t), \|\cdot\|_2)$ stands for the bracketing entropy of $\overline{\mathcal{P}}_N^{1/2}(\Theta, t)$ under the $L^2$-norm, and $t \vee \delta := \max\{t, \delta\}$. Now, we are ready to recall the results in van de Geer (2000).

**Lemma K.1** (Theorem 7.4, van de Geer (2000)). *Take $\Psi(\delta) \geq \mathcal{J}_B(\delta, \overline{\mathcal{P}}_N^{1/2}(\Theta, \delta))$ such that $\Psi(\delta)/\delta^2$ is a non-increasing function of $\delta$. Then, for a universal constant $c$ and $\sqrt{n}\delta_n^2 \geq c\Psi(\delta_n)$, we achieve that*

$$\mathbb{P}\Big(\mathbb{E}_X[h(p_{\widehat{G}_n}(\cdot|X), p_{G_*}(\cdot|X))] > \delta\Big) \leq c \exp(-n\delta^2/c^2),$$

*for any $\delta \geq \delta_n$.*

Proof of Lemma K.1 is available in van de Geer (2000). Apart from this result, we also need to introduce the upper bounds of the covering number $N(\varepsilon, \mathcal{P}_N(\Theta), \|\cdot\|_\infty)$ and the bracketing entropy $H_B(\varepsilon, \mathcal{P}_N(\Theta), \|\cdot\|_2)$ as follows:

**Lemma K.2.** *Suppose that $\Theta$ is a bounded set, then we have for any $\varepsilon \in (0, 1/2)$ that*

*(a)* $\log N(\varepsilon, \mathcal{P}_N(\Theta), \|\cdot\|_\infty) \lesssim \log(1/\varepsilon)$;

*(b)* $H_B(\varepsilon, \mathcal{P}_N(\Theta), \|\cdot\|_2) \lesssim \log(1/\varepsilon)$.

*Proof of Lemma K.2.* **Part (a).** Recall that $\Theta$ is a compact set, then there exists an $\varepsilon$-cover, which we denote as $\overline{\Theta}_\varepsilon$. Moreover, it can be verified that $|\overline{\Theta}_\varepsilon| \leq \mathcal{O}(\varepsilon^{-(d_2+1)N})$. Next, for each mixing measure $G = \sum_{i=1}^N \delta_{(W_{e_i}, \nu_i)} \in \mathcal{G}_N(\Theta)$, we consider another one $\overline{G} = \sum_{i=1}^N \delta_{(\overline{W}_{e_i}, \overline{\nu}_i)}$, where $(\overline{W}_{e_i}, \overline{\nu}_i) \in \overline{\Theta}_\varepsilon$ is the closest point to $(W_{e_i}, \nu_i)$ in this set for any $i \in [N]$. Subsequently, we demonstrate that the set

$$\mathcal{Q} := \left\{ p_{\overline{G}}(Y|X) : (\overline{W}_{e_i}, \overline{\nu}_i) \in \overline{\Theta}_\varepsilon, \forall i \in [N] \right\}$$

is an $\varepsilon$-cover of the metric space $(\mathcal{P}_N(\Theta), \|\cdot\|_\infty)$. In other words, we need to show that for any $p_G(Y|X) \in \mathcal{P}_N(\Theta)$, there exists some density $p_{\overline{G}}(Y|X) \in \mathcal{Q}$ such that $\|p_G - p_{\overline{G}}\|_\infty \lesssim \varepsilon$.

Next, we decompose the term $T_n(Y|X) := \left[ \sum_{j=1}^N \exp(\log(1 + \exp(g(X, \overline{W}_{e_j})))) \right] \cdot [p_G(Y|X) - p_{\overline{G}}(Y|X)]$ as

$$T_n(Y|X) = \sum_{i=1}^N \exp(\log(1 + \exp(g(X, W_{e_i})))) \Big[ f(Y|g(X, W_{e_i}), \nu_i) - f(Y|g(X, \overline{W}_{e_i}), \overline{\nu}_i) \Big]$$

$$+ \sum_{i=1}^N \Big[ \exp(\log(1 + \exp(g(X, W_{e_i})))) - \exp(\log(1 + \exp(g(X, \overline{W}_{e_j})))) \Big] \cdot \Big[ f(Y|g(X, \overline{W}_{e_i}), \overline{\nu}_i) - p_G(Y|X) \Big].$$

As $\Theta$ and $\mathcal{X}$ are bounded, we may assume that $\exp(\log(1 + \exp(g(X, W_{e_i})))) \leq B_1$ and $|f(Y|g(X, \overline{W}_{e_i}), \overline{\nu}_i) - p_G(Y|X)| \leq B_2$ for some positive constants $B_1, B_2$. Thus, we obtain that

$$|T_n(Y|X)| \lesssim \sum_{i=1}^N B_1 \cdot \Big[ \|W_{e_i} - \overline{W}_{e_i}\| + |\nu_i - \overline{\nu}_i| \Big] + \sum_{i=1}^N B_2 \cdot \|W_{e_i} - \overline{W}_{e_i}\| \lesssim \varepsilon.$$

Additionally, since the term $\sum_{j=1}^K \exp(|g(X, \overline{W}_{e_j})|)$ is bounded, we obtain $|p_G(Y|X) - p_{\overline{G}}(Y|X)| \lesssim \varepsilon$ for almost surely $(X, Y)$, or equivalently,

$$\|p_G - p_{\overline{G}}\|_\infty = \sup_{(X,Y) \in \mathcal{X} \times \mathcal{Y}} |p_G(Y|X) - p_{\overline{G}}(Y|X)| \lesssim \varepsilon.$$

This result indicates that $\mathcal{Q}$ is an $\varepsilon$-cover of the metric space $(\mathcal{P}_N(\Theta), \|\cdot\|_\infty)$. Therefore, we get

$$N(\varepsilon, \mathcal{P}_N(\Theta), \|\cdot\|_\infty) \leq |\overline{\Theta}_\varepsilon| \leq \mathcal{O}(\varepsilon^{-(d_2+1)N}),$$

or equivalently,

$$\log N(\varepsilon, \mathcal{P}_N(\Theta), \|\cdot\|_\infty) \leq |\overline{\Theta}_\varepsilon| \lesssim \log(1/\varepsilon).$$

**Part (b).** Firstly, we will derive an upper bound for the Gaussian experts $f(Y|g(X, W_e), \nu)$. Since $\Theta$ is a compact set, we have $|g(X, W_e)| \leq M_1$ and $M_2 \leq \nu \leq M_3$ for any $X \in \mathcal{X}$ and $(W_e, \nu) \in \Theta$. Then, it follows that $f(Y|g(X, W_e), \nu) \leq B(Y|X)$, where

$$B(Y|X) := \begin{cases} \dfrac{1}{\sqrt{2\pi M_2}} \exp(-Y^2/(8M_3^2)), & \text{for } |Y| \geq 2M_1 \\ \dfrac{1}{\sqrt{2\pi M_2}}, & \text{for } |Y| < 2M_1, \end{cases}$$

for any $X \in \mathcal{X}$. Next, let $\eta \leq \varepsilon$ be some positive constant that we choose later, then we denote $\{\pi_1, \pi_2, \ldots, \pi_N\}$ as an $\eta$-cover over $\mathcal{P}_N(\Theta)$. Based on this cover, we build the following brackets

$L_i(Y|X) := \max\{\pi_i(Y|X) - \eta, 0\}$ and $U_i(Y|X) := \max\{\pi_i(Y|X) + \eta, B(Y|X)\}$, for any $i \in [N]$. We can validate that $\mathcal{P}_N(Y|X) \subseteq \bigcup_{i=1}^N [L_i(Y|X), U_i(Y|X)]$ and $U_i(X,Y) - L_i(X,Y) \leq \min\{2\eta, B(Y|X)\}$. As a result, we have

$$\|U_i - L_i\|_2 = \Big(\int [U_i(Y|X) - L_i(Y|X)]^2 \mathrm{d}(X,Y)\Big)^{1/2} \leq 2\eta.$$

The above result implies that

$$H_B(2\eta, \mathcal{P}_N(\Theta), \|\cdot\|_2) \leq \log N(\eta, \mathcal{P}_N(\Theta), \|\cdot\|_\infty) \lesssim \log(1/\eta).$$

Then, by setting $\eta = \varepsilon/2$, we arrive at

$$H_B(\varepsilon, \mathcal{P}_N(\Theta), \|\cdot\|_1) \lesssim \log(1/\varepsilon).$$

Hence, the proof is completed. $\qquad\square$

### K.3.2 Main Proof

Since $\overline{\mathcal{P}}_N^{1/2}(\Theta, t) \subset \overline{\mathcal{P}}_N^{1/2}(\Theta)$ for any $t > 0$, we have

$$H_B(t, \overline{\mathcal{P}}_N^{1/2}(\Theta, t), \|\cdot\|_2) \leq H_B(t, \overline{\mathcal{P}}_N^{1/2}(\Theta), \|\cdot\|_2) = H_B(t/\sqrt{2}, \overline{\mathcal{P}}_N(\Theta), h), \qquad (25)$$

where the last equality is due to the relationship between the Hellinger distance $h$ and the $L^2$-norm. Note that for any two mixing measure $G$ and $G'$, Lemma 4.2 in van de Geer (2000) indicates that

$$h^2\Big(\frac{1}{2}p_G + \frac{1}{2}p_{G_*}, \frac{1}{2}p_{G'} + \frac{1}{2}p_{G_*}\Big) \leq \frac{1}{2}h^2(p_G, p_{G'}),$$

which yields $H_B(t/\sqrt{2}, \overline{\mathcal{P}}_N(\Theta), h) \leq H_B(t, \mathcal{F}_{k_1,k_2}(\Theta), h)$. This result together with equation equation (25) implies that

$$H_B(t, \overline{\mathcal{P}}_N^{1/2}(\Theta, t), \|\cdot\|_2) \leq H_B(t, \mathcal{P}_N(\Theta), h).$$

From equation (24) and part (b) of Lemma K.2, we have that

$$\begin{aligned}
\mathcal{J}_B(\delta, \overline{\mathcal{P}}_N^{1/2}(\Theta, \delta)) &= \int_{\delta^2/2^{13}}^\delta H_B^{1/2}(t, \overline{\mathcal{P}}_N^{1/2}(\Theta, t), \|\cdot\|_2) \mathrm{d}t \vee \delta \\
&\leq \int_{\delta^2/2^{13}}^\delta H_B^{1/2}(t, \overline{\mathcal{P}}_N^{1/2}(\Theta, t), h) \mathrm{d}t \vee \delta \\
&\lesssim \int_{\delta^2/2^{13}}^\delta \log(1/t) \mathrm{d}t \vee \delta.
\end{aligned}$$

Next, let $\Psi(\delta) = \delta\sqrt{\log(1/\delta)}$, then it can be verified that $\Psi(\delta)/\delta^2$ is a non-increasing function of $\delta$. Furthermore, the above result indicates that $\Psi(\delta) \geq \mathcal{J}_B(\delta, \widetilde{\mathcal{F}}_{k_1,k_2}^{1/2}(\Theta, \delta), \|\cdot\|_2)$. By considering the sequence $(\delta_n)$ defined as $\delta_n := \sqrt{\log(n)/n}$, we have $\sqrt{n}\delta_n^2 \geq c\Psi(\delta_n)$ for some universal constant $c > 0$. It follows from Lemma K.1 that

$$\mathbb{P}\Big(\mathbb{E}_X[h(p_{\widehat{G}_n}(\cdot|X), p_{G_*}(\cdot|X))] > C\sqrt{\log(n)/n}\Big) \lesssim \exp(-c\log(n)),$$

for some universal constant $C > 0$ depending only on $\Theta$. Since the Total Variation distance is upper bounded by the Hellinger distance, we deduce

$$\mathbb{P}\Big(\mathbb{E}_X[V(p_{\widehat{G}_n}(\cdot|X), p_{G_*}(\cdot|X))] > C\sqrt{\log(n)/n}\Big) \lesssim \exp(-c\log(n)),$$

or equivalently,

$$\mathbb{E}_X[V(p_{\widehat{G}_n}(\cdot|X), p_{G_*}(\cdot|X))] = \mathcal{O}_P(\sqrt{\log(n)/n}).$$

Hence, the proof is completed.

## L  LLM Usage

We affirm that LLMs did not play a significant role in the development of this work, to the extent that they could be regarded as an author. No content generation, ideation, or technical writing assistance was delegated to LLMs.

## M  Broader Impact

Although our work mostly contributes to the machine learning literature, it also draws inspiration from biology and neuroscience. Specifically, the competition mechanism is rooted in biology, has been studied in neuroscience, and has motivated a few machine learning algorithms. Our work contributed a theoretically grounded algorithm to train large-scale SMoE models, which could potentially push the frontier of the next LLM generation. Lastly, working with large models requires rather costly resources. We took serious precautions during the development of this work, including providing a guideline for hyper-parameter selection, and conducting a single experiment using the same random seed to ensure the results are reliable at a low cost.

