# OpenReview forum: "CompeteSMoE - Statistically Guaranteed Mixture of Experts Training via Competition"
_ICLR.cc/2026/Conference — Submitted to ICLR 2026_

### Official Review · Reviewer_EjcQ · 2025-10-20

**Soundness:** 3
**Presentation:** 2
**Contribution:** 3
**Rating:** 8
**Confidence:** 2

**Summary:**

The authors propose CompeteSMoE, a SMoE training algorithm based on a competition mechanism, aimed at addressing the suboptimal issue of separation between expert computation and routing decisions in the routing process of traditional SMoE. Inspired by the Winner-take-all principle, its core idea is to activate all experts and select the Top-K experts based on their neural responses. Theoretically, it is proven that this mechanism has better sample efficiency and convergence rate than traditional softmax routing. Experiments are conducted on a wide range of benchmarks, demonstrating the effectiveness of the scheme.

**Strengths:**

1 The benchmarks covered in the experiments are relatively comprehensive, including both vision and language models of different types, providing strong support for the generalizability of the scheme.

2 The authors provide detailed hyperparameter configurations, making the reproducibility of the method convincing.

3 It provides statistical guarantees for the competition mechanism in SMoE. Through the convergence analysis of Gaussian MoE models, the convergence rate of the Total Variation distance for density estimation is derived, and a lower bound is established based on Voronoi loss, filling the gap in the theoretical analysis of the competition mechanism.

**Weaknesses:**

1 Regarding the hyperparameters ω and A used in this manuscript, what is their hyperparameter sensitivity for LLMs of different sizes?

2 How does the change in the total number of experts and the number of activated experts affect the method proposed in this paper?

**Questions:**

see weakness

---

> ### Author Response · Authors · 2025-11-22
> **Response to Reviewer EjcQ**
>
> We sincerely thank Reviewer EjcQ for the thoughtful evaluation and for recognizing the **breadth of our experiments** and the **statistical guarantees** underlying the competition mechanism. We address the reviewer’s questions and concerns below:
>
> ---
>
> **\[Q1\] Regarding the hyperparameters ω and A used in this manuscript, what is their hyperparameter sensitivity for LLMs of different sizes?**
>
> We appreciate the reviewer’s insightful question regarding hyperparameter sensitivity.
>
> **Regarding $\omega$.** Across all our experiments, we find that the hyperparameter $\omega$ exhibits low sensitivity to model scale. We use the same default value for $\omega$ in four substantially different architectures LLM pretraining at 1B (Table 2, Section 5.2) and 0.3B (Figure 8, Appendix D.3), as well as VLM training at 5.1B (Table 1, Section 5.2) and 4.1B (Table 13, Appendix F). In all cases, this setting yields stable training and clear improvements over SMoE baselines, indicating that $\omega$ does not require size-specific tuning.
>
> **Regarding $A_{max}$.** To systematically assess its sensitivity across a broader range of values, we conducted ablation studies on a smaller 0.3B-parameter language model using $A_{max}​={3,6,9}$ with $N=24$ and $K=8$ experts on 13B tokens (see Table 1 below):
>
> |  | SMoE | $A_{max}= 3$ | $A_{max}= 6$ | $A_{max}= 9$ |
> | :---- | :---- | :---- | :---- | :---- |
> | Avg. Acc  | 50.39 | 50.95 | 51.33 | 51.11 |
>
> *Table 1\. Comparison of CompeteSMoE performance across maximum activation thresholds $A_{max} \in {3, 6, 9}$ on eight benchmarks.*
>
> As shown in Table 1, CompeteSMoE consistently outperforms SMoE baselines across all tested $A_{max}$ values. Performance peaks at $A_{max}=6$, with a substantial improvement from 3 to 6, and only a modest, plateauing gain from 6 to 9\. This trend demonstrates that as $A_{max}$ increases, both performance and stability improve the differences between settings become smaller, indicating greater robustness. Consistently, we also adopted $A_{max}=6$ for the 1B language model pretraining experiment in Table 2, Section 5.2, where it maintained the best performance. These findings further support our recommendation regarding the choice of $A_{max}$ in this manuscript.
>
> **Overall,** these findings confirm that CompeteSMoE is robust to both $\omega$ and $A_{max}$, consistently achieving strong results without the need for extensive hyperparameter tuning across different model scales. For robust and consistent performance, we recommend using the values of $\omega$ and $A_{\max}$ as specified in Section 2.4 and Appendix H.3.
>
> ---
>
> **\[Q2\] How does the change in the total number of experts and the number of activated experts affect the method proposed in this paper?**
>
> | Steps | K=2 CompeteSMoE | K=2 SMoE | K=2 Diff | K=4 CompeteSMoE | K=4 SMoE | K=4 Diff | K=8 CompeteSMoE | K=8 SMoE | K=8 Diff |
> | :---- | :---- | :---- | :---- | :---- | :---- | :---- | :---- | :---- | :---- |
> | 40000 | 47.17 | 46.61 | \+0.55 | 46.86 | 46.93 | \-0.07 | 47.51 | 47.14 | \+0.37 |
> | 80000 | 47.96 | 47.47 | \+0.49 | 48.21 | 47.76 | \+0.45 | 48.22 | 47.91 | \+0.31 |
> | 120000 | 49.37 | 48.65 | \+0.72 | 49.40 | 49.15 | \+0.25 | 49.46 | 49.00 | \+0.47 |
> | 160000 | 49.88 | 49.59 | \+0.29 | 50.41 | 49.61 | \+0.80 | 50.47 | 50.11 | \+0.36 |
> | 200000 | 50.09 | 49.75 | \+0.34 | 50.71 | 50.24 | \+0.47 | 51.33 | 50.39 | \+0.94 |
>
> *Table 2\. Comparison of average accuracy across eight benchmarks between SMoE and CompeteSMoE for expert configurations $K \\in {2, 4, 8}$ with $N \= 24$.*
>
> Thank you for this question. To explore this aspect of SMoE, we conducted a controlled ablation on a 0.3B-parameter language model pretrained on a 13B-token corpus (an overtrained setting under the Chinchilla rule). We vary the number of activated experts while keeping the total number fixed, evaluating $K \\in {2, 4, 8}$ with $N \= 24$ expert configurations (Figure 8, Appendix D.3). In this rebuttal, we provide the numerical values in table format in Table 2\.
>
> Across all three sparsity levels, the performance curves show a consistent and meaningful advantage for CompeteSMoE over standard SMoE throughout the entire training trajectory. This trend indicates that the method is inherently robust to changes in the degree of sparsity, and its improvements do not depend on a particular K/N ratio. These findings demonstrate that CompeteSMoE remains effective under substantially different expert-activation settings, highlighting that its benefits arise from the underlying mechanism rather than from specific architectural choices.

---

### Official Review · Reviewer_WUdu · 2025-10-28

**Soundness:** 3
**Presentation:** 3
**Contribution:** 3
**Rating:** 6
**Confidence:** 4

**Summary:**

The paper introduces CompeteSMoE, a new approach to training the routing module in Sparse Mixture of Experts (SMoE) models. The method builds on a competition mechanism inspired by the winner-take-all principle. Periodically during training, all experts are activated for each token, and routing decisions are made based on the magnitude of their neural responses. These competition-based assignments serve as supervision signals for the router, which is trained to approximate the winner-take-all routing behavior. The authors propose a practical algorithm with scheduled router training and demonstrate promising empirical performance on both visual instruction tuning (VIT) and language pretraining tasks, outperforming several existing SMoE variants.

The breadth of analysis is satisfactory, though the language pretraining section would benefit from additional experiments that show whether the efficacy of CompeteSMoE holds up to scale, as well as the most popular modern architectural choices (more in Weaknesses  and Questions)

**Strengths:**

- The paper has a clear motivation and a relatively straightforward implementation of the proposed idea.
- The presentation is clear and easy to follow.
- The formulation of periodical winner-takes-all for SMoE routing decision, alongside having the router emulate those deisions appears novel, to the best of my knowledge.
- Experiments cover both vision and language domains.
- The paper includes thorough ablations analyzing the individual contributions of the competition mechanism and the diversity loss.
- The experiment that rotates routing choices to test whether the model benefits from routing decisions or behaves randomly is an interesting addition.
- The results are generally positive, though the effect sizes on benchmarks are small.
- The analysis of wall-clock time and peak memory during training shows minimal overhead for the proposed method.

**Weaknesses:**

- The language pretraining setup uses 13B tokens for a 1B-parameter model. According to the Chinchilla rule of thumb (around 20 tokens per parameter), this corresponds to a moderate undertraining regime, while many modern models are heavily overtrained. See e.g. [Qwen3 Technical Report](https://arxiv.org/pdf/2505.09388)
- The chosen configuration of `topk = 8` out of 24 experts is not the most typical. Recent trends favor sparser settings where only a few experts are active out of a large total. An ablation over different sparsity levels would make the empirical evaluation more complete.
- The observed effect sizes on benchmarks are small. The improvements are more consistent in the vision domain, but those experiments rely on upcycled models, which complicates the interpretation relative to the more straightforward from-scratch language pretraining setting.
- Limited ablation of hyperparameters: while ad-hoc guidelines for α, β, γ, and Amax are provided, the robustness of results to these choices is not extensively analyzed in the main text.
- Minor issues with typos and unclear formulations (e.g., “200” instead of “200K” around line 1545, based on the supplementary code).

**Questions:**

- To strengthen the paper on the language pretraining front:
  1. It would be helpful to investigate how the efficiency of CompeteSMoE changes with training horizon, model size, and Mixture-of-Experts architecture configurations (total number of experts and number of selected experts).
  2. The effect size on benchmarks is small, and given the scale of experiments it's unclear how much of the variation is noise. It would be very informative to see evaluation benchmarks, training loss and validation loss trajectories throughout the entire training run, ideally for different configurations from suggestion 1.
- Why was the diversity loss applied during pretraining from scratch, since the model was not upcycled?
- Could the authors clarify the meaning of the following passage (lines 129–134)? It is somewhat unclear how joint learning of the task loss and competition policy alleviates the sample inefficiency issue or why this makes training feasible on limited hardware.
- It would be very valuable to understand the impact of frequency of competition on model quality. If memory is an issue, the authors can consider training a smaller model and thus being able to vary the competition probability up.
- Teaching the router to pick magnitude-maximising experts biases the model toward bigger activations. It would be very interesting to see the evolution of mean magnitude of tokens in the residual stream after each MoE layer, and how it evolves over time, and how all that changes from baseline SMoE to CompeteSMoE. The reason to care about this is that large activations can be difficult to quantize, which comes up in low-precision training.
- The paper would benefit from an analysis of the evolution of routing scores through training

---

> ### Author Response · Authors · 2025-11-22
> **Response to Reviewer WUdu (Part 1)**
>
> We thank Reviewer WUdu for the thoughtful review and for recognizing the **clarity**, **novelty**, **broad experimental coverage**, and thorough **ablations of our work.** We address the reviewer’s questions and concerns below.
>
> ---
>
> **\[W1, W2, Q1\]. Limitations of Experimental Design: Undertraining and Lack of Sparsity Ablation**
>
> > The language pretraining setup uses 13B tokens for a 1B-parameter model[..]
>
> > [..] Recent trends favor sparser settings where only a few experts are active out of a large total.[..]
>
> > To strengthen the paper on the language pretraining front[..]
>
> We thank the reviewers for highlighting important points regarding training regime, sparsity choices, and the robustness of our conclusions across configurations.
>
> | Steps | K=2 CompeteSMoE | K=2 SMoE | K=2 Diff | K=4 CompeteSMoE | K=4 SMoE | K=4 Diff | K=8 CompeteSMoE | K=8 SMoE | K=8 Diff |
> | :---- | :---- | :---- | :---- | :---- | :---- | :---- | :---- | :---- | :---- |
> | 40000 | 47.17 | 46.61 | \+0.55 | 46.86 | 46.93 | \-0.07 | 47.51 | 47.14 | \+0.37 |
> | 80000 | 47.96 | 47.47 | \+0.49 | 48.21 | 47.76 | \+0.45 | 48.22 | 47.91 | \+0.31 |
> | 120000 | 49.37 | 48.65 | \+0.72 | 49.40 | 49.15 | \+0.25 | 49.46 | 49.00 | \+0.47 |
> | 160000 | 49.88 | 49.59 | \+0.29 | 50.41 | 49.61 | \+0.80 | 50.47 | 50.11 | \+0.36 |
> | 200000 | 50.09 | 49.75 | \+0.34 | 50.71 | 50.24 | \+0.47 | 51.33 | 50.39 | \+0.94 |
>
> *Table 1\. Comparison of average accuracy across eight benchmarks between SMoE and CompeteSMoE for expert configurations $K \\in \{2, 4, 8\}$ with $N \= 24$.*
>
> **MoE Sparsity Choices:**
>
> We would like to highlight that there are no “standard” configurations for the number of experts (N) or the active subset (K), the field has not converged on any canonical K/N ratio for sparse MoE architectures. For example, Mixtral-8×7B\[2\] uses 2/8; Phi-3.5-MoE\[3\] adopts 2/16; DBRX-Base\[4\] uses 4/16; UniDense\[5\] applies 8/24; and FLAME-MoE\[6\] deploys 10/64. Importantly, all our comparisons between CompeteSMoE and baselines are conducted under the same configuration for fairness.
>
> **Training Horizon and (Under)/Over-Training Regimes:**
> We acknowledge that our 1B-parameter model, trained on 13B tokens, falls into a moderately undertrained regime according to the Chinchilla scaling law. Thus, in the rebuttal period, we conducted additional experiments by pre-training smaller models under the over-training regime of the Chinchilla scaling law.
>
> **To directly address concerns regarding both training horizon and sparsity,** we conducted additional experiments:
>
> - We trained a 0.3B-parameter model on the same 13B-token corpus, a regime that is substantially overtrained relative to the Chinchilla optimum for this model size.
>
> - This 0.3B model has a total $N \= 24$ experts and we considered three sparsity settings  ($K=2,4,8$) to examine the impact of experts’ selection sparsity.
>
> The training curves of this experiment are reported in Figure 8, Appendix D.3. In this rebuttal, we provide the numerical values in table format in Table 1\.
>
> The results showed that, across all configurations including training horizon, model scale, and sparsity, CompeteSMoE consistently outperforms standard SMoE. Thus, the improvements of CompeteSMoE remain robust across experiment settings. This consistency strongly indicates that our gains are due to genuine algorithmic advances, rather than being artifacts of a particular model, dataset, or architectural choice.
>
> ---
>
> **\[W3\] Clarifying Vision-Domain Results under Upcycled vs. Pretrained Models**
>
> > [..] The improvements are more consistent in the vision domain[..]
>
> We appreciate the reviewer’s observation that improvements in the vision domain appear more consistent, though they were obtained under an upcycled model setup. While the current vision experiments reflect the latter, Appendix F includes additional results where all MoE layers in the vision-language model are initialized from scratch. These results show that CompeteSMoE remains effective even without upcycled weights, supporting its robustness across both pretraining and upcycling setups.
>
> ---
>
> **\[Q4, W4, W5\] Limited hyperparameter robustness analysis and error types**
>
> > [..] very valuable to understand the impact of frequency of competition.[..]
>
> > Limited ablation of hyperparameter [..]
>
> > Minor issues with typos [..]
>
> We believe this concern may arise from a minor misunderstanding. In the original submission, the hyperparameter sensitivity analysis was already included in the appendix. In the revised version, we have moved this entire analysis into the main paper (now Section 5.4) and expanded it with additional experiments for completeness. The updated section provides a detailed examination of key hyperparameters and shows that CompeteSMoE maintains stable performance across a broad range of settings. We also thank the reviewer for pointing out minor notation issues (e.g., “200” → “200K”), which have now been corrected.

---

> ### Author Response · Authors · 2025-11-22
> **Response to Reviewer WUdu (Part 2)**
>
> **\[Q2\] Why was the diversity loss applied during pretraining from scratch, since the model was not upcycled?**
>
> Thank you for highlighting this important aspect.
>
> | Benchmark | CompeteSMoE (w/o div) | CompeteSMoE(w/ div) | Diff |
> | :---: | :---: | :---: | :---: |
> | LAMBADA | 35.83 | 36.27 | \-0.44 |
> | BLIMP | 79.70 | 80.12 | \-0.41 |
> | CBT | 88.54 | 88.89 | \-0.35 |
> | HellaSwag | 33.71 | 34.21 | \-0.50 |
> | PIQA | 62.46 | 62.89 | \-0.44 |
> | ARC-E | 37.42 | 37.46 | \-0.04 |
> | RACE | 32.90 | 32.81 | 0.10 |
> | SIQA | 37.46 | 38.02 | \-0.56 |
> | Avg. Acc | 51.00 | 51.33 | \-0.33 |
>
> *Table 2\. Comparison of model performance for CompeteSMoE trained without diversity loss (w/o div) versus with diversity loss (w/ div).*
>
> While our primary motivation for introducing the diversity loss lies in the upcycling setting, we have systematically observed that its benefits extend to the overall competition mechanism in CompeteSMoE. Specifically, diversity loss plays a crucial role in preventing expert collapse by discouraging overly similar outputs among experts, thereby fostering a more expressive and discriminative expert space as measured by the affinity score. As shown in *Table* 2, additional results on the 0.3B language model pretrained over 13B tokens demonstrate a clear drop in performance when diversity loss is omitted, underscoring its essential contribution to robust expert specialization. By consistently retaining the diversity loss even during from-scratch training we ensure that CompeteSMoE exhibits unified and stable routing dynamics across both upcycled and randomly initialized regimes. This deliberate design choice not only mitigates the risk of expert redundancy but also enhances both generalization and training stability throughout all training regimes. Thus, diversity loss is fundamental to the reliability and effectiveness of our Competition Mechanism, regardless of initialization strategy.
>
> ---
>
> **\[Q3\] Clarifying Joint Task–Policy Learning and Hardware Feasibility (Lines 129–134).**
>
> > Could the authors clarify the meaning of the following passage (lines 129–134) [..]
>
> Thank you for raising this important point. In SMoE training, the routing can be improved from two types of signal: (i) how to better assign tokens to experts; and (ii) how to adjust the experts parameter to better predict the output from inputs. While the task loss’s gradient provides information to adjust the parameters of selected experts; the non-selected experts and routing policy do not receive any signals because the TopK operator is non-differentiable. Thus, the vanilla SMoE training only provides information to update K experts for each token. In CompeteSMoE, we proposed to further match the routing policy with the competition policy. When competition is active, the router loss’s gradient provides signals to update all experts, including the ones that are not selected. Such information allows experts to learn more frequently and learn more information from each sample, thus, improving training efficiency.
>
> Empirically, from Figure 1 in the main paper, we can see that at 8 hours of training, CompeteSMoE already outperformed all baselines at their final checkpoints (\~14h).
>
> | Model                               | Task Loss | Match Competition Policy | AVG Acc | AVG Rank |
> | :---------------------------------- | :-------: | :-----------------------: | :-----: | :------: |
> | **CompeteSMoE**                     | ✓         | ✓                         | **53.21** | **1.33** |
> | CompeteSMoE — Competition Policy Only | ✗       | ✓                         | 52.84   | 2.11     |
> | SMoE                                | ✓         | ✗                         | 52.47   | 2.56     |
>
>
> *Table 3\. Ablation study showing the impact of task optimization and competition policy matching on performance across 9 benchmarks*
>
> To support this motivation, we conducted an experiment isolating the effect of each component. The results are reported in Table 3 of the rebuttal, with detailed analysis provided in Appendix C.2, jointly optimizing both the task loss and matching the competition policy yields the best performance in terms of both average accuracy and average rank, confirming the advantage of combining the two objectives. Moreover, we also observe that training the router solely to match the competition policy, without any supervision from the task loss, already outperforms the standard SMoE. This highlights the effectiveness of competition-driven learning in discovering better routing policies, even without relying on task-specific gradients. Notably, this performance gain is achieved even though the competition policy is only active in 7% of the training steps. Despite such sparse updates, the router still learns significantly better expert selection than the baseline, indicating that the competition policy provides a strong inductive bias.

---

> ### Author Response · Authors · 2025-11-22
> **Response to Reviewer WUdu (Part 3)**
>
> **\[Q5\] Magnitude-Maximizing Routing and Its Effect on Quantization**
>
> > Teaching the router to pick magnitude-maximising experts biases the model toward bigger activations.[..]
>
> We thank the reviewer for raising this insightful point. To assess whether CompeteSMoE leads to undesirable activation growth that could harm quantization, we followed the norm-based analysis used in MoEUT\[1\] and tracked the evolution of expert output magnitudes across all layers and training checkpoints (Figure 6, Appendix D.1).
>
> We found that CompeteSMoE induces slightly higher activation norms than SMoE, but this increase is (i) moderate in scale, (ii) mostly confined to middle layers (Layers 16–25), and (iii) non-monotonic across depth or time. Early and final layers show negligible change, and some deep layers even exhibit *lower* activations than baseline SMoE. These results suggest that the competition policy does not cause runaway amplification or globally inflated residuals.
>
> ---
>
> **\[Q6\] The paper would benefit from an analysis of the evolution of routing scores through training**
>
> > The paper would benefit from an analysis of the evolution of routing scores through training
>
> | Steps | Avg Expert Overlap Ratio | Avg Router Loss |
> | :---: | :---: | :---: |
> | 40000 | 0.6487 | 0.0107 |
> | 80000 | 0.7121 | 0.0058 |
> | 120000 | 0.7505 | 0.0039 |
> | 160000 | 0.7719 | 0.0031 |
> | 200000 | 0.7811 | 0.0028 |
>
> ***Table 4\.** Evolution of router learning dynamics during training in the CompeteSMoE algorithm. The first column reports the **expert overlap ratio** between the router’s Top-$K$ selections and those chosen by the competition mechanism, reflecting alignment between the two. The second column shows the **router distillation loss**, which quantifies the discrepancy between router scores and competition scores as training progresses.*
>
> Thank you for the helpful suggestion. We have added a dedicated analysis of the evolution of routing scores during training in Appendix D.2, and in this rebuttal we provide the corresponding numerical values in Table 4\. This new section reports both the expert overlap ratio and the router distillation loss across training steps, which together quantify how the router progressively aligns its routing decisions with the competition mechanism. The results clearly show a steady increase in overlap and a corresponding decrease in router loss, indicating that the router network consistently learns and stabilizes its routing behavior over time.
>
> ---
>
> **Reference:**
>
> \[1\] Csordás, Róbert, et al. "Moeut: Mixture-of-experts universal transformers." Advances in Neural Information Processing Systems 37 (2024): 28589-28614.
>
> \[2\] Mixtral-8x7B: [https://huggingface.co/mistralai/Mixtral-8x7B-v0.1](https://huggingface.co/mistralai/Mixtral-8x7B-v0.1)
>
> \[3\]  Phi-3.5-MoE : microsoft/Phi-3.5-MoE-instruct
>
> \[4\] DBRX-Base: [https://www.databricks.com/blog/introducing-dbrx-new-state-art-open-llm](https://www.databricks.com/blog/introducing-dbrx-new-state-art-open-llm)
>
> \[5\] Dong, Lintao, Wei Zhai, and Zheng-Jun Zha. "UniDense: Unleashing Diffusion Models with Meta-Routers for Universal Few-Shot Dense Prediction." Proceedings of the 32nd ACM International Conference on Multimedia. 2024\.
>
> \[6\] Kang, Hao, Zichun Yu, and Chenyan Xiong. "FLAME-MoE: A Transparent End-to-End Research Platform for Mixture-of-Experts Language Models." arXiv preprint arXiv:2505.20225 (2025).

---

### Official Review · Reviewer_6ffV · 2025-11-01

**Soundness:** 3
**Presentation:** 3
**Contribution:** 2
**Rating:** 4
**Confidence:** 3

**Summary:**

This paper introduces a novel competition-based training mechanism for SMoE models that achieves faster convergence than traditional routing. Building on this idea, the authors develop CompeteSMoE, a method that leverages competition to enhance SMoE training. To support their claims, the paper provides a theoretical convergence analysis of the proposed approach. Finally, experiments are conducted on popular vision and text benchmarks, complemented by an additional analysis of routing behavior.

**Strengths:**

Despite addressing the well-researched subfield of routing in SMoE, the authors successfully identify an original and interesting perspective on this topic. The paper is well-written and easy to follow.  A significant strength is that the authors ground their methodological claims in theoretical analysis. Furthermore, the proposed method is evaluated across both images and text, which strongly supports the paper's conclusions.

**Weaknesses:**

The claim in the abstract regarding a scalable method is not substantiated in the main text, as the experiments were conducted on a very limited scale. To support the scalability claim, additional experiments involving larger datasets and more diverse conditions would be necessary.

**Questions:**

- I am not convinced by ECR metric (Section 5.2.2 b). Intuitively, I would like router network to have a possibility to adapt to updated experts till the end of the training process. Why the authors claim that it is actually good, that the rate decaying faster?

- Section 4.2 does not appear to be directly relevant to the main objectives of the study.

- Upcycling was used as a means to bypass the costly pre-training phase. However, I believe that conducting smaller-scale experiments involving pre-training would likely yield more reliable insights.

Minor comment:
- No units in Table 4 for Train / Infer

---

> ### Author Response · Authors · 2025-11-22
> **Response to Reviewer 6ffV**
>
> We thank reviewer 6ffV for the thoughtful reading and for recognizing that CompeteSMoE offers an **original and interesting perspective**, with **clear presentation**, **solid theoretical grounding**, and **strong empirical evaluation** across both image and text modalities. Below, we address the raised concerns in detail:
>
> ---
>
> **Q1: Justification for Faster Decay in the ECR Metric and Its Implications for Router Adaptation**
> > [..] Intuitively, I would like router network to have a possibility to adapt to updated experts till the end of the training process. Why the authors claim that it is actually good, that the rate decaying faster?[..]
>
> Thank you for the acute observation. We agree that, in principle, the router should remain adaptive when the data distribution changes or when it encounters a distribution shift. Our Expert Change Rate (ECR) metric, however, is not intended to discourage such adaptivity. Instead, it is designed to measure the convergence behavior of the router on a fixed/separated dataset.
>
> Concretely, we compute ECR on some evaluation sets (in Figure 3 we used PoPE, MMStar, MMMU, and MathVista), and measure the discrepancy in experts assignment between the final checkpoint and the intermediate ones. Since the evaluation sets are static, a persistently high ECR late in training indicates that the router has high regrets: compared to the final checkpoints, the same examples keep being sent to different experts throughout training. In contrast, a faster decay in ECR means that, for in-domain inputs, the router has learned a consistent partition of the data and no longer performs unnecessary re-routing for the same samples. This does not preclude the router from adapting to truly new or out-of-domain data; it only indicates stability on the reference distribution where we measure ECR.
>
> Our observation is consistent with StableMoE \[1\], which explicitly identifies this “routing fluctuation” problem where the same input is routed to different experts during training and shows that a more stable routing strategy that mitigates such fluctuation leads to faster convergence and better overall performance.
>
> ---
>
> **Q2: “Section 4.2 does not appear to be directly relevant to the main objectives of the study.”**
>
> Thank you for your valuable feedback. We have substantially revised Section 4 to more clearly articulate its relevance to the main objectives of the study.
>
> ---
>
> **Q3: Need for Additional Small-Scale Pre-Training to Validate Upcycling**
> > [..] I believe that conducting smaller-scale experiments involving pre-training would likely yield more reliable insights.
>
> Thank you for the suggestion. Although our primary focus is on sparse upcycling, we have also performed full 1B language model pre-training from scratch, as reported in (Table 2, Section 5.2). These experiments show that CompeteSMoE consistently outperforms the standard SMoE and other baselines in the pre-training setting. Furthermore, Appendix F provides additional results where all MoE layers are initialized entirely from scratch on VLM, and CompeteSMoE remains effective under this setup. In summary, these experiments corroborate the advantage of CompeteSMoE over existing SMoE training strategies.
>
> ---
>
> **M1: “No units in Table 4 for Train / Infer”**
>
> Thank you for pointing this out. We have updated Table 4 to include proper units.
>
> ---
>
> **Reference**
>
> [1] Dai, Damai, et al. "Stablemoe: Stable routing strategy for mixture of experts." arXiv preprint arXiv:2204.08396 (2022). (ACL 2022)

---

> ### Author Response · Authors · 2025-11-25
> **Looking forward to your response**
>
> Dear Reviewer 6ffV,
>
> We would like to thank you very much for your insightful review, and we hope that our response addresses your previous concerns regarding our paper. However, as the discussion period is expected to end in the next week, please feel free to let us know if you have any further comments on our work. We would be willing to address any additional concerns from you. Otherwise, we hope that you will consider increasing your rating.
>
> Thank you again for spending time on the paper. We really appreciate it!
>
> Best regards,
>
> Authors

---

### Author Response · Authors · 2025-11-24
**General Response (Part 1/2)**

Dear Area Chairs and Reviewers,

We sincerely thank the reviewers for their careful reading, insightful feedback, and recognition of our work’s core strengths. We especially appreciate the constructive criticism, which has directly contributed to further improving the clarity, rigor, and overall quality of our manuscript.

---

**Key Strengths Highlighted by Reviewers:**

- **Contribution:** The authors successfully identify an **original and interesting** perspective on the well-researched subfield of routing in SMoE (Reviewer 6ffV). The formulation of periodical winner-takes-all for SMoE routing decision, alongside having the router emulate those decisions appears **novel** (Reviewer WUdu). The experiment that rotates routing choices (**ECR**) to test whether the model benefits from routing decisions or behaves randomly is an interesting addition (Reviewer WUdu).

- **Soundness:** **A significant strength** is that the authors ground their methodological claims in theoretical analysis (Reviewers 6ffV, EjcQ). The proposed method is evaluated across both vision and language domains, which strongly supports the paper's conclusions (Reviewers 6ffV, WUdu, EjcQ). The authors provide detailed hyperparameter configurations, making the reproducibility of the method convincing (Reviewer EjcQ).

- **Presentation:** The paper is well-written and easy to follow (Reviewers 6ffV, WUdu). The paper has a clear motivation and a relatively straightforward implementation of the proposed idea (Reviewer WUdu).

---

> ### Author Response · Authors · 2025-12-03
> **General Response (Part 2/2)**
>
> **Major Reviewer Concerns and Our Response:**
>
> - **Extensive Expansion of Experimental Design (WUdu, EjcQ, 6ffV).** In response to requests for broader validation, we have significantly extended our experimental study. New experiments include rigorous over-training regimes and evaluations across various sparsity levels, as detailed in Appendix D.3. Importantly, CompeteSMoE was benchmarked on four diverse architectures: LLM pretraining at 1B (Table 2, Section 5.2) and 0.3B (Figure 8, Appendix D.3) parameters; and VLM training at 5.1B (Table 1, Section 5.2) and 4.1B (Table 13, Appendix F) parameters. Across all configurations, **CompeteSMoE consistently outperforms existing baselines**, confirming both its robustness and broad applicability. These results reinforce the generality and effectiveness of our approach in diverse and challenging settings.
>
> - **Unified and Deepened Analysis of Training Dynamics (WUdu, EjcQ).** To meet requests for a more holistic and interpretable understanding of CompeteSMoE, we have expanded our analysis to include: (1) A comprehensive hyperparameter sensitivity study (Section 2.4, 5.4; Appendix G, H.3) now covers all relevant configurations and variations. We provide full transparency of every hyperparameter and routing setting, ensuring complete reproducibility. (2) Newly added analyses such as the evolution of expert-output magnitudes (Appendix D.1) and layer-wise router score behaviors (Appendix D.2) offer unified insights into model dynamics and decision processes. This deeper perspective allows both practitioners and theorists to more fully understand the mechanisms behind CompeteSMoE’s effectiveness.
>
> - **Clarification of algorithmic Design Choices and Metrics (6ffV, WUdu).**
>   We have further clarified several core algorithmic components in response to reviewer feedback.
>
>   **Faster ECR Decay and Expert Partitioning Stability:** In response to Reviewer 6ffV’s concern regarding the benefit of a faster Expert Change Rate (ECR) decay, we have clarified and provided reliable evidence on this point. Our analysis confirms that a faster ECR decay on fixed evaluation sets leads to more stable expert partitioning and lower routing regret, without compromising adaptivity to distribution shifts (see Q1 in the Reviewer 6ffV rebuttal). This clarification was further `endorsed by Reviewer WUdu, who described ECR as “an interesting addition” to the evaluation suite.`
>
>   **Diversity Loss During From-Scratch Pre-Training:** In response to Reviewer WUdu’s question regarding the necessity of applying diversity loss during from-scratch pre-training, we clarified its benefits in this setting. To further support our argument, we conducted an ablation study on diversity loss in language model pretraining (see Table 2, Q2 in the Reviewer WUdu rebuttal).
>
>   **Joint Optimization of Task Loss and Competition-Based Routing:** In response to Reviewer WUdu’s request for clarification on how jointly optimizing the task loss and competition-based router objectives improves training efficiency, we provided a detailed explanation and empirical validation of this approach. As demonstrated in Table 3, within our response to Q3 in the Reviewer WUdu rebuttal, this strategy yields significant practical impact and efficiency gains.
>
> ---
>
> **Summary**
>
> In summary, we have addressed the reviewers’ concerns with additional experiments and clearer explanations. We are grateful for the reviewers’ careful evaluation and firmly believe that the revised manuscript now presents a more compelling and comprehensive case for *CompeteSMoE*.
>
> Best regards,
>
> The Authors

---

### Meta-Review · Area_Chair_MYGf · 2025-12-12

**Summary:**

The authors addressed some of the reviewer’s 6ffV concerns, particularly clarifying the role of the ECR metric and adding the requested results. However, the rebuttal does not properly justify the scalability claim, nor provides evidence that the method scales beyond the small-scale experiments presented. Additionally, while the authors present pretraining experiments, the response does not clearly demonstrate that these experiments validate the upcycling conclusions.

For the reviewers WUdu and EjcQ, the authors conducted additional experiments and analyses that address most of the reviewers concerns. A few issues, such as exploration of more extreme sparsity settings and longer training, were only partly resolved, but overall the revisions strengthen the empirical foundation and clarity of the paper.

**Reviewer Concerns:**

Concerns by reviewer 6ffV are not fully addressed.

**Reviewer Scores:**

Probably EjcQ probably keep the positive score. WUdu either keep the current semi-positive score or maybe increase it. Probably, 6ffV keeps the current rate.

---

### Decision · Program_Chairs · 2026-01-26

Reject